# DiVeQ: Differentiable Vector Quantization Using the Reparameterization Trick

**Mohammad Hassan Vali[1], Tom Bäckström[2] & Arno Solin[1]**
[1]ELLIS Institute Finland & Department of Computer Science, Aalto University, Finland
[2]Department of Information and Communications Engineering, Aalto University, Finland
{mohammad.vali, tom.backstrom, arno.solin}@aalto.fi

## Abstract

Vector quantization is common in deep models, yet its hard assignments block gradients and hinder end-to-end training. We propose DiVeQ, which treats quantization as adding an error vector that mimics the quantization distortion, keeping the forward pass hard while letting gradients flow. We also present a space-filling variant (SF-DiVeQ) that assigns input to a curve constructed by the lines connecting codewords, resulting in less quantization error and full codebook usage. Both methods train end-to-end without requiring auxiliary losses or temperature schedules. In VQ-VAE image compression, VQGAN image generation, and DAC speech coding tasks across various data sets, our proposed methods improve reconstruction and sample quality over alternative quantization approaches.

## 1 Introduction

Vector quantization (VQ, Gersho & Gray, 2012) is a classical compression technique for discretizing continuous data distributions into a finite set of representative vectors, resulting in a *codebook*. Each data sample is assigned to its nearest codebook vector (*codeword*), enabling compact discrete representations. Within deep learning, VQ was popularized by the vector-quantized variational autoencoder (VQ-VAE, van den Oord et al., 2017), where discretization of latent representations yields substantial improvements in compression and generation quality. Since then, VQ has become a central component in architectures for images (Yu et al., 2021), video (Yan et al., 2021), and speech (Kumar et al., 2023), serving as a fundamental module in the tool-stack for deep neural networks (DNNs).

Mapping a continuous data sample to its closest codeword is mathematically non-differentiable. Hence, when using VQ in the computational graph of a DNN, the gradients will not pass through the VQ layer in the backward pass. For instance, in a VQ-VAE, the encoder parameters will not receive any gradients from the VQ layer. This issue is known as the *gradient collapse* problem (Vali, 2025). There exists a variety of solutions to this problem, each with its own challenges and drawbacks (see Table 1).

In this paper, we revisit differentiable vector quantization through the lens of the reparameterization trick (Kingma & Welling, 2013). We propose *Differentiable Vector Quantization* (DiVeQ), a method that models quantization as the addition of a simulated quantization error vector (see Fig. 1). The direction of this vector is aligned with the nearest codeword, while its magnitude equals the input–codeword distance, thereby preserving hard assignments in the forward pass while enabling meaningful gradient flow. Unlike prior approaches such as NSVQ (Vali & Bäckström, 2022), which

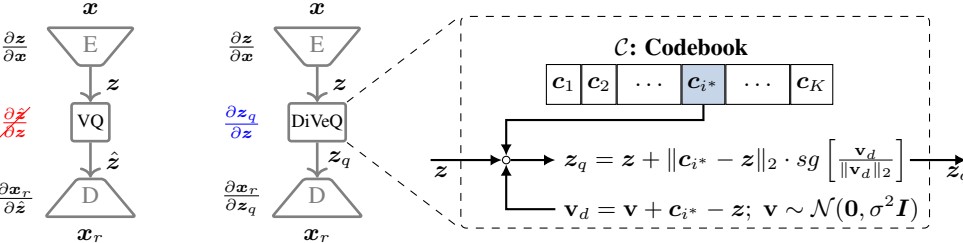

Figure 1: We replace the non-differentiable VQ operation ($\hat{z} = c_{i^*} = \arg\min_{c_j} \|z - c_j\|_2$) on the left with differentiable vector quantization (DiVeQ) on the right that lets the gradients flow.

Table 1: Summary of training properties for different VQ optimization methods in Sec. 2.

| | STE | EMA | RT | ST-GS | NSVQ | DiVeQ | SF-DiVeQ |
|---|---|---|---|---|---|---|---|
| No auxiliary loss terms | ✗ | ✗ | ✗ | ✗ | ✓ | ✓ | ✓ |
| No complicated parameter tuning | ✗ | ✗ | ✗ | ✗ | ✓ | ✓ | ✓ |
| Unbiased codebook gradients | ✗ | N/A | ✓ | ✗ | ✓ | ✓ | ✓ |
| No train–test mismatch | ✓ | ✓ | ✓ | ✗ | ✗ | ✓ | ✓ |
| End-to-end training | ✗ | ✗ | ✗ | ✓ | ✓ | ✓ | ✓ |
| Precise nearest-codeword assignment | ✓ | ✓ | ✓ | ✓ | ✗ | ✓ | ✓ |
| Not prone to codebook misalignment | ✗ | ✗ | ✗ | ✗ | ✗ | ✗ | ✓ |
| Avoiding codebook collapse | ✗ | ✗ | ✗ | ✗ | ✗ | ✗ | ✓ |

approximate the quantization error with limited directional fidelity, DiVeQ ensures that the differentiable surrogate remains geometrically consistent with the underlying nearest-neighbor operation.

Getting intuition from Vali & Bäckström (2023a) and building on this foundation, we further introduce *Space-Filling DiVeQ* (SF-DiVeQ), which extends quantization from discrete codewords to continuous line segments connecting neighboring codewords. This construction simultaneously mitigates quantization error and alleviates codebook under-utilization without requiring replacement heuristics. By quantizing along structured manifolds within the codebook, SF-DiVeQ achieves full utilization and improves representational efficiency.

We evaluate DiVeQ and SF-DiVeQ on VQ-VAE (van den Oord et al., 2017) for image compression, VQGAN (Esser et al., 2021) for image generation across CELEBA-HQ, FFHQ, AFHQ, and LSUN Bedroom and Church, and the DAC (Kumar et al., 2023) model for speech coding on the VCTK data set. Our methods consistently improve image and speech reconstruction fidelity and maintain sample quality compared to existing quantization strategies. Importantly, DiVeQ and SF-DiVeQ act as *drop-in replacements* for standard VQ layers, requiring only minimal changes to model code. Our reference implementation is available at `https://github.com/AaltoML/DiVeQ` and as a PyPI package at `https://pypi.org/project/diveq/`.

The contributions of this paper are as follows:

- We propose DiVeQ, a differentiable vector quantization technique that enables end-to-end training with hard forward assignments, avoiding auxiliary losses and complicated tunings.

- We further propose SF-DiVeQ, a space-filling variant, which quantizes along codeword connections, ensuring reduced quantization error and full codebook utilization without any auxiliary losses or codebook reinitialization. Contrary to all other methods, SF-DiVeQ avoids misalignment of latent and codebook representations.

- For VQ optimization methods prone to *codebook collapse*, we present an improved codebook replacement algorithm, achieving faster and more stable utilization than prior methods (see App. B.1). Also, we show that DiVeQ and SF-DiVeQ are advantageously applicable to other VQ variants like Residual VQ with superior performance to the other techniques (see App. C.9).

## 2   BACKGROUND AND RELATED WORK

Vector quantization (VQ) has been extensively studied in both signal processing and modern deep learning, and a wide range of strategies have been developed to overcome its non-differentiability. Since the core challenge addressed in this paper is precisely the integration of VQ into end-to-end trainable neural architectures, we provide a structured review of existing solutions. This detailed discussion is necessary because, although many methods yield partial remedies, they differ substantially in terms of optimization objectives, gradient fidelity, codebook utilization, and computational overhead. Our goal in this section is therefore twofold: *(i)* to establish a precise baseline understanding of the mechanisms underlying each class of methods, and *(ii)* to highlight their limitations in relation to the desiderata of differentiable quantization.

VQ (Gersho & Gray, 2012) clusters a continuous distribution to a limited set of codewords, by mapping a latent vector $z \in \mathbb{R}^D$ to the nearest codeword $c_{i^*} \in \mathbb{R}^D$ of a codebook $\mathcal{C} = \{c_1, \ldots, c_K\}$, where $i^* = \arg\min_j \|z - c_j\|_2$. This mapping introduces a distortion $\xi_Q = z - c_{i^*}$, known as

quantization error. Since the nearest-neighbor assignment is non-differentiable, VQ blocks gradients during backpropagation, leading to the well-known *gradient collapse* problem (Vali, 2025). To explain existing techniques, we consider the case of training a VQ codebook in a vector-quantized variational autoencoder (VQ-VAE, van den Oord et al., 2017). In VQ-VAE (Fig. 1), the input $\boldsymbol{x}$ is fed to the encoder $E$ to obtain the continuous latent variable $\boldsymbol{z}$, which is then discretized by VQ to $\hat{\boldsymbol{z}}$. The discrete latent variable $\hat{\boldsymbol{z}}$ is fed to the decoder $D$ to reconstruct the input $\boldsymbol{x}_r$:

$$\boldsymbol{z} = E(\boldsymbol{x}) \longrightarrow \hat{\boldsymbol{z}} = \mathrm{VQ}(\boldsymbol{z}) \; ; \; \hat{\boldsymbol{z}} = \boldsymbol{c}_{i^*} = \arg\min_{\boldsymbol{c}_j} \|z - \boldsymbol{c}_j\|_2 \longrightarrow \boldsymbol{x}_r = D(\hat{\boldsymbol{z}}). \tag{1}$$

Since $\arg\min$ is not differentiable ($\frac{\partial \hat{\boldsymbol{z}}}{\partial \boldsymbol{z}}$ does not exist), VQ does not pass any gradients from $\boldsymbol{x}_r$ to $\boldsymbol{x}$.

**Straight-Through Estimator (STE, Bengio et al., 2013)** deals with this issue by copying the gradients through the non-differentiable VQ function, while assuming $\frac{\partial \hat{\boldsymbol{z}}}{\partial \boldsymbol{z}} = \mathbf{1}$, such that it defines the final quantized latent as $\boldsymbol{z}_q = \boldsymbol{z} + sg[\hat{\boldsymbol{z}} - \boldsymbol{z}]$. The loss function that STE uses is

$$\mathrm{Loss} = \mathrm{MSE}(\boldsymbol{x}, \boldsymbol{x}_r) + \alpha \cdot \|sg\,[\boldsymbol{z}] - \boldsymbol{c}_{i^*}\|_2^2 + \beta \cdot \|\boldsymbol{z} - sg\,[\boldsymbol{c}_{i^*}]\|_2^2, \tag{2}$$

where $sg[\cdot]$ refers to the stop gradient operator, and MSE is the mean squared error. The first term is the reconstruction loss that optimizes the encoder and decoder parameters by copying the gradients intact over VQ in backpropagation. The second term is the codebook loss that optimizes the selected codewords (*i.e.*, $\boldsymbol{c}_{i^*}$), and the third term is the commitment loss that optimizes the encoder parameters to make the encoder output $\boldsymbol{z}$ close to $\boldsymbol{c}_{i^*}$.

**Exponential Moving Averages (EMA, van den Oord et al., 2017)** also copies the gradients intact through VQ via $\boldsymbol{z}_q = \boldsymbol{z} + sg[\hat{\boldsymbol{z}} - \boldsymbol{z}]$. However, EMA updates the codebook vectors as a function of moving averages of latent variables $\boldsymbol{z}$. EMA keeps the reconstruction and commitment losses, and skips the codebook loss in Eq. (2) to update the selected codewords as

$$\boldsymbol{c}_{i^*}^{(t)} := \frac{\boldsymbol{g}_{i^*}^{(t)}}{h_{i^*}^{(t)}} \text{ s.t. } h_{i^*}^{(t)} := \gamma \cdot h_{i^*}^{(t-1)} + (1-\gamma) \cdot n_{i^*}^{(t)} \quad \text{and} \quad \boldsymbol{g}_{i^*}^{(t)} := \gamma \cdot \boldsymbol{g}_{i^*}^{(t-1)} + (1-\gamma) \cdot \sum_{j=1}^{n_{i^*}^{(t)}} \boldsymbol{z}_{i^*,j}^{(t)}, \tag{3}$$

where at training iteration $t$, $\{\boldsymbol{z}_{i^*,1}, \boldsymbol{z}_{i^*,2}, \dots, \boldsymbol{z}_{i^*,n_{i^*}}\}$ is the set of $n_{i^*}$ latent variables that are closest to the selected codeword $\boldsymbol{c}_{i^*}$, and $\gamma \in (0, 1)$ is the decay rate that controls how much past observations influence the updates.

**Rotation Trick (RT, Fifty et al., 2025)** passes gradients through VQ by transforming the latent $\boldsymbol{z}$ to its closest codeword $\boldsymbol{c}_{i^*}$ with a combination of rotation and rescaling such that it defines

$$\boldsymbol{z}_q = sg[\rho \boldsymbol{R}]\boldsymbol{z} \quad \text{s.t.} \quad \rho = \frac{\|\boldsymbol{c}_{i^*}\|_2}{\|\boldsymbol{z}\|_2} \quad \text{and} \quad \boldsymbol{R} = (\boldsymbol{I} - 2\boldsymbol{r}\boldsymbol{r}^\top + 2\bar{\boldsymbol{c}}_{i^*}\bar{\boldsymbol{z}}^\top), \tag{4}$$

where $\bar{\boldsymbol{z}} = \frac{\boldsymbol{z}}{\|\boldsymbol{z}\|_2}$, $\bar{\boldsymbol{c}}_{i^*} = \frac{\boldsymbol{c}_{i^*}}{\|\boldsymbol{c}_{i^*}\|_2}$, and $\boldsymbol{r} = \frac{\bar{\boldsymbol{z}} + \bar{\boldsymbol{c}}_{i^*}}{\|\bar{\boldsymbol{z}} + \bar{\boldsymbol{c}}_{i^*}\|_2}$. RT uses the same loss as in Eq. (2), while the codebook can be updated via gradients using codebook loss or via EMA.

**Gumbel-Softmax (GS, Jang et al., 2016)** samples differentiable variables $\mathbf{y}_i$ from a continuous distribution that approximates a discrete $K$-class categorical distribution over $K$ codewords

$$\mathbf{y}_i = \mathrm{softmax}\left(\left(\log\left(\pi_i\right) + \mathbf{g}_i\right)/\tau\right) = \frac{\exp\left(\left(\log\left(\pi_i\right) + \mathbf{g}_i\right)/\tau\right)}{\sum_{j=1}^{K} \exp\left(\left(\log\left(\pi_j\right) + \mathbf{g}_j\right)/\tau\right)}, \quad i \in \{1, 2, \dots, K\}, \tag{5}$$

where $\{\pi_1, \pi_2, \dots, \pi_K\}$ are class probabilities of the categorical distribution and $\{\mathbf{g}_1, \mathbf{g}_2, \dots, \mathbf{g}_K\}$ are i.i.d. samples drawn from $\mathrm{Gumbel}(0, 1)$ distribution. The scalar $\tau$ is the softmax temperature. When $\tau$ approaches 0, the Gumbel-Softmax distribution converges to a discrete categorical distribution. To train a VQ-VAE, the loss function that Gumbel-Softmax uses is

$$\mathrm{Loss} = \mathrm{MSE}(\boldsymbol{x}, \boldsymbol{x}_r) + \varphi \cdot D_{\mathrm{KL}}\left[Q(\boldsymbol{z} \mid \boldsymbol{x}) \,\|\, P(\boldsymbol{z})\right], \tag{6}$$

where $D_{\mathrm{KL}}$ is the Kullback–Leibler divergence that pushes the encoder's posterior distribution $Q(\boldsymbol{z} \mid \boldsymbol{x})$ to be close to the uniform prior distribution of $P(\boldsymbol{z}) = \frac{1}{K}$. The $D_{\mathrm{KL}}$ term encourages all $K$ codewords to be used for quantization. As stated in Jang et al. (2016), for VQ tasks it is better to use the Straight-Through version of Gumbel-Softmax (ST-GS), in which $\mathbf{y}_i$ is discretized using $\arg\max$ in the forward pass, and gradients are computed from the continuous $\mathbf{y}_i$ in Eq. (5).

**Noise Substitution in Vector Quantization (NSVQ, Vali & Bäckström, 2022)** simulates the VQ distortion by adding noise to the latent vector to mimic the original quantization error $\boldsymbol{\xi}_Q$:

$$\boldsymbol{z}_q = \boldsymbol{z} + \boldsymbol{\xi}_Q = \boldsymbol{z} + \|\boldsymbol{z} - \hat{\boldsymbol{z}}\|_2 \cdot \frac{\mathbf{v}}{\|\mathbf{v}\|_2}; \quad \hat{\boldsymbol{z}} = \boldsymbol{c}_{i^*} = \arg\min_{\boldsymbol{c}_j} \|\boldsymbol{z} - \boldsymbol{c}_j\|_2 \text{ and } \mathbf{v} \sim \mathcal{N}(\mathbf{0}, \boldsymbol{I}). \tag{7}$$

In this way, $z_q$ is a differentiable function of the latent $z$ and the selected codeword $c_{i^*}$. As shown in Fig. 2, as NSVQ samples the noise $\mathbf{v}$ from $\mathcal{N}(\mathbf{0}, \boldsymbol{I})$, the quantized latent $z_q$ can be mapped to any random direction on the surface of a hypersphere with the radius of quantization error that equals $\|z - c_{i^*}\|_2$. Fig. 2 shows the probability that $z_q$ incurs higher quantization errors than the actual error (of $\|z - c_{i^*}\|_2$) equals $\theta_2/360°$. For any radius, $\theta_2 = 240°$. Hence, NSVQ incurs a higher quantization error than the true error with the probability of $240°/360° \approx 0.67$. In higher dimensions, this probability approaches 1, making NSVQ increasingly likely to overshoot the true nearest-codeword distortion. Furthermore, due to high randomness in the direction of $z_q$, it is challenging for the codebook to converge to its optimum location. This randomness also causes a train–test mismatch, which leads to poor reconstructions, because NSVQ applies different mappings for the same $z$ during training and testing.

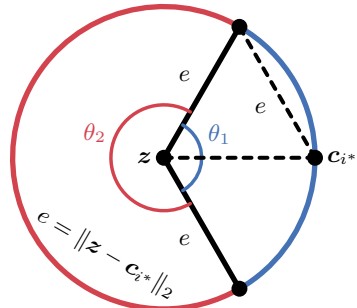

Figure 2: **Illustration of NSVQ quantization.** Input $z$ is mapped to a random point on the circle. The mapping overshoots the true quantization error with probability $\theta_2/360° \approx 0.67$, leading to a higher distortion than the nearest-codeword assignment.

**Issues with the State-of-the-Art** All of the methods mentioned above have their own drawbacks, which are highlighted in Table 1. To optimize the VQ codebook, they *(i)* change the hyperplane of optimization by adding auxiliary loss terms (Eqs. (2) and (6)) to the main objective loss (STE, EMA, RT, ST-GS), *(ii)* need additional hyperparameter (*i.e.*, $\alpha, \beta, \gamma, \varphi, \tau$) tuning (STE, EMA, RT, ST-GS), *(iii)* do not train the codebook end-to-end (STE, EMA, RT), *(iv)* cause biased gradients via mismatch between forward and backward passes (STE, ST-GS), *(v)* cause train–test mismatch as they apply different quantization for training and testing (ST-GS, NSVQ), *(vi)* suffer from *codebook collapse* (STE, EMA, RT, ST-GS, NSVQ), and *(vii)* are prone to misaligned latent and codebook representations (STE, EMA, RT, ST-GS, NSVQ) (see App. C.8).

**Other Related Work** There exist many approaches that still use STE to pass gradients through the VQ layer (Chang et al., 2022; Rombach et al., 2022; Zhu et al., 2023; Huang et al., 2023; Dong et al., 2023; Lee et al., 2022; Huh et al., 2023), each with small modifications to improve something specific. For instance, some methods use STE while resolving *codebook collapse* with different codebook replacement and reinitialization strategies (Kolesnikov et al., 2022; Łańcucki et al., 2020; Zheng & Vedaldi, 2023; Dhariwal et al., 2020; Zeghidour et al., 2021), or by using different distance metrics than Euclidean (Yu et al., 2021; Goswami et al., 2024). Others use stochastic sampling (Maddison et al., 2017; Takida et al., 2022; Chen et al., 2024) while increasing codebook utilization by additional loss terms (Zhang et al., 2023; Yu et al., 2023), or soft quantization to backpropagate gradients through the VQ layer (Gautam et al., 2023; Agustsson et al., 2017). Additionally, some recent methods use NSVQ (Gómez et al., 2023; Walsh et al., 2024; Lee et al., 2024; Wang et al., 2025; Ye et al., 2025; Zhu et al., 2025), and RT (Kim et al., 2025; Xue et al., 2025; Bae & Shin, 2025) techniques. In rather unique lines of work, Finite Scalar Quantization (FSQ, Mentzer et al., 2023), Random Projection Quantizer (RPQ, Chiu et al., 2022), Binary Spherical Quantization (BSQ, Zhao et al., 2024), and Lookup-Free Quantization (LFQ, Yu et al., 2023) constrain the codebook to follow a predefined geometric structure and do not train the codebook together with the model.

## 3 METHODS

We address the problem of making vector quantization differentiable while preserving hard assignments in the forward pass. Our goal is to construct a differentiable surrogate $z_q(z, \mathcal{C})$ that *(i)* coincides with the hard nearest-neighbor assignment in the small-variance limit, *(ii)* yields stable and geometrically faithful gradients for both encoder and codebook, and *(iii)* avoids auxiliary losses, hyperparameter tuning, or train–test mismatches. We first introduce *DiVeQ* (Sec. 3.1), which reparameterizes the quantization error to align its direction with the nearest codeword. We then extend this construction to *SF-DiVeQ* (Sec. 3.2), which quantizes along line segments between neighboring codewords to reduce error and promote codebook utilization.

### 3.1 DIVEQ: DIFFERENTIABLE VQ VIA DIRECTIONAL REPARAMETERIZATION

DiVeQ models quantization as adding a simulated error vector whose *magnitude* equals the input–

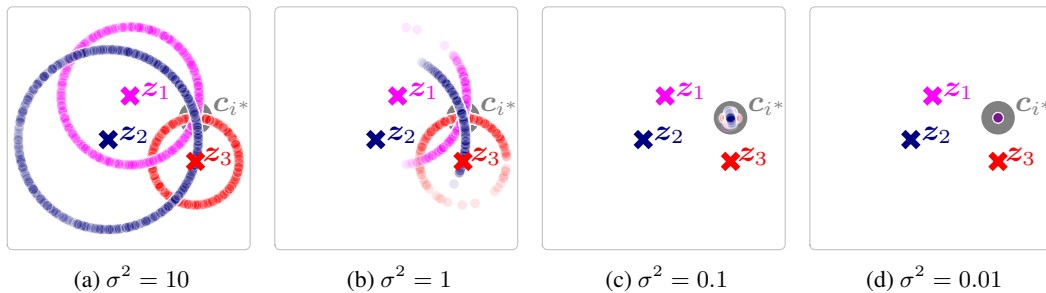

Figure 3: Impact of $\sigma^2$ in DiVeQ quantization accuracy. Each panel shows mappings of input $\boldsymbol{z}_i$ to its closest codeword $\boldsymbol{c}_{i^*}$ using our proposed DiVeQ (Eq. (8)) when sampling 1000 random directional vectors $\mathbf{v}_d$ from $\mathcal{N}(\mathbf{0}, \sigma^2\boldsymbol{I})$. DiVeQ quantization accuracy increases when $\sigma^2 \to 0$ (see App. C.5).

codeword distance and whose *direction* is aligned with the nearest codeword. Similar to NSVQ (see Sec. 2), to obtain a differentiable quantized input $\boldsymbol{z}_q$, DiVeQ approximates VQ as addition of the quantization error $\boldsymbol{\xi}_Q$ to the input $\boldsymbol{z}$, *i.e.*, $\boldsymbol{z}_q = \boldsymbol{z} + \boldsymbol{\xi}_Q$. However, in contrast to NSVQ, DiVeQ models $\boldsymbol{\xi}_Q$ such that $\boldsymbol{z}_q$ points to the closest codeword $\boldsymbol{c}_{i^*}$ precisely and thus, $\boldsymbol{z}_q$ results in the original quantization error of $\|\boldsymbol{z} - \boldsymbol{c}_{i^*}\|_2$ (see Fig. 3). To this end, by getting intuition from the reparameterization trick (Kingma & Welling, 2013), we define a directional noise $\mathbf{v}_d = \mathbf{v} + \vec{\boldsymbol{d}}$ such that $\vec{\boldsymbol{d}} = \boldsymbol{c}_{i^*} - \boldsymbol{z}$ and $\mathbf{v} \sim \mathcal{N}(\mathbf{0}, \sigma^2\boldsymbol{I})$. Then,

$$\boldsymbol{z}_q = \boldsymbol{z} + \boldsymbol{\xi}_Q = \boldsymbol{z} + \|\vec{\boldsymbol{d}}\|_2 \cdot sg\left[\frac{\mathbf{v}_d}{\|\mathbf{v}_d\|_2}\right] = \boldsymbol{z} + \|\boldsymbol{c}_{i^*} - \boldsymbol{z}\|_2 \cdot sg\left[\frac{\mathbf{v}_d}{\|\mathbf{v}_d\|_2}\right]; \quad \boldsymbol{c}_{i^*} = \arg\min_{\boldsymbol{c}_j}\|\boldsymbol{z} - \boldsymbol{c}_j\|_2, \tag{8}$$

where $sg[\cdot]$ is the stop gradient operator. Here, by changing the variance $\sigma^2$ from $\infty$ to zero (Fig. 3a to Fig. 3d), the quantization accuracy will increase, as $\boldsymbol{z}_q$ starts from pointing to a hypersphere (as in NSVQ) and approaches pointing exactly to the selected codeword $\boldsymbol{c}_{i^*}$. The limits give:

$$\lim_{\sigma^2 \to \infty} \mathbf{v}_d = \lim_{\sigma^2 \to \infty}\left[\mathcal{N}(\mathbf{0}, \sigma^2\boldsymbol{I}) + \vec{\boldsymbol{d}}\right] = \mathcal{N}(\boldsymbol{c}_{i^*} - \boldsymbol{z}, \infty) \Rightarrow \boldsymbol{z}_q = \boldsymbol{z} + \|\boldsymbol{c}_{i^*} - \boldsymbol{z}\|_2 \cdot sg\left[\mathbf{s}\right],$$

$$\lim_{\sigma^2 \to 0} \mathbf{v}_d = \lim_{\sigma^2 \to 0}\left[\mathcal{N}(\mathbf{0}, \sigma^2\boldsymbol{I}) + \vec{\boldsymbol{d}}\right] \approx \vec{\boldsymbol{d}} = \boldsymbol{c}_{i^*} - \boldsymbol{z} \Rightarrow \boldsymbol{z}_q = \boldsymbol{z} + \|\boldsymbol{c}_{i^*} - \boldsymbol{z}\|_2 \cdot sg\left[\frac{\boldsymbol{c}_{i^*} - \boldsymbol{z}}{\|\boldsymbol{c}_{i^*} - \boldsymbol{z}\|_2}\right] = \boldsymbol{c}_{i^*}, \tag{9}$$

where $\mathbf{s}$ is a random variable almost uniformly distributed on the surface of the $D$-dim unit hypersphere of $\mathbb{S} = \{\forall \mathbf{s} \in \mathbb{R}^D : \|\mathbf{s}\|_2 = 1\}$. When using Eq. (8) during training, $\boldsymbol{z}_q$ is a differentiable function of $\boldsymbol{z}$ and $\boldsymbol{c}_{i^*}$, and thus it can be used in end-to-end training of VQ together with other trainable modules in a neural network. Hence, the gradients can be calculated as

$$\frac{\partial \boldsymbol{z}_q}{\partial \boldsymbol{z}} = \mathbf{1} + \boldsymbol{a} \cdot \frac{\boldsymbol{z} - \boldsymbol{c}_{i^*}}{\|\boldsymbol{c}_{i^*} - \boldsymbol{z}\|_2} \quad \text{and} \quad \frac{\partial \boldsymbol{z}_q}{\partial \boldsymbol{c}_{i^*}} = \boldsymbol{a} \cdot \frac{\boldsymbol{c}_{i^*} - \boldsymbol{z}}{\|\boldsymbol{c}_{i^*} - \boldsymbol{z}\|_2} \quad \text{s.t.} \quad \boldsymbol{a} = sg\left[\frac{\boldsymbol{c}_{i^*} - \boldsymbol{z}}{\|\boldsymbol{c}_{i^*} - \boldsymbol{z}\|_2}\right]. \tag{10}$$

## 3.2 SF-DiVeQ: space-filling differentiable vector quantization

Space-Filling Vector Quantization (SFVQ, Vali & Bäckström, 2023a) is a modification of VQ that maps an input to a piece-wise continuous curve (see Fig. 1 in Vali & Bäckström, 2023a). Contrary to VQ that quantizes the input exclusively on the codewords, SFVQ is a curve that gets turned and twisted inside a $D$-dim distribution and quantizes the input on a continuous curve whose corner points are the codewords of the SFVQ base codebook $\mathcal{C}$. SFVQ adopts a dithering trick to generate the curve. For each training iteration, by having an input $\boldsymbol{z} \in \mathbb{R}^D$ and a base codebook $\mathcal{C}$ with $K$ codewords, SFVQ generates a dithered codebook $\mathcal{C}^d = \{\boldsymbol{c}_1^d, \ldots, \boldsymbol{c}_{K-1}^d\}$ with $K - 1$ codewords by sampling at random places on the line connecting two subsequent codewords of $\boldsymbol{c}_j$ and $\boldsymbol{c}_{j+1}$. The input $\boldsymbol{z}$ is then quantized to the closest codeword from the dithered codebook $\mathcal{C}^d$ as

$$\hat{\boldsymbol{z}} = \arg\min_{\boldsymbol{c}_j^d}\|\boldsymbol{z} - \boldsymbol{c}_j^d\|_2 = \boldsymbol{c}_{i^*}^d = (1 - \lambda_{i^*})\boldsymbol{c}_{i^*} + \lambda_{i^*}\boldsymbol{c}_{i^*+1}, \quad j, i^* \in \{1, \ldots, K-1\}, \tag{11}$$

where $\boldsymbol{c}_{i^*}^d$ is the closest dithered codeword that is generated by the interpolation of two subsequent codewords of $\boldsymbol{c}_{i^*}$ and $\boldsymbol{c}_{i^*+1}$ from the base codebook $\mathcal{C}$. $\lambda_j$ is the interpolation factor for the codewords $\boldsymbol{c}_j$ and $\boldsymbol{c}_{j+1}$ that is sampled from uniform distribution of $U(0, 1)$. Generating new dithered

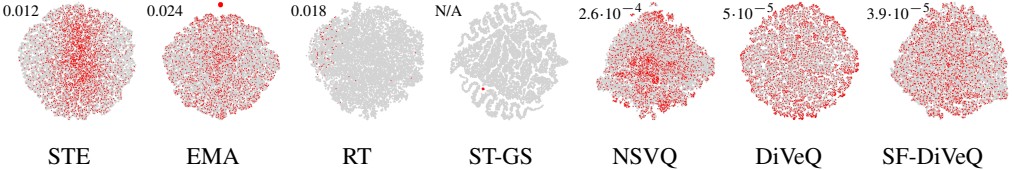

Figure 4: **Codebook misalignment:** t-SNE plots of the learned codebook $\mathcal{C}_z$ (*red crosses*) and latent $\mathcal{P}_z$ (*gray points*) representations for different VQ methods in VQ-VAE compression. The figure shows the misalignment between $\mathcal{C}_z$ and $\mathcal{P}_z$ (discussed in Sec. 4) for different methods. The plots refer to the cases highlighted in Fig. 26. The numbers report distortion per bit↓ (see App. C.8).

codebooks by random interpolations (using $\lambda_j$) on the line connecting $\boldsymbol{c}_j$ and $\boldsymbol{c}_{j+1}$ over and over for different training batches establishes a continuous topology between subsequent codewords of $\mathcal{C}$ and locates SFVQ codewords such that the lines connecting subsequent codewords to be inside the distribution space. The reason is that the points on the SFVQ curve are representatives of dithered codewords $\boldsymbol{c}_j^d$ that should be valid quantization points for the input data.

**Proposed SF-DiVeQ** When training VQ, if a codeword is not selected for quantization, it will not receive any gradients and, as a result, it will not get updated. This problem is known as *codebook collapse* (Vali, 2025; Mentzer et al., 2023), in which a subset of codebook vectors remains inactive and will not get updated during training. To resolve this issue, NSVQ proposed a codebook replacement procedure such that after a specific number of training batches, inactive codewords will be replaced by a perturbation of the active ones. To maximize the codebook vectors' usage during training, it is better and safer always to apply codebook replacement regardless of the VQ optimization technique (*e.g.*, STE, EMA, RT, ST-GS, NSVQ, or DiVeQ). According to Huh et al. (2023), codebook replacement techniques work well and do not degrade the model performance. That is why we adopt codebook replacement in this paper. However, codebook replacement is a heuristic method that adds to the complications of VQ training. By leveraging intuition from SFVQ, we propose SF-DiVeQ, a differentiable VQ technique that eliminates the need for codebook replacement. SF-DiVeQ quantizes the input $\boldsymbol{z}$ to a random location on the line connecting two subsequent codewords as

$$\boldsymbol{z}_q = \boldsymbol{z} + \|\boldsymbol{c}_{i^*} - \boldsymbol{z}\|_2 \cdot sg\left[\frac{(1 - \lambda_{i^*})\mathbf{v}_{d_{i^*}}}{\|\mathbf{v}_{d_{i^*}}\|_2}\right] + \|\boldsymbol{c}_{i^*+1} - \boldsymbol{z}\|_2 \cdot sg\left[\frac{\lambda_{i^*}\mathbf{v}_{d_{i^*+1}}}{\|\mathbf{v}_{d_{i^*+1}}\|_2}\right] \; ; \; i^* \in \{1, \dots, K-1\},$$
(12)

where $\boldsymbol{c}_{i^*}$ and $\boldsymbol{c}_{i^*+1}$ are the two codewords whose interpolation is the closest quantization point to the input $\boldsymbol{z}$, and $\mathbf{v}_{d_{i^*}} = \mathbf{v} + (\boldsymbol{c}_{i^*} - \boldsymbol{z})$ and $\mathbf{v}_{d_{i^*+1}} = \mathbf{v} + (\boldsymbol{c}_{i^*+1} - \boldsymbol{z})$ such that $\mathbf{v} \sim \mathcal{N}(\mathbf{0}, \sigma^2 \boldsymbol{I})$. $\lambda_{i^*}$ is the interpolation factor that is sampled from the uniform distribution of $U(0,1)$ for each training batch. As discussed earlier, since during training SF-DiVeQ quantizes $\boldsymbol{z}$ on the lines connecting subsequent codewords, these lines will be pulled inside the distribution space as they should be valid quantization points. As a result of this property, SF-DiVeQ prevents misalignment of codebook and latent representations (see Fig. 4), and it does not need any heuristic codebook replacement during training. Furthermore, as SF-DiVeQ quantizes $\boldsymbol{z}$ on a curve (rather than exclusively on the codewords), it has many more degrees of freedom for quantization than ordinary VQ methods. Therefore, it potentially results in smaller quantization errors than the other methods. Apart from DiVeQ and SF-DiVeQ, we also propose two variants of them, which are discussed in App. B.2.

## 4 EXPERIMENTS

We compare the performance of our proposed DiVeQ and SF-DiVeQ with other approaches of STE, EMA, RT, ST-GS, and NSVQ in three different applications of image compression, image generation, and speech coding. For image compression and speech coding tasks, we use the VQ-VAE of van den Oord et al. (2017) and the DAC model in Kumar et al. (2023), respectively, with minor modifications to have a basic compression model that can clearly reflect the performance difference between various VQ methods, and for image generation, we use VQGAN (Esser et al., 2021). Experimental setup and relevant results are provided in the following. For more details on the implementation, model architectures, and hyperparameters, see App. A.

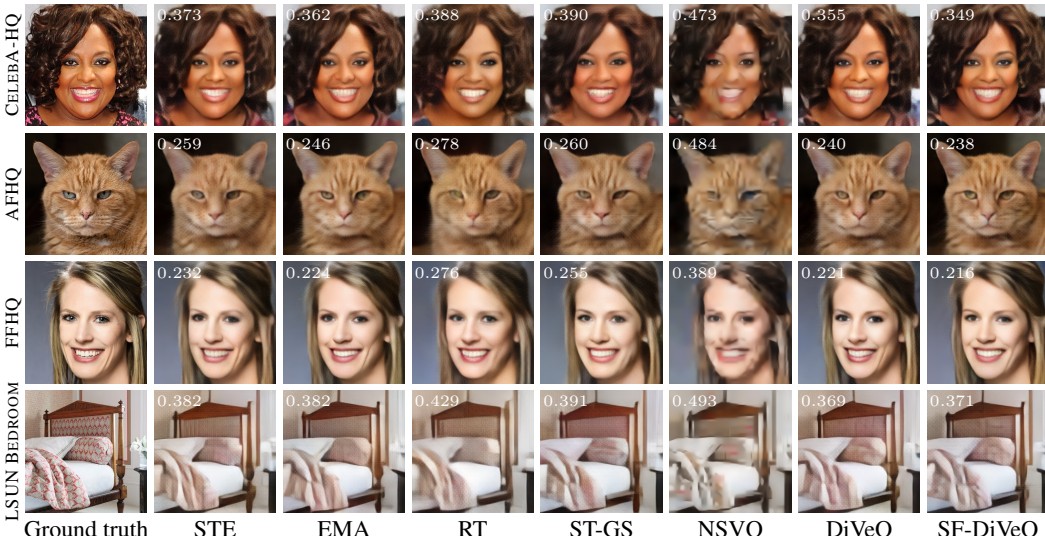

Figure 5: **DiVeQ and SF-DiVeQ improve image reconstruction.** Qualitative comparison of reconstructed images in VQ-VAE compression task for different VQ optimization methods with an 11-bit codebook (*i.e.*, codebook size of $2^{11} = 2048$). We report LPIPS↓ values in the left-hand corners.

Following the sensitivity experiments in App. C.5, we do not consider the variance $\sigma^2$ an actual hyperparameter that needs tuning as long as it is small ($\sigma^2 \leq 10^{-2}$). Different values for the variance, when $\sigma^2 \leq 10^{-2}$, only lead to marginal change in the performance of both DiVeQ and SF-DiVeQ in both VQ-VAE image compression and VQGAN image generation tasks (see Fig. 20, Fig. 21, and Table 10). For the results in the main paper, we use $\sigma^2 = 10^{-3}$ (for image compression and speech coding) and $\sigma^2 = 10^{-2}$ (for image generation), with more results in App. C.5. Note that the values are not tuned on a per-data set basis.

**Loss Functions** In VQ-VAE compression, to enhance the quality of reconstructions for $256 \times 256$ images, in addition to the MSE reconstruction loss in Eqs. (2) and (6), we add LPIPS (VGG-16) as the perceptual loss with a weighting coefficient of one. For VQGAN (Esser et al., 2021) and DAC (Kumar et al., 2023) models, we use the same loss functions as those used in the original models.

**Evaluation Metrics** During inference, the trained models with different VQ optimization techniques are used to reconstruct the test set images and speech samples for VQ-VAE compression and DAC-based codec, respectively, and to generate new images for VQGAN. For inference of all VQ methods, the input $z$ is mapped to the closest codeword $c_{i*}$ with hard VQ using $\arg\min$, except SF-DiVeQ, in which the input is mapped to the nearest point on the space-filling curve (see Sec. 2 in Vali & Bäckström (2023a)). Structural Similarity Index Measure (SSIM), Peak Signal to Noise Ratio (PSNR) (both from *scikit-image* library), and Learned Perceptual Image Patch Similarity (LPIPS, Zhang et al., 2018) are used to evaluate the quality of the reconstructions for image compression, and Fréchet Inception Distance (FID) score (Parmar et al., 2022) is used to assess the quality of VQGAN generations. Log Spectral Distance (LSD), Mel-Frequency Cepstral Coefficients (MFCC) distance, Perceptual Evaluation of Speech Quality (PESQ), and Short-Time Objective Intelligibility (STOI) are used to evaluate the decompressed speech quality.

**Data Sets** In both image compression and generation tasks, we conduct the experiments over AFHQ (Choi et al., 2020), CELEBA-HQ (Karras et al., 2018), FFHQ (Karras et al., 2019), LSUN Bedroom, and LSUN Church (Yu et al., 2016) data sets with resolution of $256 \times 256$. The data sets contain 15803, 30k, 70k, 70k, 70k images, respectively. For image compression, all data sets are divided into 80/20% train–test splits. For image generation, we use the train sets for training except for AFHQ and CELEBA-HQ, in which we use the full data sets, as we need more data for the task. To compute the FID score, we generate the same number of images used for training. In speech coding, we use the CSTR VCTK data set (Yamagishi et al., 2019) divided into 80/20% train–test split with no overlapping speakers and speech files between the train and test sets (more details in App. A.3).

**Codebook Replacement and Initialization** In the NSVQ paper (Vali & Bäckström, 2022), the NSVQ method is compared with STE and EMA, such that the codebook replacement is only used

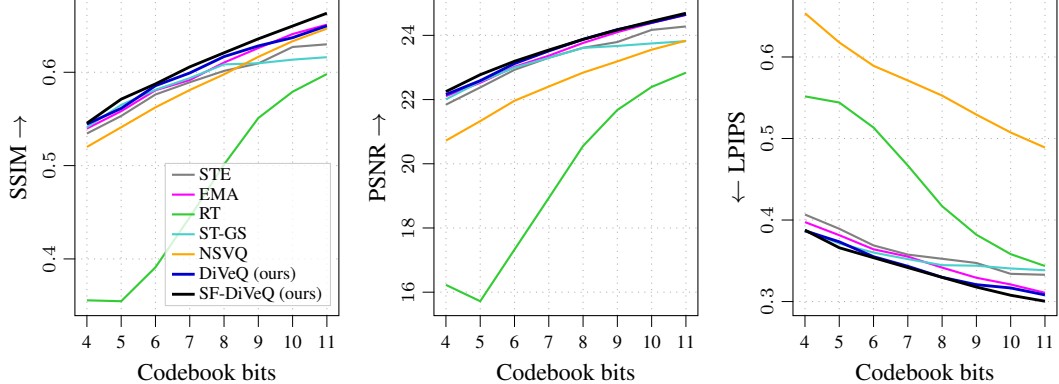

Figure 6: **Consistent improvement in image reconstruction.** Quantitative comparison of reconstructed images from AFHQ test set for different VQ optimization methods over different codebook sizes. Each curve is the average of the metric over all test set reconstructions and over three different individual runs. Results for FFHQ, CELEBA-HQ, and LSUN are in Figs. 12 to 15 in App. C.1.

for NSVQ. To have a fair comparison of different VQ training methods, our proposed codebook replacement (App. B.1) is used for all methods (except SF-DiVeQ, which does not require it). To evaluate how much codebook replacement contributes to the overall performance, in App. C.6, we skipped codebook replacement in the VQ-VAE compression task, where the results show that our proposed DiVeQ consistently outperforms other VQ optimization methods. Moreover, in all experiments, codebook initialization is the same for all methods except for SF-DiVeQ, which uses a *custom* initialization, in which the models are trained without VQ for two epochs, then SF-DiVeQ starts, such that the codebook is initialized by the mean of latent vectors of the last 20–50 training batches. In App. C.7, we show that the performance of SF-DiVeQ with *random* initialization (*i.e.*, in which the codebook is randomly initialized by the latent vectors of the first training batch) remains almost the same as the *custom* initialization in the VQ-VAE compression task.

**VQ-VAE Compression Results** After training VQ-VAE with different VQ training methods, the trained codebook is used to reconstruct the test set images. Fig. 5 shows ground truth images from different data sets and their corresponding reconstructions using learned codebooks from different VQ techniques. Apart from qualitative comparison, all test set images (from different data sets) are reconstructed, and then the SSIM, PSNR, and LPIPS metrics are computed for all of these reconstructions. Fig. 6 shows the quantitative comparison of all VQ methods over different codebook sizes for the AFHQ data set. The reported value for each metric is the average of that metric over all test set reconstructions and over three different individual runs. In the figure, our DiVeQ and SF-DiVeQ outperform other VQ optimization methods for different codebook sizes and for all three metrics. Our proposed methods yield higher SSIM and PSNR values, and lower LPIPS with a big margin to RT and NSVQ, and with a smaller margin to STE, EMA, and ST-GS. In addition, the performance gap of our methods to STE and ST-GS grows with the increase in codebook size, while this gap decreases compared to the RT method. The last takeaway is that the RT performs poorly for small codebook sizes. App. C.1 provides similar quantitative comparisons for other data sets.

**VQGAN Generation Results** After training the VQGAN's generator and transformer models for each data set, we compute the FID between the original train set and generated samples. Table 2 presents the obtained FID values for different VQ optimization methods for CELEBA-HQ data set. Table 7 in App. C.2 provides similar FID comparisons for other data sets. We trained the generator (*i.e.*, VQ-VAE) with two different hyperparameter settings of HP$_1$ : ($lr = 2.5 \cdot 10^{-5}$, batch size $= 8$) and HP$_2$ : ($lr = 2.5 \cdot 10^{-4}$, batch size $= 32$).

Table 2: **DiVeQ & SF-DiVeQ robustify training for challenging small codebooks, and they are less sensitive to hyperparameter settings.** FID↓ scores for VQ optimization methods on CELEBA-HQ. Values *in red* refer to the misalignment cases as in Fig. 4.

| CELEBA-HQ data set | $lr = 2.5 \cdot 10^{-5}$ batch size $= 8$ | | | | $lr = 2.5 \cdot 10^{-4}$ batch size $= 32$ | | | |
|---|---|---|---|---|---|---|---|---|
| Approach \ bits | 8 | 9 | 10 | 12 | 8 | 9 | 10 | 12 |
| STE | 6.64 | 5.57 | **5.28** | 6.69 | 334 | 7.54 | 7.34 | 9.45 |
| EMA | 6.86 | 6.30 | 6.32 | 6.24 | **8.42** | 7.42 | 6.97 | 9.41 |
| RT | 9.32 | 7.55 | 6.40 | **5.44** | 12.3 | 9.33 | **6.53** | **6.58** |
| ST-GS | 8.47 | 6.81 | 5.48 | 5.47 | 309 | 41.1 | 197 | 155 |
| NSVQ | 81.5 | 70.4 | 59.2 | 48.9 | 78.4 | 70.1 | 62.1 | 49.6 |
| DiVeQ (ours) | **5.90** | 6.69 | 6.32 | 7.69 | 8.44 | 8.01 | 7.59 | 9.54 |
| SF-DiVeQ (ours) | 6.24 | **5.21** | 5.57 | 6.00 | 8.46 | **6.66** | 7.02 | 7.4 |

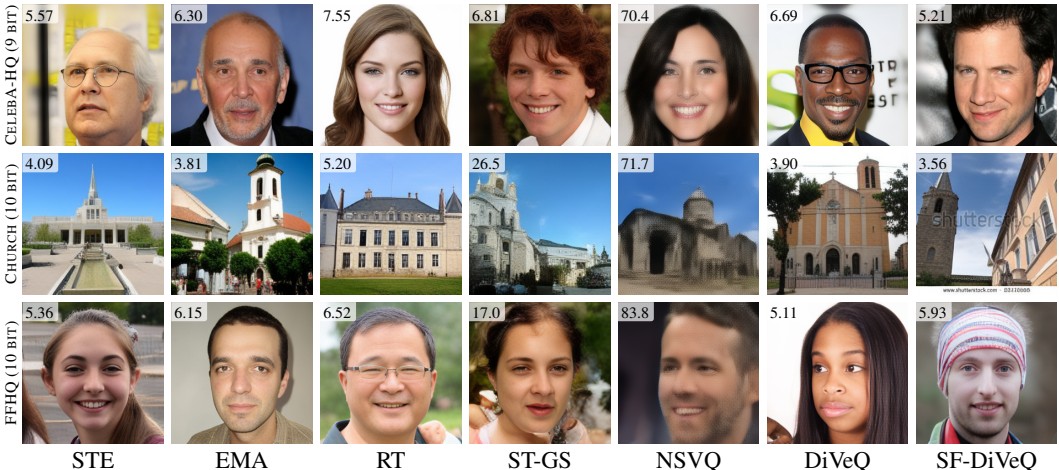

Figure 7: **Generation task.** Qualitative comparison of randomly generated images in the VQGAN generation task for different VQ optimization methods. We report FID↓ values in the left-hand corners. More generations are provided in Figs. 30 to 34 in App. C.11.

The $HP_1$ uses a small batch size and learning rate to ensure convergence for all VQ methods while it is less likely to highlight the robustness of our approaches to the use of different hyperparameter settings. Using a larger batch size and a higher learning rate (*i.e.*, $HP_2$) might cause misalignment between codebook and latent representations, where generations result in high FID values. The cases where misalignment happens are highlighted *in red* in Table 2 and Table 7 (*i.e.*, cases where FID > 30). According to the tables, our proposed DiVeQ is less prone to misalignment compared to other VQ methods, whereas no misalignment happens for our proposed SF-DiVeQ.

The evaluations reveal that RT, ST-GS, and NSVQ reduce FID with the increase in codebook size, and (similar to STE and EMA) our methods maintain the generation quality over different codebook sizes such that they achieve lower FIDs than the other methods for small codebooks (*i.e.*, 8- and 9-bit codebooks) where the generation task becomes more challenging. Furthermore, by pairing up the FIDs for the RT method with the results in the VQ-VAE compression task, we can claim that RT performs poorly at codebook learning for small codebooks in both compression and generation tasks. Note that the FID metric reflects distributional similarity, not necessarily the reconstruction quality. Hence, the FID value should not necessarily improve with the increase in codebook size $K$, because *(i)* the transformer might struggle with large $K$ where the entropy of token indices is high, and *(ii)* sampling is inherently a random process that might end up in different FID values when sampling several times from a fixed pretrained transformer. Fig. 7 shows a qualitative comparison of VQGAN generations for different VQ methods (see App. C.11 for more qualitative comparisons).

**Misalignment of Codebook and Latent Representations**   When training a neural network (*e.g.*, VQ-VAE), the latent representation $\mathcal{P}_z$ of the network layers is constantly shifting. For high learning rates and large batch sizes, the update shift in $\mathcal{P}_z$ can be large, causing a misalignment between $\mathcal{P}_z$ and codebook representation $\mathcal{C}_z$. Misalignment means that $\mathcal{C}_z$ does not fit well with $\mathcal{P}_z$, and this phenomenon is also called *internal codebook covariate shift* (Huh et al., 2023).

According to Shannon's rate-distortion theory (Shannon, 1959), the rate-distortion function $D(R)$ is a continuous, strictly decreasing, convex function. Therefore, in a lossy compression task, if the codebook size grows, then the distortion $D$ must decrease, and as a result, the quantitative metrics improve (assuming the metrics fully reflect the amount of distortion). After plotting the quantitative results of trained VQ-VAEs with different learning rates and batch sizes (see the ablation studies on batch size and learning rate in Apps. C.3 and C.4), we spot that the rate-distortion theory does not hold in some experiments, such that the increase in codebook size does not result in improvement of quantitative metrics. Hence, we hypothesize that the reason could be the misalignment of $\mathcal{P}_z$ and $\mathcal{C}_z$. In Fig. 26, we highlighted those cases that are potentially prone to misalignment with *red circles*. For illustration purposes, we provide t-SNE plots of these cases in Fig. 4, which confirms our hypothesis that misalignment is the main culprit. It is important to note that by observing the quantitative results (*e.g.*, Fig. 26), we spot the misalignment cases where the rate-distortion theory does not hold, or when the quantitative metrics do not improve by the increase in codebook

Table 3: **Consistent improvement in speech decompression.** Quantitative comparison of decompressed speech samples from the VCTK test set for different VQ optimization methods over different codebook sizes when batch size $= 64$. Each value shows the average of the metric over all test set decompressed samples. Results for other batch sizes are provided in Tables 8 and 9.

| Approach \ bits | Log spectral distance↓ | | | | MFCC distance↓ | | | | PESQ↑ | | | | STOI↑ | | | |
|---|---|---|---|---|---|---|---|---|---|---|---|---|---|---|---|---|
| | 10 | 11 | 12 | 13 | 10 | 11 | 12 | 13 | 10 | 11 | 12 | 13 | 10 | 11 | 12 | 13 |
| STE | 1.11 | 3.20 | 1.14 | 1.11 | 93.3 | 344 | 105 | 96.0 | 1.22 | 1.04 | 1.14 | 1.22 | 0.75 | 0.40 | 0.71 | 0.75 |
| EMA | 3.40 | 1.03 | 1.03 | 1.02 | 317 | 72.6 | 72.4 | 69.1 | 1.03 | 1.55 | 1.59 | 1.67 | 0.39 | 0.83 | 0.84 | 0.84 |
| RT | 1.09 | 1.05 | 1.04 | 1.05 | 92.6 | 84.0 | 82.8 | 80.8 | 1.27 | 1.35 | 1.43 | 1.41 | 0.76 | 0.80 | 0.80 | 0.81 |
| ST-GS | 1.13 | 3.32 | 3.45 | 1.13 | 97.9 | 349 | 333 | 97.9 | 1.19 | 1.03 | 1.04 | 1.21 | 0.76 | 0.41 | 0.39 | 0.76 |
| NSVQ | 1.11 | 1.09 | 1.10 | 1.07 | 108 | 103 | 101 | 93.6 | 1.35 | 1.43 | 1.49 | 1.56 | 0.79 | 0.81 | 0.82 | 0.83 |
| DiVeQ (ours) | **1.04** | 1.04 | 1.04 | 1.02 | 77.2 | 75.5 | 73.9 | 72.6 | 1.41 | **1.55** | 1.53 | 1.64 | 0.82 | 0.83 | 0.83 | **0.85** |
| SF-DiVeQ (ours) | 1.05 | **1.02** | **1.01** | **1.01** | **74.6** | **71.8** | **68.3** | **66.8** | **1.49** | 1.52 | **1.62** | **1.75** | **0.83** | **0.84** | **0.85** | **0.85** |

size. Therefore, the t-SNE plots should only be interpreted as an intuitive visualization of codebook homogeneity with respect to the latents within each method separately. Furthermore, we consider a case as misalignment even if all of the codewords are inside the latent representation $\mathcal{P}_z$ (like STE in Fig. 4), but they are not properly scattered within $\mathcal{P}_z$.

Note that codebook replacement (App. B.1) can help reduce the risk of misalignment occurring, but it does not completely avoid it. Fig. 4 shows the learned $\mathcal{P}_z$ and $\mathcal{C}_z$ representations for the cases where misalignment happens (*i.e.*, the cases highlighted in Fig. 26, where rate-distortion theory does not hold). Different types of misalignment occur for all methods (except for SF-DiVeQ), even if they use codebook replacement. For STE, RT, and NSVQ, the codewords are located inside $\mathcal{P}_z$, but they are not homogeneously scattered. In ST-GS, $\mathcal{C}_z$ is completely out of $\mathcal{P}_z$, whereas in EMA, a large portion of codewords are outside $\mathcal{P}_z$. A much milder misalignment happens for our proposed DiVeQ, as a small portion of codewords are scattered densely within $\mathcal{P}_z$, particularly in the corners. However, SF-DiVeQ scatters the codewords homogeneously within $\mathcal{P}_z$. In our experiments, we do not notice any sign of misalignment for SF-DiVeQ. We hypothesize the reason is SF-DiVeQ's training strategy, which pulls codewords inside $\mathcal{P}_z$ without requiring heuristic codebook replacement.

**Speech Coding Results** As a final experiment, we evaluate speech decompression on the VCTK data set. We train a DAC-based speech coding model (*cf.* Kumar et al., 2023) with different VQ methods and compute the LSD, MFCC distance, PESQ, and STOI metrics between the original and decompressed speech. Table 3 shows the average of each metric over all VCTK test samples for different codebook sizes when batch size $= 64$, and Tables 8 and 9 in App. C.3 show the same comparisons for batch size $\in \{32, 16\}$. In Tables 3, 8, 9, entries highlighted *in red* correspond to runs where codebook–latent misalignment occurs, and the decoded speech is severely degraded and unintelligible. The tables show that our proposed DiVeQ and SF-DiVeQ consistently yield higher decompressed-speech quality than alternative VQ optimization methods. Moreover, in Tables 3, 8, 9, STE, EMA, and ST-GS fail to converge in some configurations and exhibit misalignment, whereas the remaining methods avoid this issue. For this experiment, we intentionally use a simplified DAC-based codec with a single codebook and smaller encoder and decoder networks than in the original setup, in order to isolate the effect of the VQ method (see App. A.3). Decompressed speech samples are provided in the supplementary material for subjective comparison.

## 5  DISCUSSION AND CONCLUSION

In this paper, we introduced DiVeQ and SF-DiVeQ, two new differentiable vector quantization methods that can be trained end-to-end in a neural network. We provided a comprehensive overview of the limitations of existing approaches (Sec. 2) that motivated our approach. In Sec. 3.1, we showed how DiVeQ uses a directional reparameterization of the quantization error to preserve hard assignments in the forward pass while allowing gradients to flow. We further extended this idea to SF-DiVeQ (Sec. 3.2), which quantizes along codeword connections (*i.e.*, space-filling) to reduce error and ensure full codebook utilization. In Sec. 4, we demonstrated that both methods outperform existing quantization strategies in VQ-VAE, VQGAN, and DAC models across several benchmarks. Also, they are advantageously applicable to other VQ variants like Residual VQ (experiments in App. C.9).

Importantly, DiVeQ and SF-DiVeQ act as **drop-in replacements** for standard VQ layers and require no auxiliary losses, temperature schedules, or special heuristics. We advocate their use as choices for differentiable quantization in deep generative models.

ACKNOWLEDGMENTS

This work was supported by the Research Council of Finland Flagship programme: Finnish Center for Artificial Intelligence FCAI. We acknowledge funding from the Research Council of Finland (grants 339730 and 362408). We acknowledge the computational resources provided by the Aalto Science-IT project and CSC–IT Center for Science, Finland. We also gratefully acknowledge the support from Esteban Gómez in conducting the DAC-based speech coding experiments.

REPRODUCIBILITY STATEMENT

In App. A, we provide comprehensive implementation details of our VQ-VAE image compression, VQGAN image generation, and DAC-based speech coding models. The section includes model architectures, hyperparameters, and the specifications of how we implement other VQ optimization methods. For image compression and generation tasks, all the experiments are done using the well-known benchmark data sets of AFHQ, CELEBA-HQ, FFHQ, LSUN Bedroom, and LSUN Church obtained from https://www.kaggle.com/datasets.

We provide a code implementation to train both VQ-VAE image compression and VQGAN image generation models (including generator, discriminator, and transformer) together with a README.md file explaining the structure of the contents in the supplementary material. In addition, in relevance to our DAC-based speech coding experiment, we add some speech samples to the supplementary materials for subjective comparisons. Implementations of all VQ optimization methods are also included in the supplementary materials. Our reference implementation is available at https://github.com/AaltoML/DiVeQ. DiVeQ is also available as a PyPI package at https://pypi.org/project/diveq/.

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

## APPENDICES

Here, we provide an overview of the contents covered in the Appendices. In App. A, we provide a comprehensive and detailed explanation of the implementations for training the models in VQ-VAE image compression, VQGAN image generation, and DAC-based speech coding tasks. In App. B, we present other proposals of the paper including *(i)* new proposed codebook replacement (App. B.1), and *(ii)* new variants of DiVeQ and SF-DiVeQ techniques (App. B.2). App. C provides additional qualitative and quantitative evaluations of all VQ optimization methods to substantiate the superior performance of our proposed DiVeQ and SF-DiVeQ techniques in the established VQ-VAE image compression, VQGAN image generation, and DAC speech coding tasks.

## LIST OF APPENDICES

## A  IMPLEMENTATION DETAILS

In this paper, we establish the VQ-VAE image compression, VQGAN image generation, and DAC speech coding tasks to compare the performance of different VQ optimization methods within these three frameworks. This section provides the implementation details of these three tasks. There are some hyperparameters that are shared for all three of these tasks, which are mentioned in the following.

**Mutual hyperparameters for different tasks** In all VQ-VAE compression, VQGAN generation, and DAC speech coding tasks, we set the loss coefficients in Eqs. (2) and (6) as $\alpha = \varphi = 1$, and $\beta = 0.25$. The EMA decay factor is fixed at $\gamma = 0.99$ for all experiments in the paper. The ST-GS temperature $\tau$ is exponentially annealed from $\tau_{\text{start}} = 1$ to $\tau_{\text{min}} = 0.1$ over the training epochs (for more details see App. A.4).

## A.1 VQ-VAE COMPRESSION

**Hyperparameters** In VQ-VAE compression task (for Fig. 6 in Sec. 4 and Figs. 12 to 15 in App. C.1), we train all VQ methods for 100 epochs with the batch size of 32 using Adam optimizer with the initial learning rate of $lr = 5.5 \cdot 10^{-4}$ that is halved after 40 and 70 epochs. All these experiments are done over eight different codebook bits of $B = \{4, 5, 6, 7, 8, 9, 10, 11\}$. For example, for $B = 8$, the number of codewords equals $K = 2^8 = 256$. To demonstrate that our proposed DiVeQ and SF-DiVeQ techniques perform consistently superior to the other VQ optimization methods in the image compression task, we do ablation studies over different batch sizes and learning rates which are presented in App. C.3 and App. C.4, respectively. The variance $\sigma^2$ of directional noise $\mathbf{v}_d$ for DiVeQ and SF-DiVeQ is set to $\sigma^2 = 10^{-3}$ for VQ-VAE compression (see the ablation study on $\sigma^2$ in App. C.5).

Our proposed codebook replacement (App. B.1) is actively applied during training for all VQ optimization methods (except SF-DiVeQ, which does not require it). The replacement is done in two different phases of

$$
\begin{cases}
iter \leq 2000 & \text{; replacement after each 100 training iterations} \\
2000 < iter \leq N_{iter} - 1000 & \text{; replacement after each 500 training iterations,}
\end{cases}
$$

where $iter$ is the training iteration number, and $N_{iter}$ is the total number of training iterations. The discarding threshold for the replacement equals $0.01$, which means that the codebook replacement discards the codewords that are used less than 1% during the period that the replacement is done.

**Model architecture** For the image compression task, we aim to use the original VQ-VAE proposed in van den Oord et al. (2017). The VQ-VAE implementation is provided in DeepMind's GitHub[1], and its PyTorch version in *zalandoresearch* GitHub repository[2]. These implementations are meant for images of size $32 \times 32$. Hence, to make the VQ-VAE suitable for compressing images of size $256 \times 256$, we make some minor modifications to these implementations and add the LPIPS (VGG-16) as the perceptual loss to enhance the quality of reconstructions.

Table 4 shows the architecture of the VQ-VAE used in our image compression experiments. The encoder consists of four downsampling blocks, a stack of six residual blocks, and a pre-VQ Conv2D layer. Each downsampling block is a strided 2D convolutional layer with $\text{kernel size} = 4$ and $\text{stride} = 2$ followed by a ReLU activation function. Each residual block is comprised of ReLU, $3 \times 3$ Conv2D ($\text{stride} = 1$), ReLU, and $1 \times 1$ Conv2D ($\text{stride} = 1$) in this order. The pre-VQ Conv2D layer is meant to map the input channels to match the VQ embedding dimension. The decoder has a symmetric architecture to the encoder, but in a reverse order, with a difference that it uses transposed 2D convolutions in the upsampling blocks.

Table 4: Architecture of the VQ-VAE model used in the image compression task.

| Encoder | Decoder |
|---|---|
| $\boldsymbol{x} \in \mathbb{R}^{H \times W \times C}$ | $\boldsymbol{z}_q \in \mathbb{R}^{h \times w \times C''}$ |
| $4 \times \{\text{Downsampling Block}\} \to \mathbb{R}^{h \times w \times C'}$ | $\text{Conv2D} \to \mathbb{R}^{h \times w \times C'}$ |
| $6 \times \{\text{Residual Block}\} \to \mathbb{R}^{h \times w \times C'}$ | $6 \times \{\text{Residual Block}\} \to \mathbb{R}^{h \times w \times C'}$ |
| $\text{Conv2D} \to \mathbb{R}^{h \times w \times C''}$ | $4 \times \{\text{Upsampling Block}\} \to \mathbb{R}^{H \times W \times C}$ |
| $H = W = 256, \quad h = w = 16, \quad C = 3, \quad C' = 256, \quad C'' = 512$ | |

[1] https://github.com/google-deepmind/sonnet/blob/v2/examples/vqvae_example.ipynb

[2] https://github.com/zalandoresearch/pytorch-vq-vae

## A.2 VQGAN GENERATION

**Hyperparameters** In VQGAN generation task, we train the generator model (*i.e.*, VQ-VAE) with different VQ optimization methods for 100 epochs over codebook bits of $B = \{8, 9, 10, 12\}$ and over two different hyperparameter settings of $\text{HP}_1 : (lr = 2.5 \cdot 10^{-5}, \text{batch size} = 8)$ and $\text{HP}_2 : (lr = 2.5 \cdot 10^{-4}, \text{batch size} = 32)$. During training, the initial learning rate is halved after 50 and 75 epochs. The discriminator starts at epoch 50 when half of the training is passed. All GPT transformers (with different configurations in Table 5) are trained with a batch size of 32 and an initial learning rate of $lr = 4.5 \cdot 10^{-5}$ with cosine decay that reduces the learning rate to 1% of its initial value. For sampling from VQGAN, we use the temperature of $t = 1.0$ with different top-$k$ values over different codebook sizes (see Table 6). The variance $\sigma^2$ of directional noise $\mathbf{v}_d$ for DiVeQ and SF-DiVeQ is set to $\sigma^2 = 10^{-2}$ for VQGAN generation.

When training the generator (VQ-VAE) model for all VQ optimization methods (except SF-DiVeQ), we apply our proposed codebook replacement (App. B.1) actively during training in two phases of

$$
\begin{cases}
iter \leq 5000 & ; \text{ replacement after each 50 training iterations} \\
5000 < iter \leq N_{iter} - 3000 & ; \text{ replacement after each 300 training iterations,}
\end{cases}
$$

where *iter* is the training iteration number, and $N_{iter}$ is the total number of training iterations. The discarding threshold for the replacement equals 0.01, which means that the codebook replacement discards the codewords that are used less than 1% during the period that the replacement is done.

**Model architectures** For the VQGAN generation task, we use the same architectures as in Esser et al. (2021) for the generator (VQ-VAE), discriminator, and transformer models. We adopt the implementation from *dome272* GitHub repository[3] with some suggested minor modifications from this repository[4].

**Training transformers** For all data sets, the GPT transformer is trained with a batch size of 32 and an initial learning rate of $lr = 4.5 \cdot 10^{-5}$ with a cosine decay that reduces the learning rate to 1% of its initial value over the course of training. In addition, we adopt a linear warm-up for the learning rate such that $lr$ starts with a really small value and linearly reaches its initial learning rate during the first 10k training iterations.

When training the transformer, one important variable to tune is the probability of keeping the ground truth indices, which is called $p_{\text{keep}} \in (0, 1)$. In other words, the true indices of the tokens are masked with the probability of $1 - p_{\text{keep}}$. The reason to mask the true tokens is that it helps the transformer to become more robust against its own mistakes, such that after one wrong prediction during sampling, it does not propagate this error with a snowball effect to all other next predictions. In addition, masking true tokens helps the transformer not to memorize the token dependencies, which leads to better generalization and prevents overfitting.

For all transformers, we set a scheduled masking rate that increases the $p_{\text{keep}}$ from the initial value of 0.5 to 0.95 over training epochs. In the beginning, we mask half of the true token indices, forcing the transformer to learn large missing parts, *i.e.*, the transformer learns dependencies of coarse structures in the images. As the training goes to the end, a small portion of token indices will be masked, and as a result, the transformer learns dependencies of finer details in the images.

To train the transformers for FFHQ and CELEBA-HQ data sets, we use the configurations suggested in Esser et al. (2021). Also, we use the same transformer configuration for the LSUN Bedroom and LSUN Church data sets. Since the AFHQ data set is much smaller than the other data sets (containing 15803 images), we adopted a different and lighter transformer for it. Table 5 shows the transformers configuration for different data sets. Note that when experimenting with different codebook sizes, we keep the transformer configuration fixed.

It is noteworthy to mention again here that since having more data favors for the VQGAN generation task, for CELEBA-HQ and AFHQ data sets, we use the full data sets to train the generator (VQ-VAE) and transformer, and to compute the FID scores. CELEBA-HQ and AFHQ full data sets contain 30k and 15803 images, respectively. However, for FFHQ and both LSUN data sets, we only consider the train sets that contain 56k images.

---

[3]https://github.com/dome272/VQGAN-pytorch
[4]https://github.com/aa1234241/vqgan

Table 5: Configurations used to train the transformers for different data sets in the VQGAN generation task. $n_{\text{layer}}$ is the number of transformer blocks, $n_{\text{head}}$ refers to the number of attention heads, $D_{\text{embedding}}$ is the dimensionality of the transformer embeddings, $s$ is the length of the sequence, *#params* is the number of transformer parameters, *dropout* refers to the dropout rate used to train the transformer, and $D_{\text{latent}}$ is the dimensionality of each codeword.

| Data set | $n_{\text{layer}}$ | $n_{\text{head}}$ | $D_{\text{embedding}}$ | length $(s)$ | *dropout* | *#params* $[M]$ | $D_{\text{latent}}$ |
|---|---|---|---|---|---|---|---|
| CELEBA-HQ | 28 | 16 | 1024 | 256 | 0.1 | $\approx 355$ | 256 |
| FFHQ | 28 | 16 | 1024 | 256 | 0.1 | $\approx 355$ | 256 |
| LSUN Bedroom | 28 | 16 | 1024 | 256 | 0.1 | $\approx 355$ | 256 |
| LSUN Church | 28 | 16 | 1024 | 256 | 0.1 | $\approx 355$ | 256 |
| AFHQ | 20 | 16 | 768 | 256 | 0.1 | $\approx 143$ | 256 |

**Sampling from VQGAN** When the transformers are trained, for each data set, we sample the same number of images as the train set (full data sets for CELEBA-HQ and AFHQ) with the configuration mentioned in Table 6. To compute the FID score between the train set and new sampled images, we use *clean-fid* from *GaParmar* GitHub repository[5].

Table 6: Configuration used for generating images from trained VQGAN models for different codebook sizes. This configuration is fixed for different data sets.

| Codebook bits | Codebook size $(K)$ | Sampling temperature $(t)$ | top-$k$ |
|---|---|---|---|
| 8 | 256 | 1.0 | 75 |
| 9 | 512 | 1.0 | 150 |
| 10 | 1024 | 1.0 | 300 |
| 12 | 4096 | 1.0 | 300 |

### A.3 DAC SPEECH CODING

**Hyperparameters** In the speech coding task (for Table 3 in Sec. 4), we train all VQ methods for 300 epochs with the batch size of 64 using the AdamW optimizer with a learning rate of $lr = 10^{-4}$ and $betas = (0.8, 0.99)$. All these experiments are done over four different codebook bits of $B = \{10, 11, 12, 13\}$. For example, for $B = 10$ the number of codewords equals $K = 2^{10} = 1024$. To demonstrate that our proposed DiVeQ and SF-DiVeQ techniques perform consistently superior to the other VQ optimization methods in the speech coding task, we do ablation studies over different batch sizes, which are presented in Tables 8 and 9 in App. C.3. The variance $\sigma^2$ of directional noise $\mathbf{v}_d$ for DiVeQ and SF-DiVeQ is set to $\sigma^2 = 10^{-3}$ in the experiments.

Note that our proposed codebook replacement (App. B.1) is actively applied during training for all VQ optimization methods (except SF-DiVeQ, which does not require it). The replacement is done with the same schedule as in the VQGAN generation task, explained in App. A.2.

**Model architecture** For the speech coding task, we use a modified version of the Descript Audio Codec (DAC, Kumar et al., 2023) called Pikku NAC[6], which consists of a convolutional encoder, a residual vector quantization (RVQ) layer, and a decoder. Both the decoder and the encoder have four layers that follow the same architecture as DAC, with a stride pattern of $2:4:8:8$ in the encoder and $8:8:4:2$ in the decoder, resulting in a frame size of $512$. However, unlike the DAC model, Pikku NAC uses 32 encoder channels, 256 decoder channels, and a kernel size of 5, resulting in a significantly smaller model with only 5M encoder parameters and 2M decoder parameters. The 8-dim latent space is quantized with a 9-layer RVQ (*i.e.*, using 9 codebooks), where each codebook contains 1024 codewords, resulting in 156k parameters.

In our experiments, we change the residual VQ layer in Pikku NAC to an ordinary VQ layer, which contains only one codebook. To compensate for the loss in the number of codewords used for the quantization, we did the experiments using large codebook sizes (*i.e.*, $K = \{2^{10}, 2^{11}, 2^{12}, 2^{13}\}$). Both the DAC and Pikku NAC models project the 512-dimensional latent space to 8 dimensions

---

[5]https://github.com/GaParmar/clean-fid
[6]https://github.com/eagomez2/pikku-nac

($D = 8$) before quantization and project the quantized latent vectors back into dimension 512. However, in order not to have a tight bottleneck and make the quantization task more challenging, we keep the latent space to be of dimension 512. We achieve this by keeping the convolution layers that apply the projections, but making them project to the same dimension of 512.

**Data** As mentioned in Sec. 4, for the speech coding task, we use the CSTR VCTK data set[7] (Yamagishi et al., 2019) with an 80/20% train-test split without any overlapping speakers and speech files between train and test sets. This data set contains 109 native English speakers with different accents, each reading about 400 sentences. As an additional curation step, we used signal-to-noise ratio (SNR) thresholding to remove silent segments between utterances and discarded samples with significant background noise. Since Pikku NAC is prepared for training wideband speech that is sampled at 16 kHz, we downsample the data during training to meet this sampling rate.

### A.4 GUMBEL-SOFTMAX IMPLEMENTATION

For VQ-VAE compression, VQGAN generation, and DAC speech coding, the implementation of Straight-Through Gumbel-Softmax (ST-GS) is adopted from *karpathy* GitHub repository[8]. The ST-GS temperature $\tau$ is exponentially annealed from $\tau_{\text{start}} = 1$ to $\tau_{\text{min}} = 0.1$ over training epochs such that

$$\tau = \max\{\tau_{\text{start}} * \eta^{\text{epoch}}, \tau_{\text{min}}\}; \quad \eta = \left(\frac{\tau_{\text{min}}}{\tau_{\text{start}}}\right)^{\frac{1}{N_{\text{epochs}}}}, \tag{13}$$

where $\eta$ is the decay rate for the temperature annealing, epoch is the current epoch number, and $N_{\text{epochs}}$ is the total number of training epochs.

### A.5 ROTATION TRICK IMPLEMENTATION

We adopt the implementation of the Rotation Trick (RT) from *lucidrains* GitHub repository[9]. Following what is suggested in the RT paper (Fifty et al., 2025), for our VQ-VAE compression task, we first trained the VQ-VAE by updating the codebook via Exponential Moving Averages (EMA) with the decay rate of $\gamma = 0.8$. Since the results were not good enough, we changed the decay rate to $\gamma = 0.99$, which slightly improved the quality of reconstructions. However, we finally noticed that by skipping the EMA for updating the codebook and using codebook loss (2nd loss term in Eq. (2)), we achieve the best possible performance that we can get from the Rotation Trick in our VQ-VAE compression task. For the VQGAN generation task, the RT paper suggested updating the VQ codebook via gradients using codebook loss. Therefore, for both of our VQ-VAE compression and VQGAN generation tasks, we train the RT using the codebook loss.

### A.6 SUGGESTIONS FOR TRAINING THE CODEBOOK USING SF-DIVEQ

As discussed in Sec. 4, for training the codebook via SF-DiVeQ, it is recommended to skip quantization for several initial epochs (or several thousand training iterations) in order to reach a more stable latent representation that already passed its initial big distribution shifts. After that, initialize the codebook vectors with the average of recent latent vectors and start discretizing the latent representation using the SF-DiVeQ approach.

For the initialization of the codebook, it is highly recommended to follow these two guidelines; *(i)* take the very recent latent vectors from the last 20 to 50 training iterations as the set of vectors for initialization, *(ii)* according to the batch size, initialize each codebook vector to be the average of at least 20 to 40 recent latent vectors. For instance, suppose the encoder compresses the input image of size $256 \times 256$ to a latent of size $16 \times 16$. So, for a batch size of 8, each training iteration would generate $8 \times 16 \times 16 = 2048$ latent vectors. Hence, for a codebook of size $K = 1024$, to initialize each codebook vector with the average of 40 latent vectors, we need to capture the latent vectors of the last 20 training iterations such that

$$\frac{N_{\text{latents}}}{K} = 40 \quad \Rightarrow \quad N_{\text{iters}} = \frac{N_{\text{latents}}}{2048} = \frac{40 \times 1024}{2048} = 20 \tag{14}$$

---

[7] https://datashare.ed.ac.uk/handle/10283/2950
[8] https://github.com/karpathy/deep-vector-quantization
[9] https://github.com/lucidrains/vector-quantize-pytorch

where $N_{\text{latents}}$ is the number of recent captured latent vectors, and $N_{\text{iters}}$ is the number of recent training iterations required for obtaining $N_{\text{latents}}$ latent vectors. Therefore, for smaller batch sizes, it is better to capture more training iterations compared to larger batch sizes.

The most important point in training codebook via SF-DiVeQ is to track the codebook usage percentage over training epochs (the plot at the top in Fig. 29). If quantization with SF-DiVeQ is started after big latent distribution shifts and by capturing enough recent latent vectors, the initial codebook usage would be a high value (approximately close to the maximum possible value that equals the number of codebook vectors $K$). Then, according to the SF-DiVeQ property, it pulls the codewords that are outside the latent representation inside during training iterations. Therefore, the codebook usage will reach its maximum value $K$ after some epochs of starting to quantize the latent space, and will be kept at its maximum until the end of training. Fig. 29 presents an example of how codebook usage percentage looks for a proper SF-DiVeQ training.

## B  OTHER PROPOSALS

### B.1  NEW PROPOSED CODEBOOK REPLACEMENT

According to Huh et al. (2023), codebook replacement is a suitable solution to avoid *codebook collapse* (Vali, 2025; Mentzer et al., 2023) for different VQ optimization techniques. In NSVQ (Vali & Bäckström, 2022), a codebook replacement method is proposed such that after a period of training iterations, unused codewords are replaced with a permutation of some used codewords. Selection from the used codewords is completely random, regardless of the importance of the used codewords. However, in this paper, we propose a new codebook replacement technique that replaces unused codewords with the used ones based on the importance of the used codewords. In other words, we use an *importance sampling* strategy that computes the probability of codeword occurrences (during a period of training iterations), and replaces the unused codewords by selecting from the used codewords according to these probabilities. Therefore, a codeword with a higher occurrence probability is more likely to be selected for the replacement. This is a logical approach for replacement, as in quantization applications, the main goal is to reduce the average quantization error, and when the number of assigned inputs to a codeword is high, it is better to define a new codeword in that region to reduce the average quantization error.

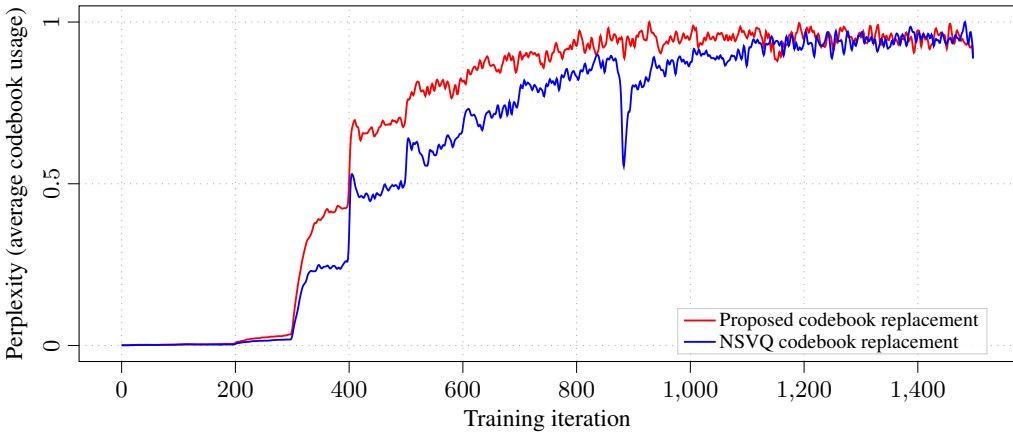

Figure 8: **Proposed codebook replacement achieves a more stable codebook utilization faster than NSVQ's replacement method.** Comparison of the perplexity of our proposed codebook replacement with NSVQ's replacement technique for the initial 1500 iterations when training the VQ-VAEs on CELEBA-HQ data set using the DiVeQ approach. The codebook size equals $K = 2048$.

Fig. 8 compares the perplexity (or average codebook usage) of our proposed codebook replacement and NSVQ's replacement technique (Vali & Bäckström, 2022) in the VQ-VAE compression of the CELEBA-HQ data set using the DiVeQ approach with an 11-bit codebook (with $K = 2^{11} = 2048$ codewords). For each training iteration, perplexity shows the number of selected codewords involved

in quantizing that batch of data, and it is computed as

$$\text{perplexity} = \exp(H(\mathcal{C})) \quad \text{s.t.} \quad H(\mathcal{C}) = -\sum_{k=1}^{K} p_k \cdot \log p_k, \tag{15}$$

where $H(\mathcal{C})$ is the entropy of the codebook index distribution, and $p_k$ is the empirical usage probability of the $k$-th codebook vector. Fig. 8 is plotted for the first 1500 training iterations, while the curves are smoothed via Blackman windowing with a window size of 11 and normalized by their maximum value. According to the figure, our proposed codebook replacement reaches higher perplexity values faster than NSVQ's replacement approach, and as a result, it provides a longer training opportunity for the active codewords to be trained for more training iterations.

Fig. 9 shows the quantitative results of the reconstructions when using our proposed codebook replacement and NSVQ's replacement method, shown in Fig. 8. We observe that our proposed codebook replacement results in slightly better performance by having higher SSIM, PSNR, and lower LPIPS values. Note that based on our investigations, our proposed codebook replacement can result in similar or better performance than NSVQ's replacement over different experiments. Therefore, we use our proposed codebook replacement for all VQ optimization techniques (*i.e.*, STE, EMA, RT, ST-GS, NSVQ, and DiVeQ) and for all experiments in this paper.

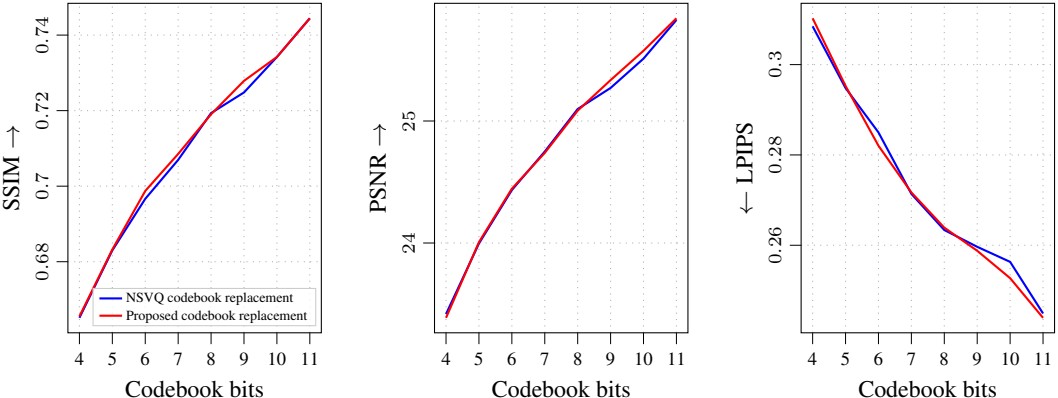

Figure 9: **Proposed codebook replacement improves the quality of reconstructions compared to NSVQ's replacement method.** Quantitative comparison of reconstructed images from the CELEBA-HQ test set when using our proposed codebook replacement vs. NSVQ's replacement method. The VQ-VAEs are trained via the DiVeQ approach over different codebook sizes.

### B.2 PROPOSED VARIANTS OF DIVEQ AND SF-DIVEQ

Apart from what is proposed for the formulation of our proposed DiVeQ (Eq. (8)) and SF-DiVeQ (Eq. (12)), we also propose two new variants of them by skipping the directional noise $\mathbf{v}_d$ and using the stop gradient operator (*i.e.*, *detach* in PyTorch). We call these two variants DiVeQ-*detach* and SF-DiVeQ-*detach*, and their formulation is

$$\text{DiVeQ-}detach : \boldsymbol{z}_q = \boldsymbol{z} + \|\boldsymbol{c}_{i^*} - \boldsymbol{z}\|_2 \cdot sg\left[\frac{\boldsymbol{c}_{i^*} - \boldsymbol{z}}{\|\boldsymbol{c}_{i^*} - \boldsymbol{z}\|_2}\right],$$

$$\text{SF-DiVeQ-}detach : \boldsymbol{z}_q = \boldsymbol{z} + \|\boldsymbol{c}_{i^*} - \boldsymbol{z}\|_2 \cdot sg\left[\frac{(1 - \lambda_{i^*})(\boldsymbol{c}_{i^*} - \boldsymbol{z})}{\|\boldsymbol{c}_{i^*} - \boldsymbol{z}\|_2}\right] \tag{16}$$

$$+ \|\boldsymbol{c}_{i^*+1} - \boldsymbol{z}\|_2 \cdot sg\left[\frac{\lambda_{i^*}(\boldsymbol{c}_{i^*+1} - \boldsymbol{z})}{\|\boldsymbol{c}_{i^*+1} - \boldsymbol{z}\|_2}\right].$$

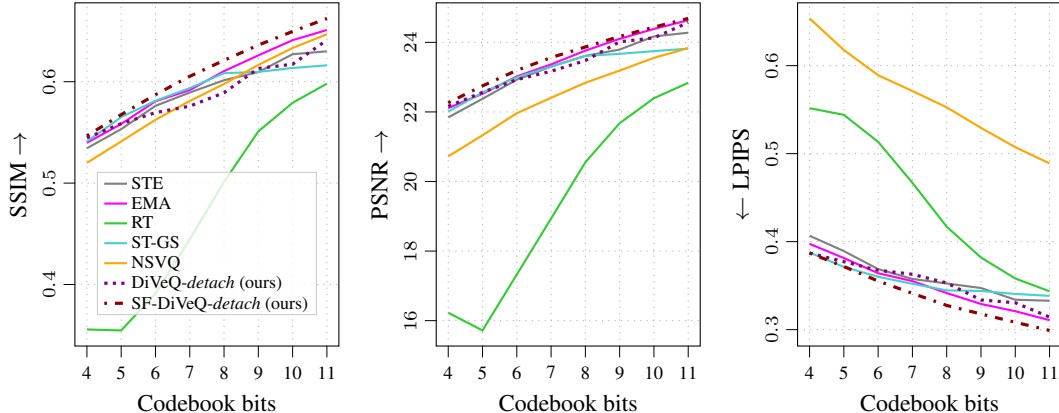

Figure 10: **Variants of our proposed methods: SF-DiVeQ-*detach* consistently improves image reconstruction and DiVeQ-*detach* maintains the reconstruction quality.** Quantitative comparison of reconstructed images from the AFHQ test set using variants of our proposed methods vs. other VQ optimization approaches. Each curve is the average of the metric over all test set reconstructions and over three different individual runs.

We experiment with these two variants only in the VQ-VAE compression task to evaluate how they perform compared to DiVeQ and SF-DiVeQ, and also other existing techniques. We train the VQ-VAE compression model with a learning rate of $5.5 \cdot 10^{-4}$ and a batch size of 32 over 100 epochs on the AFHQ data set. Fig. 10 shows the quality of reconstructions for DiVeQ-*detach* and SF-DiVeQ-*detach* compared to other approaches of STE, EMA, RT, ST-GS, and NSVQ. According to the figure, similar to its original version, SF-DiVeQ-*detach* consistently outperforms other methods for all three objective metrics and across all different codebook sizes. However, DiVeQ-*detach* performs comparable to other VQ methods. In addition, Fig. 11 compares the quality of reconstructions of DiVeQ-*detach* and SF-DiVeQ-*detach* with their original approaches of DiVeQ and SF-DiVeQ. According to Fig. 11, DiVeQ performs clearly superior to its variant (*i.e.*, DiVeQ-*detach*), while SF-DiVeQ and its variant perform almost similarly.

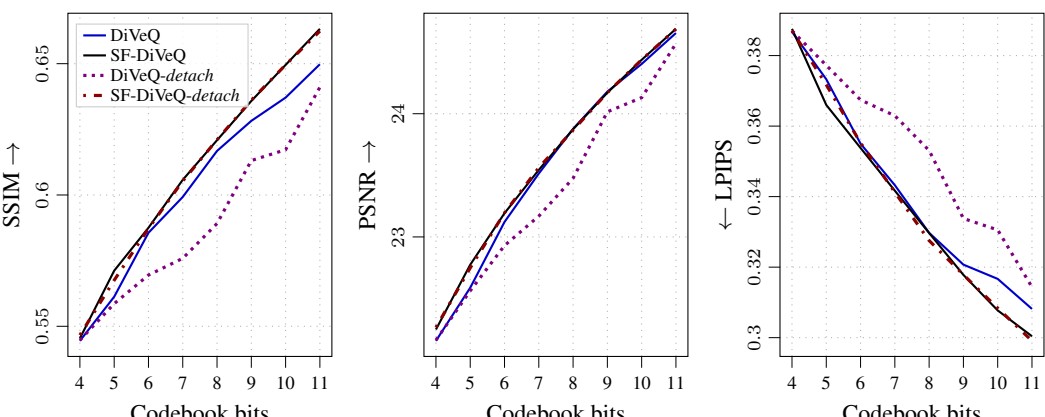

Figure 11: **Variants of our proposed methods: Regarding reconstruction quality, SF-DiVeQ-*detach* performs almost similar and DiVeQ-*detach* performs worse than their original versions.** Quantitative comparison of reconstructed images from the AFHQ test set using variants of our proposed methods vs. their original versions. Each curve is the average of the metric over all test set reconstructions and over three different individual runs.

According to VQ-VAE compression experiments on other data sets of CELEBA-HQ, FFHQ, LSUN Bedroom, and LSUN Church, we find that DiVeQ and SF-DiVeQ always result in higher quality reconstructions compared to their variants. The performance gap is not significant, and even in some cases (like SF-DiVeQ in Fig. 11), DiVeQ-*detach* and SF-DiVeQ-*detach* can perform close to their original versions. The main difference is that the *detach* variants do not need to set the variance $\sigma^2$ (in directional noise $\mathbf{v}_d$). However, we do not consider this an advantage, since the variance in DiVeQ and SF-DiVeQ can be used to control the amount of generalization for the downstream task.

## C  ADDITIONAL RESULTS

### C.1  QUANTITATIVE RESULTS ON OTHER DATA SETS FOR VQ-VAE COMPRESSION

Apart from the results obtained for the AFHQ data set in Fig. 6, we also assess the quality of reconstructions in the VQ-VAE compression task for CELEBA-HQ, FFHQ, LSUN Bedroom, and LSUN Church data sets. As mentioned earlier, for all data sets, we train the VQ-VAE models using the configurations mentioned in App. A.1. After training, we compress and reconstruct the test set images by doing hard VQ (using *argmin*) in the latent space. Then, we compute the SSIM, PSNR, and LPIPS metrics for the reconstructions by having the original ground-truth image available. Figs. 12 to 15 show the results for CELEBA-HQ, FFHQ, LSUN Bedroom, and LSUN Church data sets, respectively. Note that in the plots, the reported value for each metric is the average of that metric over all test set reconstructions and over three different individual runs.

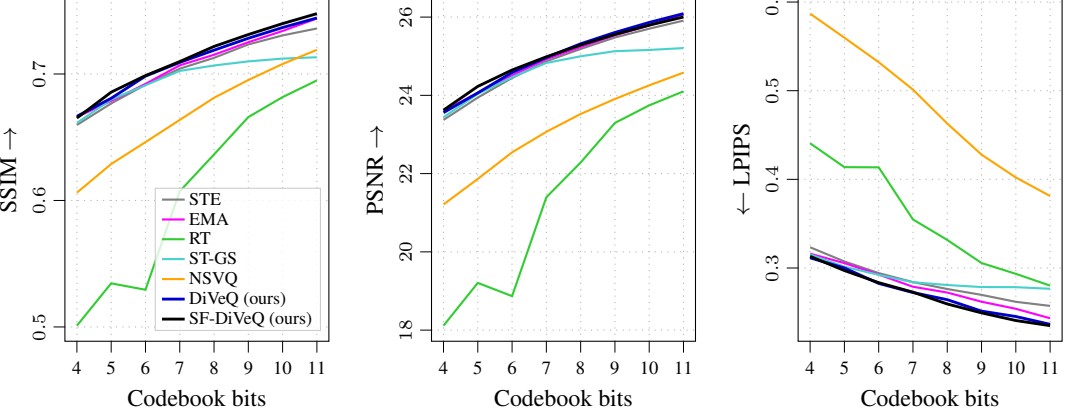

Figure 12: **The proposed DiVeQ and SF-DiVeQ lead to consistent improvement in image reconstruction quality.** Quantitative comparison of reconstructed images from the CELEBA-HQ test set for different VQ optimization methods over different codebook sizes. Each curve is the average of the metric over all test set reconstructions and over three different individual runs.

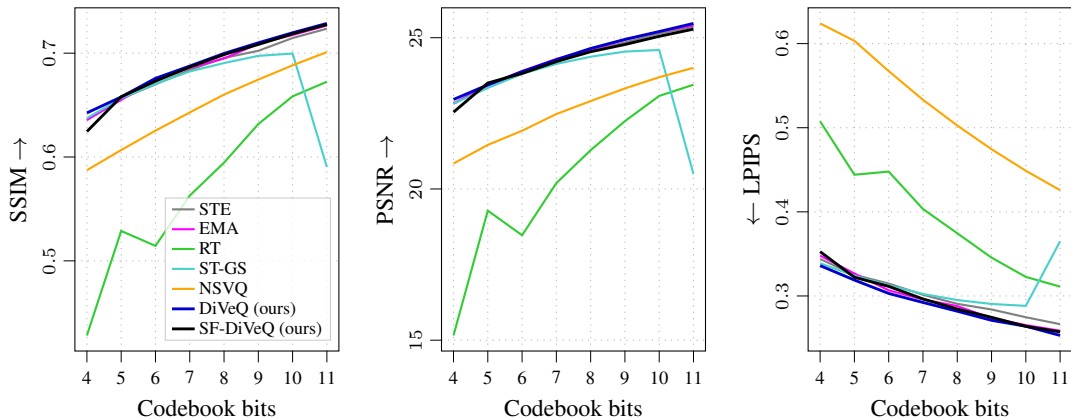

Figure 13: **The proposed DiVeQ and SF-DiVeQ lead to consistent improvement in image reconstruction quality.** Quantitative comparison of reconstructed images from the FFHQ test set for different VQ optimization methods over different codebook sizes. Each curve is the average of the metric over all test set reconstructions and over three different individual runs.

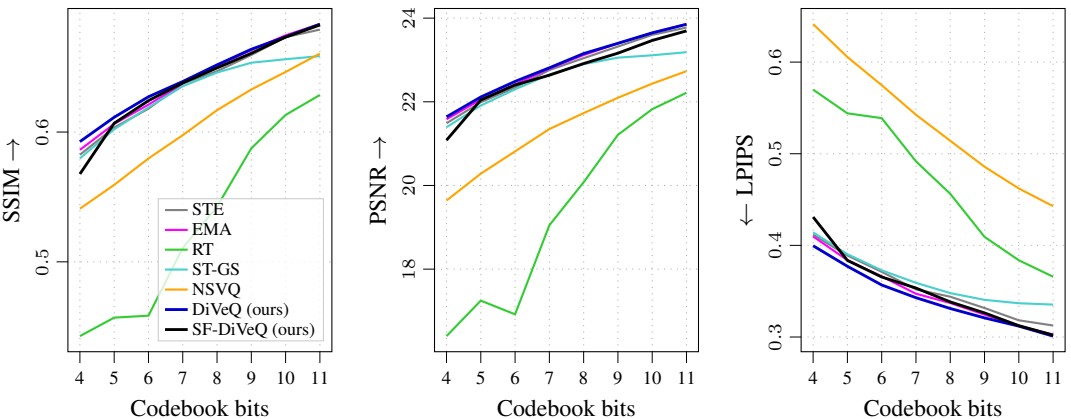

Figure 14: **The proposed DiVeQ and SF-DiVeQ lead to consistent improvement in image reconstruction quality.** Quantitative comparison of reconstructed images from the LSUN Bedroom test set for different VQ optimization methods over different codebook sizes. Each curve is the average of the metric over all test set reconstructions and over three different individual runs.

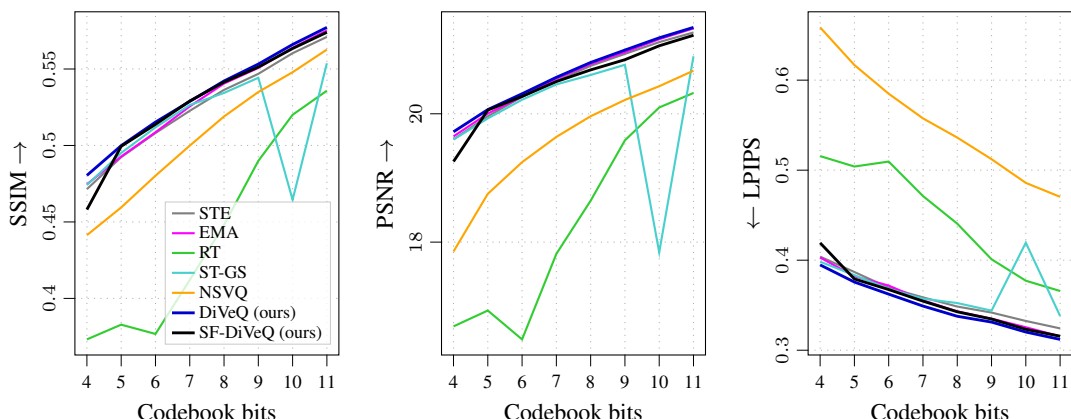

Figure 15: **The proposed DiVeQ and SF-DiVeQ lead to consistent improvement in image reconstruction quality.** Quantitative comparison of reconstructed images from the LSUN Church test set for different VQ optimization methods over different codebook sizes. Each curve is the average of the metric over all test set reconstructions and over three different individual runs.

## C.2 QUANTITATIVE RESULTS ON OTHER DATA SETS FOR VQGAN GENERATION

Apart from the FID values obtained for CELEBA-HQ data set in Table 2, we also evaluate the quality of generations for other data sets by computing the FID scores. As mentioned earlier, for all data sets, we train the VQGAN generator (*i.e.*, VQ-VAE), discriminator, and transformer models using the configurations mentioned in App. A.2 over two different hyperparameter settings of $HP_1 : (lr = 2.5 \cdot 10^{-5}, \text{batch size} = 8)$ and $HP_2 : (lr = 2.5 \cdot 10^{-4}, \text{batch size} = 32)$. Then after the models are trained, we sample new generations from them and compute the FID value between the original training set and the generated samples (see Table 6 for sampling configuration). Table 7 presents the obtained FID scores for FFHQ, AFHQ, LSUN Bedroom, and LSUN Church data sets using different VQ optimization methods and over various codebook sizes. Figs. 30 to 34 provide the qualitative comparison of VQGAN generations for all data sets with a 9-bit codebook.

Table 7: **The proposed DiVeQ and SF-DiVeQ robustify training and maintain the generation quality for challenging small codebooks, while being less sensitive to hyperparameter settings.** The table presents the FID↓ scores for different VQ optimization methods over different codebook sizes for FFHQ, LSUN Bedroom, LSUN Church, and AFHQ data sets. Values *in red* refer to the cases where misalignment happens (see Fig. 4).

| Data set | Approach | $lr = 2.5 \cdot 10^{-5}$ batch size $= 8$ Codebook bits | | | | $lr = 2.5 \cdot 10^{-4}$ batch size $= 32$ Codebook bits | | | |
|---|---|---|---|---|---|---|---|---|---|
| | | 8 | 9 | 10 | 12 | 8 | 9 | 10 | 12 |
| FFHQ | STE | 6.74 | 6.01 | 5.36 | 7.31 | 273 | **8.77** | **7.49** | 356 |
| | EMA | **6.53** | 6.57 | 6.15 | 619 | **8.65** | 9.00 | 8.35 | 11.5 |
| | RT | 14.2 | 7.19 | 6.52 | **4.73** | 63.3 | 11.7 | 9.35 | **7.86** |
| | ST-GS | 22.5 | 21.9 | 17.0 | 7.97 | 54.4 | 274 | 36.8 | 70.6 |
| | NSVQ | 98.0 | 88.8 | 83.8 | 65.1 | 84.0 | 76.0 | 71.4 | 63.1 |
| | DiVeQ (ours) | 6.70 | **5.61** | **5.11** | 8.22 | 8.97 | 9.23 | 8.27 | 10.3 |
| | SF-DiVeQ (ours) | 7.91 | 6.21 | 5.93 | 8.60 | 10.3 | 9.41 | 8.04 | 9.74 |
| LSUN Bedroom | STE | **5.27** | 4.89 | **4.96** | 7.54 | 225 | 67.5 | 6.38 | 8.04 |
| | EMA | 6.04 | 5.36 | 5.56 | **5.13** | 8.05 | 584 | 584 | 584 |
| | RT | 17.4 | 8.65 | 6.69 | 5.18 | 255 | 53.5 | 42.9 | 25.5 |
| | ST-GS | 35.9 | 37.2 | 36.4 | 23.0 | 172 | 223 | 187 | 210 |
| | NSVQ | 67.8 | 50.5 | 45.6 | 37.5 | 61.5 | 50.1 | 44.0 | 37.5 |
| | DiVeQ (ours) | 5.71 | **4.87** | 5.40 | 7.94 | 7.34 | 347 | 6.58 | 9.03 |
| | SF-DiVeQ (ours) | 5.96 | 6.01 | 5.40 | 7.79 | **7.16** | **6.69** | **6.36** | **7.62** |
| LSUN Church | STE | **4.33** | **3.74** | 4.09 | 4.91 | **6.69** | 9.92 | **5.67** | 5.92 |
| | EMA | 4.39 | 4.37 | 3.81 | **3.88** | 6.92 | 5.84 | 6.31 | 5.24 |
| | RT | 9.33 | 6.37 | 5.20 | 4.31 | 78.5 | 142 | 134 | 259 |
| | ST-GS | 24.4 | 26.9 | 26.5 | 13.7 | 48.3 | 259 | N/A | 108 |
| | NSVQ | 107 | 92.8 | 71.7 | 50.9 | 85.6 | 76.9 | 61.6 | 47.8 |
| | DiVeQ (ours) | 4.41 | 3.92 | 3.90 | 4.58 | 7.91 | 228 | 9.17 | 6.11 |
| | SF-DiVeQ (ours) | 4.35 | 3.99 | **3.56** | 4.40 | 7.74 | **5.74** | 10.9 | **4.57** |
| AFHQ | STE | 5.45 | **5.24** | 5.41 | 6.48 | 250 | **6.50** | 253 | **7.91** |
| | EMA | **5.12** | 5.56 | **5.23** | 6.95 | **6.68** | 6.60 | 6.89 | 8.59 |
| | RT | 6.88 | 6.31 | 5.55 | **5.54** | 250 | 248 | 288 | 56.3 |
| | ST-GS | 31.3 | 26.5 | 20.9 | 14.4 | 302 | 509 | 231 | 224 |
| | NSVQ | 92.0 | 81.3 | 72.2 | 47.1 | 70.0 | 64.0 | 53.0 | 45.0 |
| | DiVeQ (ours) | 6.30 | 5.63 | 5.78 | 7.33 | 7.60 | 6.82 | **6.75** | 8.49 |
| | SF-DiVeQ (ours) | 5.63 | 5.40 | 5.69 | 6.86 | 7.95 | 7.04 | 6.87 | 8.39 |

## C.3 ABLATION ON BATCH SIZE

In this section, we study the effect of the batch size on different VQ optimization methods. To this end, in the VQ-VAE compression task, we train all different methods with batch sizes of $64$ and $128$ at the learning rate of $lr = 5.5 \cdot 10^{-4}$ on the AFHQ data set for $100$ epochs. The learning rate is halved after $40$ and $70$ epochs. Fig. 16 and Fig. 17 show the quality of reconstructions for batch sizes $64$ and $128$, respectively. Note that in the plots, the reported value for each metric is the average of that metric over all test set reconstructions and over three different individual runs. Taking Fig. 6 (for batch size 32), Figs. 16 and 17 (for batch sizes 64 and 128) into account, we can claim that by increasing the batch size, the performance gap between our proposed methods and other approaches (except EMA) becomes larger. For larger batch sizes, EMA performs comparably but slightly worse than our DiVeQ and SF-DiVeQ. The reason is that EMA updates the codebook with the average of latent vectors, and as a result, larger batch sizes yield less noisy and more representative updates.

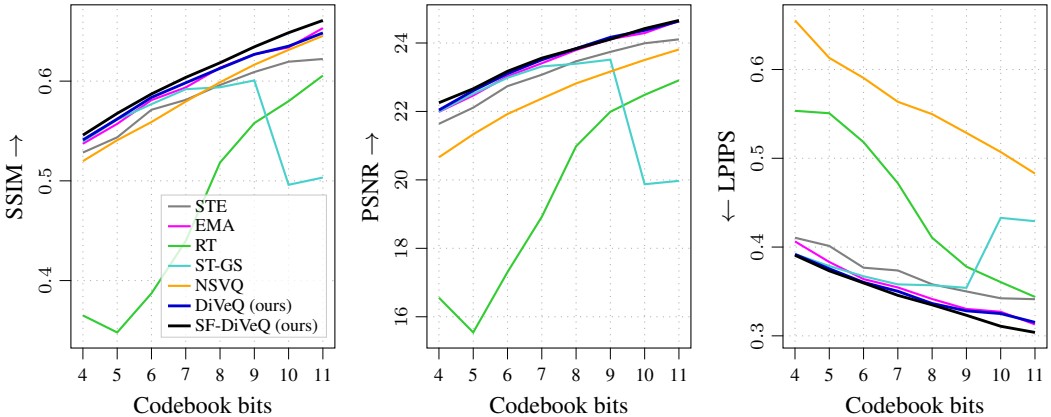

Figure 16: **Ablation on batch size: The proposed DiVeQ and SF-DiVeQ lead to consistent improvement in image reconstruction quality over different batch sizes (this figure, batch size = 64).** Quantitative comparison of reconstructed images from the AFHQ test set for different VQ optimization methods over different codebook sizes. Each curve is the average of the metric over all test set reconstructions and over three different individual runs.

In the VQ-VAE compression task, it is expected that the increase in codebook size always enhances the quality of reconstructions. However, this does not happen for some cases in Fig. 16 and Fig. 17. The reason is that the codebook representation $\mathcal{C}_z$ is not well fitted to the latent distribution $\mathcal{P}_z$, and as a result, a misalignment between codebook and latent representations is happened (see Fig. 4, Sec. 4, and App. C.8 for more details). In Fig. 16, the misalignments happen for [ST-GS|codebook bits=10,11], [RT|codebook bits=5], and in Fig. 17 they arise for [STE|codebook bits=11], [EMA|codebook bits=11], [ST-GS|codebook bits=9,11].

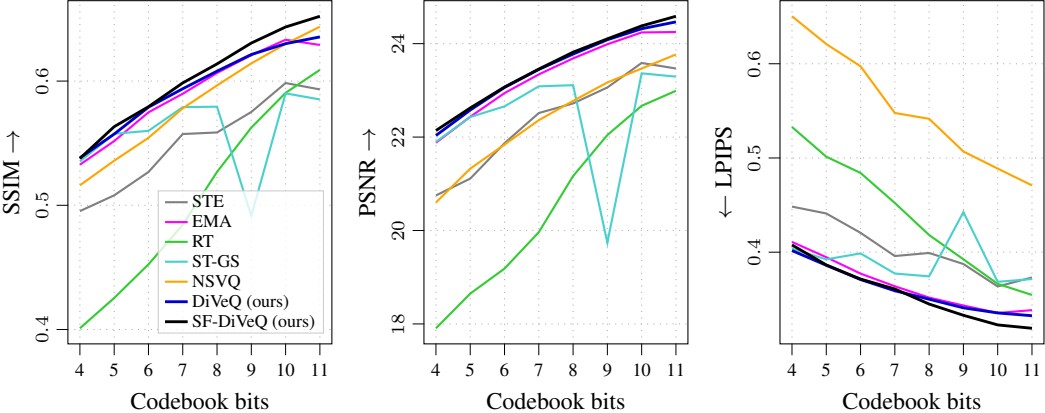

Figure 17: **Ablation on batch size: The proposed DiVeQ and SF-DiVeQ lead to consistent improvement in image reconstruction quality over different batch sizes (this figure, batch size = 128).** Quantitative comparison of reconstructed images from the AFHQ test set for different VQ optimization methods over different codebook sizes. Each curve is the average of the metric over all test set reconstructions and over three different individual runs.

We also evaluate how our proposed DiVeQ and SF-DiVeQ perform compared to other methods when changing the batch size in the DAC speech coding task. We train the DAC-based speech coding model using different VQ methods with batch sizes of 32 and 16 at the learning rate of $lr = 10^{-4}$ on the VCTK data set for 300 epochs. Table 8 and Table 9 show the quality of decompressed speech from the VCTK test samples for batch sizes of 32 and 16, respectively. The reported value for each metric refers to the average of that metric over all test set samples. In addition, entries highlighted *in red* correspond to runs where codebook–latent misalignment occurs, and the decoded speech is severely degraded and unintelligible. According to the quantitative comparisons in Tables 3, 8, 9,

our proposed DiVeQ and SF-DiVeQ consistently result in a superior quality for the decompressed speech samples when changing the training batch size.

Table 8: **Ablation on batch size: The proposed DiVeQ and SF-DiVeQ lead to consistent improvement in decompressed speech quality over different batch sizes (this table, batch size = 32).** Quantitative comparison of decompressed speech samples from the VCTK test set for different VQ optimization methods over different codebook sizes. Each value is the average of the metric over all test set samples.

| Approach \ bits | Log spectral distance↓ | | | | MFCC distance↓ | | | | PESQ↑ | | | | STOI↑ | | | |
|---|---|---|---|---|---|---|---|---|---|---|---|---|---|---|---|---|
| | 10 | 11 | 12 | 13 | 10 | 11 | 12 | 13 | 10 | 11 | 12 | 13 | 10 | 11 | 12 | 13 |
| STE | 1.13 | 1.07 | 1.05 | 1.06 | 104 | 88.9 | 86.6 | 88.5 | 1.16 | 1.28 | 1.32 | 1.31 | 0.71 | 0.78 | 0.78 | 0.77 |
| EMA | **1.01** | **1.00** | **1.00** | 1.24 | 75.0 | 72.4 | 70.2 | 147 | **1.51** | 1.57 | 1.61 | 1.08 | **0.83** | **0.84** | **0.85** | 0.59 |
| RT | 1.08 | 1.04 | 1.02 | 1.02 | 94.9 | 85.5 | 80.6 | 77.5 | 1.25 | 1.35 | 1.39 | 1.43 | 0.76 | 0.79 | 0.80 | 0.82 |
| ST-GS | 1.09 | 3.50 | 3.42 | 3.35 | 93.6 | 346 | 335 | 328 | 1.23 | 1.03 | 1.04 | 1.04 | 0.77 | 0.39 | 0.42 | 0.42 |
| NSVQ | 1.09 | 1.06 | 1.05 | 1.04 | 109 | 104 | 99.2 | 95.4 | 1.36 | 1.44 | 1.53 | 1.60 | 0.80 | 0.82 | 0.83 | 0.84 |
| DiVeQ (ours) | 1.03 | 1.01 | **1.00** | 0.99 | 79.5 | 74.8 | 72.1 | 68.9 | 1.41 | 1.54 | 1.66 | 1.74 | 0.81 | 0.83 | **0.85** | 0.85 |
| SF-DiVeQ (ours) | **1.01** | **1.00** | **1.00** | 0.98 | 73.6 | 71.5 | 69.5 | 65.5 | 1.47 | **1.65** | **1.69** | **1.78** | **0.83** | **0.84** | **0.85** | **0.86** |

Table 9: **Ablation on batch size: The proposed DiVeQ and SF-DiVeQ lead to consistent improvement in decompressed speech quality over different batch sizes (this table, batch size = 16).** Quantitative comparison of decompressed speech samples from the VCTK test set for different VQ optimization methods over different codebook sizes. Each value is the average of the metric over all test set samples.

| Approach \ bits | Log spectral distance↓ | | | | MFCC distance↓ | | | | PESQ↑ | | | | STOI↑ | | | |
|---|---|---|---|---|---|---|---|---|---|---|---|---|---|---|---|---|
| | 10 | 11 | 12 | 13 | 10 | 11 | 12 | 13 | 10 | 11 | 12 | 13 | 10 | 11 | 12 | 13 |
| STE | 1.04 | 1.07 | 1.03 | 1.04 | 84.2 | 91.7 | 83.2 | 83.3 | 1.42 | 1.20 | 1.40 | 1.38 | 0.81 | 0.75 | 0.80 | 0.80 |
| EMA | **1.00** | 0.99 | 0.99 | 7.70 | **76.1** | 73.8 | 71.0 | 488 | 1.52 | 1.56 | 1.64 | 1.06 | **0.83** | **0.84** | **0.85** | 0.50 |
| RT | 1.07 | 1.04 | 1.02 | 1.01 | 94.8 | 86.9 | 79.8 | 77.1 | 1.24 | 1.34 | 1.36 | 1.49 | 0.77 | 0.79 | 0.81 | 0.82 |
| ST-GS | 3.38 | 3.36 | 3.41 | 3.45 | 310 | 316 | 328 | 347 | 1.08 | 1.06 | 1.04 | 1.05 | 0.45 | 0.41 | 0.43 | 0.42 |
| NSVQ | 1.06 | 1.05 | 1.04 | 1.02 | 112 | 108 | 103 | 95.6 | 1.36 | 1.45 | 1.54 | 1.62 | 0.81 | 0.81 | 0.83 | 0.84 |
| DiVeQ (ours) | 1.04 | 1.01 | **0.99** | **0.98** | 80.6 | 75.4 | **71.0** | 69.4 | 1.50 | **1.57** | **1.68** | **1.76** | 0.82 | 0.83 | **0.85** | **0.85** |
| SF-DiVeQ (ours) | 1.01 | 1.00 | 1.00 | 0.99 | 76.4 | 74.3 | 73.9 | 73.2 | **1.56** | 1.50 | 1.58 | 1.65 | **0.83** | 0.83 | 0.84 | 0.84 |

## C.4 ABLATION ON LEARNING RATE

In this section, we evaluate the performance of different VQ optimization techniques in the VQ-VAE compression task on the CELEBA-HQ data set using different learning rates. The ablation is performed over two new learning rates of $lr = \{10^{-3}, 10^{-4}\}$ while training the models for 100 epochs with the batch size of 32. During training, the learning rates are halved after 40 and 70 epochs. Fig. 18 and Fig. 19 show the quality of reconstructions when $lr = 10^{-3}$ and $lr = 10^{-4}$, respectively. Note that in the plots, the reported value for each metric is the average of that metric over all test set reconstructions and over three different individual runs. According to Fig. 18, DiVeQ and SF-DiVeQ outperform other methods, achieving higher SSIM, PSNR values, and obtaining lower LPIPS values. According to Huh et al. (2023), when increasing the learning rate in VQ-VAEs, the model is highly prone to misaligned codebook and latent representations. Therefore, in Fig. 18, we find that some misalignments are happened for [ST-GS|codebook bits=6,8,9,10,11], [EMA|codebook bits=7], and [DiVeQ|codebook bits=11]. The misalignments are visible as sudden jumps in the objective metrics, and happen in cases where the increase in codebook size does not lead to enhancement in the objective metrics (more details in Fig. 4, Sec. 4, and App. C.8).

In Fig. 19, again, our proposed methods obtain better objective metrics than other methods, while performing comparable to EMA. The reason is that EMA updates the codebook without gradient descent, and thus, there is no explicit learning rate defined for its codebook learning. The decay rate is the hyperparameter responsible for the speed of the codebook updates. As we do not have a proper definition on how to set the decay rate corresponding to a specific learning rate, we use the fixed decay rate of $\gamma = 0.99$ for EMA in all experiments in the ablation study on learning rates (*i.e.*, Figs. 18 and 19). According to Fig. 19, a misalignment is happened for [RT|codebook bits=6].

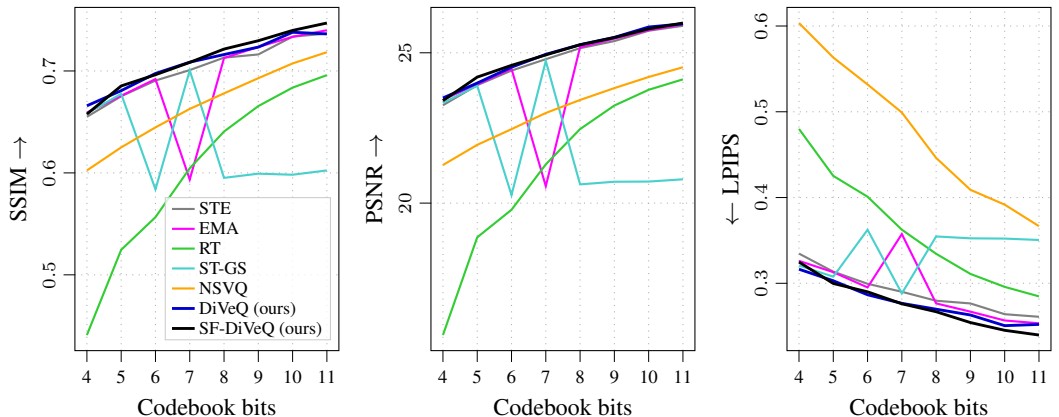

Figure 18: **Ablation on learning rate: The proposed DiVeQ and SF-DiVeQ lead to consistent improvement in image reconstruction quality over different learning rates (this figure, $lr = 10^{-3}$).** Quantitative comparison of reconstructed images from the CELEBA-HQ test set for different VQ optimization methods over different codebook sizes. Each curve is the average of the metric over all test set reconstructions and over three different individual runs.

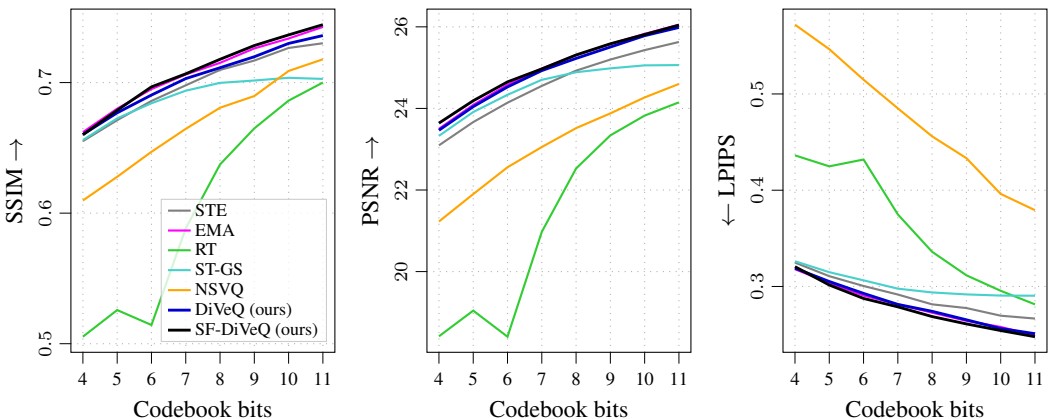

Figure 19: **Ablation on learning rate: The proposed DiVeQ and SF-DiVeQ lead to consistent improvement in image reconstruction quality over different learning rates (this figure, $lr = 10^{-4}$).** Quantitative comparison of reconstructed images from the CELEBA-HQ test set for different VQ optimization methods over different codebook sizes. Each curve is the average of the metric over all test set reconstructions and over three different individual runs.

## C.5 ABLATION ON VARIANCE $\sigma^2$ OF DIVEQ AND SF-DIVEQ

To assess the impact of variance $\sigma^2$ in our proposed DiVeQ and SF-DiVeQ, we did ablation studies using different variances in both VQ-VAE compression and VQGAN generation tasks. This section aims to show that the performance of our proposed DiVeQ and SF-DiVeQ is largely insensitive to the selection of the variance $\sigma^2$. In other words, our methods do not require tuning the variance, and using a small-enough variance ($\sigma^2 \leq 10^{-2}$) is the only requirement to ensure DiVeQ and SF-DiVeQ perform better than the other VQ optimization methods.

In the VQ-VAE compression task, we did an ablation study with the CELEBA-HQ data set using different variances of $\sigma^2 = \{10^{-1}, 10^{-2}, 10^{-3}, 10^{-4}\}$. Fig. 20 and Fig. 21 show the quality of reconstructions over different variances for the DiVeQ and SF-DiVeQ techniques, respectively. Note that in the plots, the reported value for each metric is the average of that metric over all test set reconstructions and over three different individual runs. Conforming with what we presented in Fig. 3, results in Figs. 20 and 21 confirm that to achieve higher reconstruction quality, the variance should be small to have more precise nearest-codeword mappings of inputs $z$ to the selected codewords $c_{i^*}$. That is why in Figs. 20 and 21, when $\sigma^2 = 10^{-1}$ (which is still considered a high value for the variance), the reconstructions are of low quality.

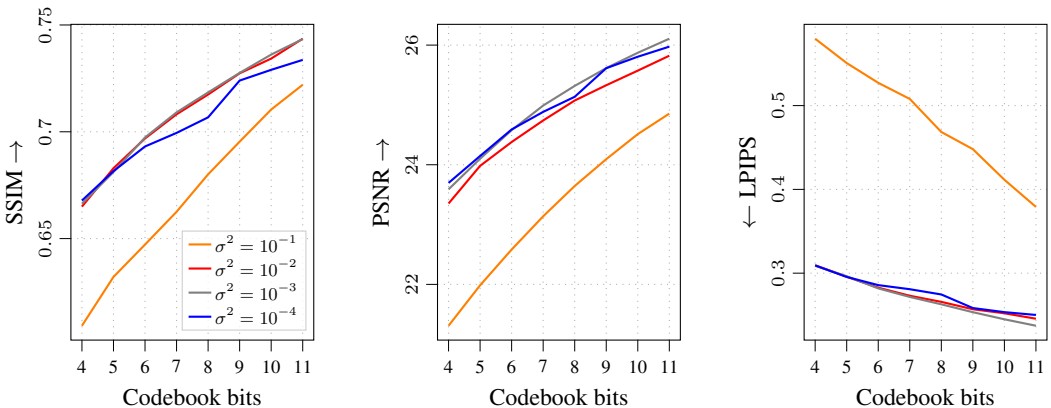

Figure 20: **Ablation on variance $\sigma^2$ for DiVeQ: Quality of image reconstruction is improved when reducing the variance $\sigma^2$.** Quantitative comparison of reconstructed images from the CELEBA-HQ test set for DiVeQ optimization technique over different variances. Each curve is the average of the metric over all test set reconstructions and over three different individual runs.

On the other hand, we observe comparable quality for the reconstructions when $\sigma^2 = \{10^{-2}, 10^{-3}, 10^{-4}\}$. Based on the results, we choose $\sigma^2 = 10^{-3}$ for the VQ-VAE compression task. Note that we choose the variance value only based on the results obtained from the CELEBA-HQ data set, and we blindly use this value (*i.e.*, $\sigma^2 = 10^{-3}$) for other data sets where our proposed techniques outperform other VQ methods (see App. C.1). From a different perspective, we can think of $\sigma^2$ as a hyperparameter that marginally controls the amount of generalization for the VQ nearest-codeword assignments, but does not need a proper tuning as long as $\sigma^2 \leq 10^{-2}$. The larger the variance $\sigma^2$ is, the more generalization will be injected in nearest-codeword assignments.

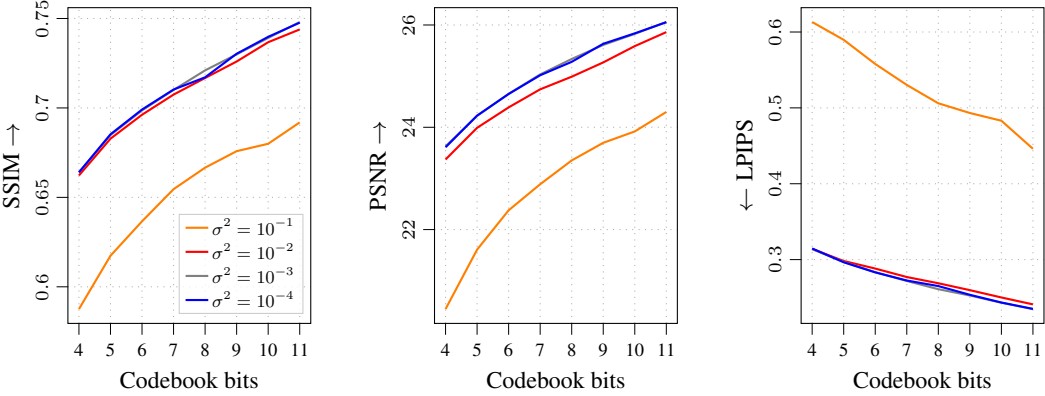

Figure 21: **Ablation on variance $\sigma^2$ for SF-DiVeQ: Quality of image reconstruction is improved when reducing the variance $\sigma^2$.** Quantitative comparison of reconstructed images from the CELEBA-HQ test set for SF-DiVeQ optimization technique over different variances. Each curve is the average of the metric over all test set reconstructions and over three different individual runs.

In addition, we did another ablation study in the VQGAN generation task with the FFHQ, LSUN Bedroom, and LSUN Church data sets using different variances of $\sigma^2 = \{10^{-2}, 10^{-3}, 10^{-4}\}$. Table 10 shows the obtained FID values over different variances for DiVeQ and SF-DiVeQ techniques. Following the results in the table, for a specific codebook size, the performance of both DiVeQ and SF-DiVeQ remains almost similar over different variances. This shows that the performance of DiVeQ and SF-DiVeQ is not sensitive to the variance $\sigma^2$.

Table 10: **Ablation on variance $\sigma^2$ for DiVeQ and SF-DiVeQ: Quality of generations remains quite the same over different variances.** The table presents the FID↓ scores for the proposed DiVeQ and SF-DiVeQ over different variances for FFHQ, LSUN Bedroom, and Church data sets.

| Data set | Approach | Variance $\sigma^2$ | Codebook bits | | | |
|---|---|---|---|---|---|---|
| | | | 8 | 9 | 10 | 12 |
| FFHQ | DiVeQ (ours) | $10^{-2}$ | 6.70 | 5.61 | 5.11 | 8.22 |
| | | $10^{-3}$ | 7.12 | 6.22 | 5.37 | 8.47 |
| | | $10^{-4}$ | 7.32 | 5.86 | 5.45 | 8.59 |
| | SF-DiVeQ (ours) | $10^{-2}$ | 7.91 | 6.21 | 5.93 | 8.60 |
| | | $10^{-3}$ | 6.91 | 7.16 | 5.72 | 7.81 |
| | | $10^{-4}$ | 7.01 | 6.24 | 5.69 | 7.66 |
| LSUN Bedroom | DiVeQ (ours) | $10^{-2}$ | 5.71 | 4.87 | 5.40 | 7.94 |
| | | $10^{-3}$ | 5.53 | 5.39 | 5.83 | 8.32 |
| | | $10^{-4}$ | 5.60 | 5.81 | 5.42 | 7.69 |
| | SF-DiVeQ (ours) | $10^{-2}$ | 5.96 | 6.01 | 5.40 | 7.79 |
| | | $10^{-3}$ | 5.79 | 5.28 | 5.21 | 7.02 |
| | | $10^{-4}$ | 5.75 | 5.34 | 4.91 | 6.55 |
| LSUN Church | DiVeQ (ours) | $10^{-2}$ | 4.41 | 3.92 | 3.90 | 4.58 |
| | | $10^{-3}$ | 4.64 | 4.36 | 4.04 | 5.85 |
| | | $10^{-4}$ | 4.37 | 4.14 | 4.29 | 6.48 |
| | SF-DiVeQ (ours) | $10^{-2}$ | 4.35 | 3.99 | 3.56 | 4.40 |
| | | $10^{-3}$ | 4.32 | 4.21 | 3.72 | 3.96 |
| | | $10^{-4}$ | 4.46 | 4.23 | 3.26 | 3.75 |

## C.6 ABLATION ON CODEBOOK REPLACEMENT

In all experiments in the paper, we use codebook replacement (App. B.1) for all VQ methods when optimizing the codebook (except for SF-DiVeQ, which does not need codebook replacement). This raises the questions that *(i)* how much codebook replacement contributes to the overall performance for all different VQ methods, *(ii)* whether codebook replacement is more in favor of our proposed DiVeQ than the other methods, and *(iii)* whether codebook replacement actually improves the performance when training VQ codebook. In this section, we provide a set of experiments that show that by skipping the codebook replacement, our proposed DiVeQ still obtains higher quality reconstructions and faster convergence than other VQ optimization methods.

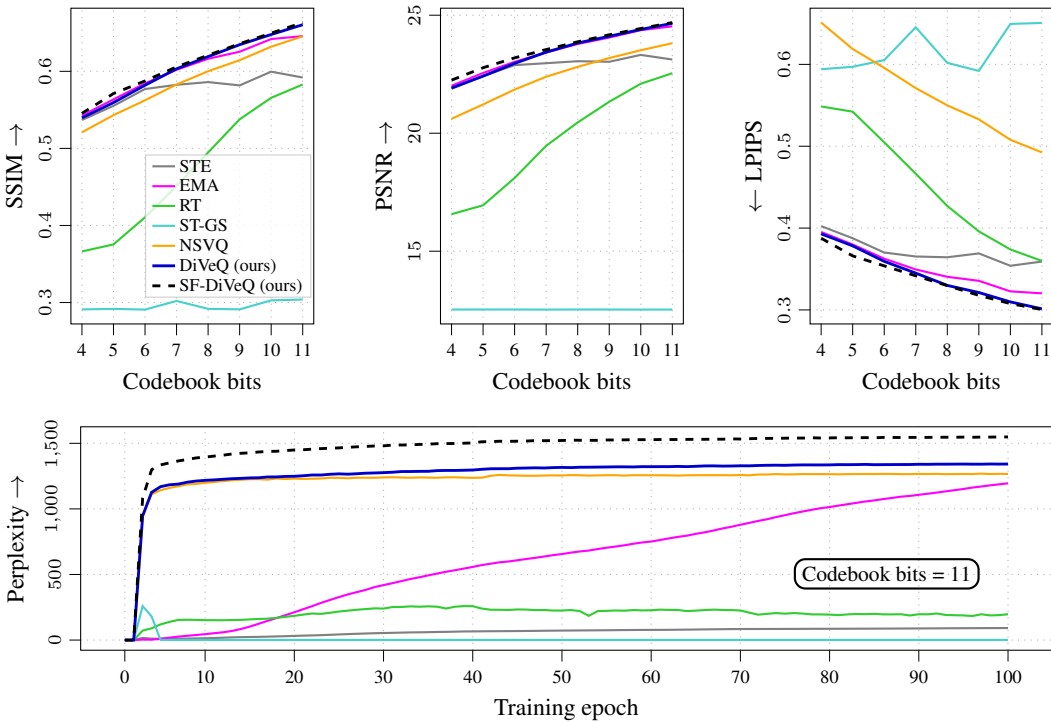

Figure 22: **Ablation on codebook replacement: By skipping codebook replacement, the proposed DiVeQ still leads to consistent improvement in image reconstruction quality and achieves the highest perplexity.** (top) Quantitative comparison of reconstructed images from the AFHQ test set for different VQ optimization methods over different codebook sizes. Each curve is the average of the metric over all test set reconstructions and over three different individual runs. (bottom) Perplexity (or average codebook usage) during training for one individual run with an 11-bit codebook.

In the VQ-VAE compression task, we trained all VQ optimization methods by skipping codebook replacement. All models are trained with a batch size of 32 at a learning rate of $lr = 5.5 \cdot 10^{-4}$ on the AFHQ data set for 100 epochs. Fig. 22 shows the quality of reconstructions along with the perplexity (Eq. (15)) tracked over training epochs for different VQ methods while the codebook replacement is skipped. Note that in the plots, the reported value for each metric is the average of that metric over all test set reconstructions and over three different individual runs. Based on the results in the figure, even if we skip the codebook replacement, our proposed DiVeQ achieves higher quality reconstructions than other VQ methods, while the performance gap increases with the increase in codebook size.

Comparison of the results in Fig. 22 (without codebook replacement) with Fig. 6 (with codebook replacement) reveals that the performance of STE, RT, and especially ST-GS becomes worse without applying codebook replacement, while the performance remains quite the same for EMA, NSVQ, and our proposed DiVeQ. In the perplexity plot in Fig. 22, our proposed DiVeQ achieves the high-

est perplexity among other VQ methods (except SF-DiVeQ) in the early stages of training, and consistently keeps the perplexity high until the end of training. However, it takes the whole 100 training epochs for the EMA approach to reach a comparable perplexity as our proposed DiVeQ, which means that DiVeQ converges faster than EMA. Comparing EMA perplexity in Fig. 22 and Fig. 29 shows that codebook replacement helps EMA to converge faster, such that it reaches similar perplexity as our DiVeQ at epoch 45. In Fig. 23, we provide the quality of reconstructions over different training epochs for the individual experiment for which the perplexity is shown in Fig. 22 with an 11-bit codebook. Fig. 23 demonstrates that by initializing VQ with a well-fitted codebook and by skipping the codebook replacement, our proposed DiVeQ can converge faster than other VQ methods.

In Fig. 22, we also plotted the results for the SF-DiVeQ method by copying them directly from Fig. 6 with the aim of having them as a baseline for further comparisons. As discussed in Sec. 4, to initialize the SF-DiVeQ codebook, the quantization is skipped for the first two epochs, and then the codebook is initialized by the average of the latest latent vectors at the end of the second epoch. Note that for the experiments of this section (that skip the codebook replacement), we initialize the codebooks of all VQ methods in a similar way as the SF-DiVeQ approach because of two reasons. First, we can quantitatively compare all VQ methods with SF-DiVeQ when all methods have similar initialization, and none of them use codebook replacement (SF-DiVeQ does not need codebook replacement at all). Second, to evaluate how different VQ methods optimize the codebook and how good their gradients are for updating the codebook, a proper experiment is to initialize all of them with a codebook that is well-fitted to the current latent distribution. In this way, all codewords should be active when VQ is started, and then afterwards, everything will be dependent on how each technique treats this well-fitted codebook. The metrics and perplexity in Fig. 22 show that our proposed DiVeQ outperforms other methods by introducing well-defined gradients for the codebook and keeping the perplexity (or codebook usage) high throughout the training.

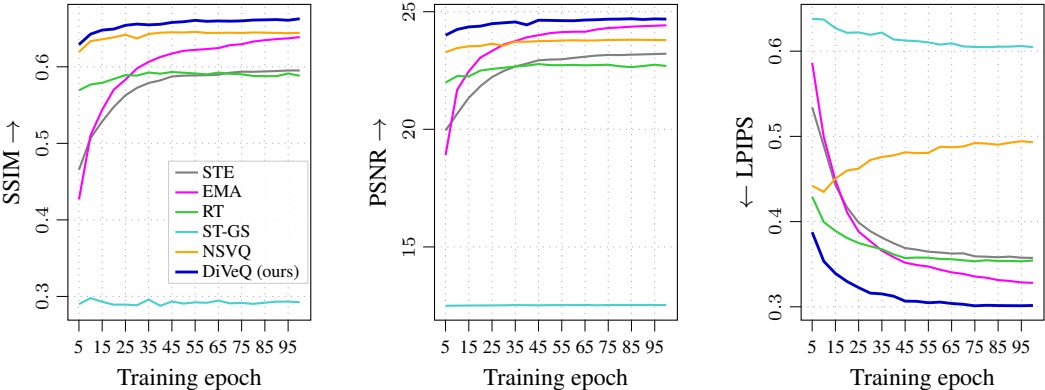

Figure 23: **Convergence speed: By skipping codebook replacement, the proposed DiVeQ converges faster than other VQ methods by reaching high image reconstruction qualities in the early stages of training.** Quantitative comparison of reconstructed images from the AFHQ test set for different VQ optimization methods over different training epochs. The curves are for the individual run shown as the perplexity plot in Fig. 22, with an 11-bit codebook.

## C.7 ABLATION ON SF-DIVEQ INITIALIZATION

In all experiments in the paper, we initialize SF-DiVeQ with a *custom* initialization in which the quantization is skipped for the first two epochs, and then SF-DiVeQ starts quantizing the latent space such that its codebook is initialized with the average of latent vectors obtained from the latest 20–50 training batches. This raises the question that *(i)* how much this *custom* initialization contributes to the overall performance of SF-DiVeQ, *(ii)* how other VQ methods perform with similar *custom* initialization (see the answer in Fig. 22), and *(iii)* whether SF-DiVeQ is still capable of pulling inactive codewords inside the latent distribution using a *random* initialization. In this section, we provide a set of experiments showing that SF-DiVeQ with *random* initialization can reach similar reconstruction quality and perplexity compared to the *custom* initialization.

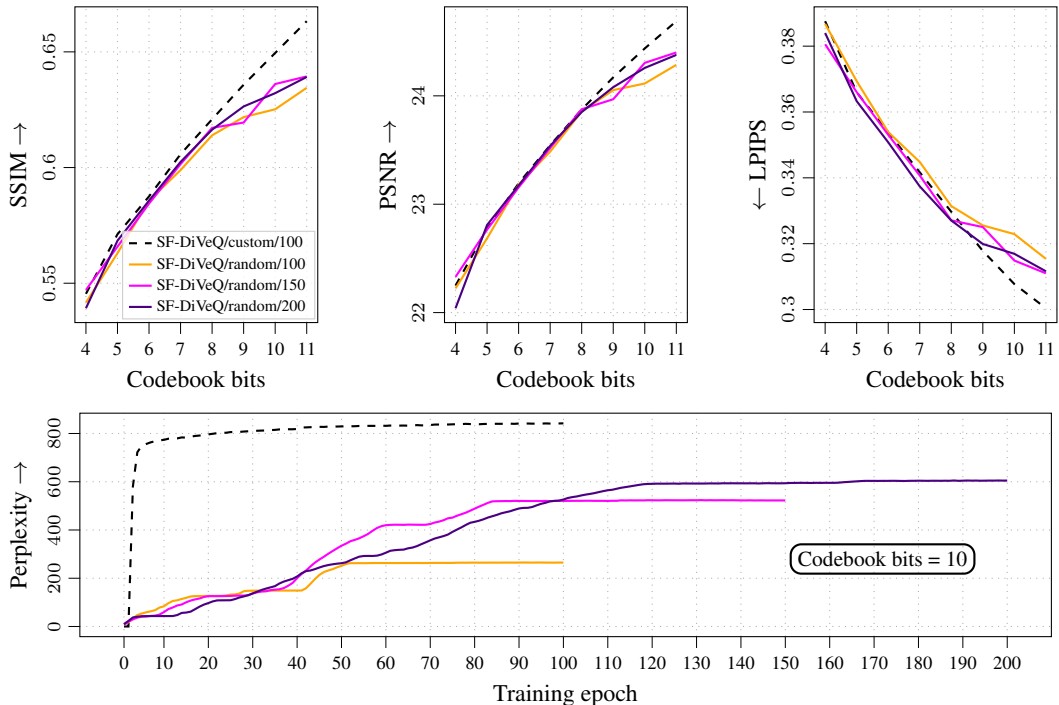

Figure 24: **Ablation on SF-DiVeQ initialization: By using *random* initializations, the proposed SF-DiVeQ still performs comparable to *custom* initialization for small and moderate codebook sizes, while performing worse with the increase in the codebook size.** (top) Quantitative comparison of reconstructed images from the AFHQ test set for SF-DiVeQ, such that it uses different initializations (*custom* and *random*) and is trained for different epochs ($\{100, 150, 200\}$). Each curve is the average of the metric over all test set reconstructions and over three different individual runs. (bottom) Perplexity (or average codebook usage) during training for one individual run with a 10-bit codebook.

In the VQ-VAE compression task, we train SF-DiVeQ by *random* initialization for different numbers of training epochs ($\{100, 150, 200\}$). All experiments are trained with a batch size of 32 at the initial learning rate of $lr = 5.5 \cdot 10^{-4}$ on the AFHQ data set, such that the learning rate is halved after 50% and 75% of training epochs. Fig. 24 shows the quality of reconstructions along with the perplexity (Eq. (15)) tracked over training epochs for different SF-DiVeQ experiments. Note that in the plots, the reported value for each metric is the average of that metric over all test set reconstructions and over three different individual runs. We also provided the SF-DiVeQ results with *custom* initialization (captured from Fig. 6) to have them as a baseline for comparisons.

According to Fig. 24, our proposed SF-DiVeQ with *random* initializations still performs comparable to the case with *custom* initialization for small and moderate codebook sizes. However, SF-DiVeQ performance degrades with the increase in codebook size, and the reason stems from the perplexity plot, which shows the average codebook usage for one individual experiment with a 10-bit codebook. The perplexity values reveal three important facts; *(i)* in *random* initializations, the perplexity is regularly increased during the training, which confirms the fact that SF-DiVeQ consistently pulls inactive codewords inside the latent distribution, *(ii)* for big codebook sizes, the *custom* initialization helps SF-DiVeQ to perform better such that SF-DiVeQ starts with a high codebook usage from the beginning of training while keeping most (or all of (see Fig. 29)) the codewords active until the end of training. Whereas, this does not happen for the cases of *random* initialization, and *(iii)* in *random* initializations, the more SF-DiVeQ is trained, the higher perplexity it reaches, and thus, the better performance it achieves. The reason is that for longer training epochs, SF-DiVeQ has more time to pull more inactive codewords inside the latent distribution.

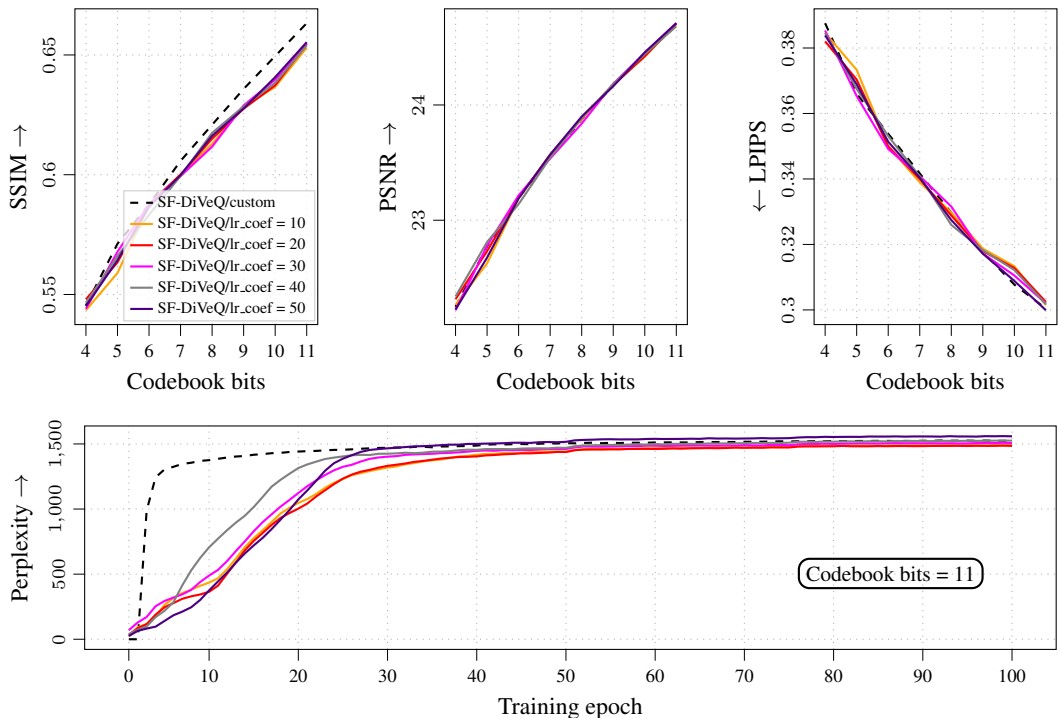

Figure 25: **Ablation on SF-DiVeQ initialization: By using learning rates that are multiple times higher than other model parameters, SF-DiVeQ with *random* initialization can reach similar reconstruction quality and perplexity as SF-DiVeQ with *custom* initialization.** (top) Quantitative comparison of reconstructed images from the AFHQ test set for SF-DiVeQ while its codebook learning rate is lr_coef times higher than other model parameters. Each curve is the average of the metric over all test set reconstructions and over three different individual runs. (bottom) Perplexity (or average codebook usage) during training for one individual run with an 11-bit codebook.

According to the perplexity curves for *random* initializations in Fig. 24, we notice that the perplexity is not increased anymore when SF-DiVeQ training reaches its halfway point. We hypothesize the reason could be that the learning rate is halved after 50% and 75% of training epochs, and a lower learning rate results in smaller updates in the codebook, and hence, the remaining inactive codewords will not be pulled inside the latent distribution anymore. Therefore, the codebook learning rate could have a big effect on the performance of SF-DiVeQ with *random* initialization. Motivated by this hypothesis, we train SF-DiVeQ with *random* initializations using different learning rates for the codebook. Fig. 25 shows the quality of reconstructions along with the perplexity tracked over training epochs when the learning rate of the codebook is higher than the model learning rate by different coefficients (*i.e.*, lr_coef = $\{10, 20, 30, 40, 50\}$). Note that in the plots, the reported value for each metric is the average of that metric over all test set reconstructions and over three different individual runs. We also provided the SF-DiVeQ results with *custom* initialization (captured from Fig. 6) to have it as a baseline for comparisons, which is shown as SF-DiVeQ/custom.

Fig. 25 demonstrates that even by using *random* initializations, SF-DiVeQ can achieve similar performance to the case of *custom* initialization for all different codebook sizes, if we adopt a higher learning rate for the codebook. The perplexity plots in Fig. 25 substantiate our hypothesis that a higher learning rate for the codebook encourages SF-DiVeQ to pull the inactive codewords faster inside the latent space, and thus, quickly reach the same (or even higher) perplexity as in *custom* initialization (*i.e.*, SF-DiVeQ/custom). Furthermore, the higher the learning rate is, the more perplexity SF-DiVeQ reaches. Note that for the experiments of Fig. 25, we still apply the learning rate scheduling by halving the learning rate of all parameters after 50% and 75% of training epochs.

### C.8 MISALIGNMENT OF CODEBOOK AND LATENT REPRESENTATIONS

As discussed in Sec. 4, when training the VQ-VAE for the compression task using different learning rates and batch sizes, there would be cases in which Shannon's rate-distortion theory (Shannon, 1959) does not hold. In other words, in these cases, when the codebook size is increased, the quality of reconstructions will not be improved with respect to the objective metrics. In Fig. 26, we provide the quantitative results for the cases where the rate-distortion theory does not hold, and as a result, a misalignment between codebook and latent representations occurs (see Fig. 4). The curves represent the results for VQ-VAE trained on CELEBA-HQ data set with various training setups.

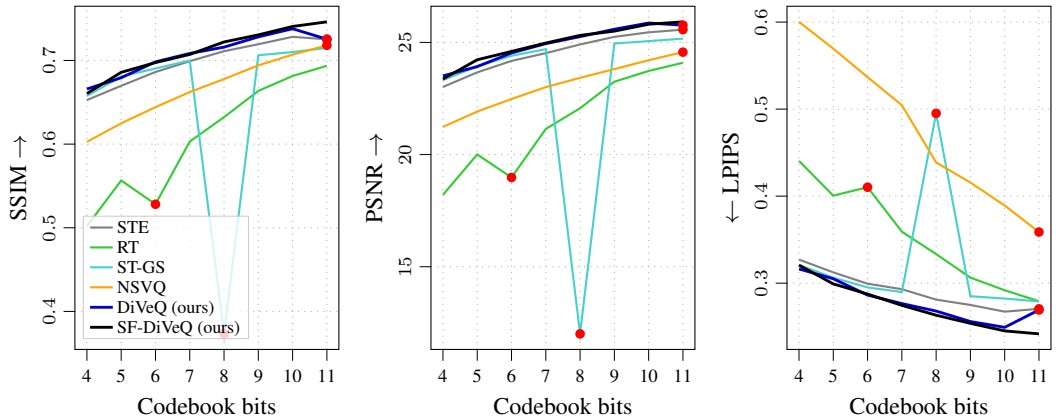

Figure 26: **Misalignment of codebook and latent representations happens where the increase in codebook size does not improve the objective metrics.** The figure shows the quantitative results in VQ-VAE compression experiments on CELEBA-HQ data set for different VQ methods. Cases where misalignments happen are highlighted in *red circles*. Each curve shows the results only for one individual run. For the EMA approach, we consider the misalignment that happened in Fig. 17 for the 11-bit codebook.

Misalignments are highlighted with *red circles* in Fig. 26, which refer to the cases of [STE|$lr = 10^{-4}$, codebook bits=11], [RT|$lr = 5.5 \cdot 10^{-4}$, codebook bits=6], [ST-GS|$lr = 10^{-3}$, codebook bits=8], [NSVQ|$lr = 10^{-3}$, codebook bits=11], and [DiVeQ|$lr = 10^{-3}$, codebook bits=11]. Note that for the EMA approach, we consider the misalignment that happened in Fig. 17, when the batch size is 128 and $lr = 5.5 \cdot 10^{-4}$. According to all of our experiments, for our proposed SF-DiVeQ, the rate-distortion theory always holds true and as a result, no misalignment happens for SF-DiVeQ. For the sake of comparison, we consider the case of [SF-DiVeQ|$lr = 10^{-3}$, codebook bits=11] for plotting in Fig. 4. Note that unlike other plots in the paper, Fig. 26 shows the results for only one individual experiment, not the average on several different individual runs.

The reported values in Fig. 4 show the distortion per bit ($D_{\text{per-bit}} \downarrow$) for each of the quantization cases. Since the misalignments of different VQ methods happen in different codebook sizes, it is not fair to report the ordinary distortion for each of the quantization cases. Because smaller codebook sizes result in higher amounts of distortion. Therefore, for quantitative evaluation of quantization cases in Fig. 4, we measure the $D_{\text{per-bit}}$ metric that measures how much distortion remains per bit. $D_{\text{per-bit}}$ is computed as

$$D_{\text{per-bit}} = \frac{D}{H(\mathcal{C})} \quad \text{s.t.} \quad D = \frac{1}{N}\sum_{n=1}^{N}\|\boldsymbol{z}_n - \hat{\boldsymbol{z}}_n\|_2^2 \quad \text{and} \quad H(\mathcal{C}) = -\sum_{k=1}^{K} p_k \cdot \log_2 p_k, \quad (17)$$

where $D$ is the quantization distortion, $H(\mathcal{C})$ is entropy of the codebook index distribution, and $p_k$ refers to the empirical usage probability of the $k$-th codebook vector. Here, $N$ and $K$ refer to the number of distribution samples and the number of codewords, respectively. $D_{\text{per-bit}}$ is a suitable and fair metric to use to compare the case of misalignments, as it reflects the distortion ($D$) mixed with the effective codebook usage ($H(\mathcal{C})$).

## C.9 DiVeQ and SF-DiVeQ in Residual Vector Quantization

As mentioned in Sec. 1 of the paper, DiVeQ and SF-DiVeQ can also be used to train other variants of vector quantization like Residual VQ (RVQ, Chen et al., 2010; Vali & Bäckström, 2023b). Different from ordinary vector quantization, RVQ uses multiple codebooks to quantize a distribution. Suppose an RVQ with three codebooks of $\{\mathcal{C}^1, \mathcal{C}^2, \mathcal{C}^3\}$. The first stage of RVQ quantizes the input $z$ to $\hat{z}_1$ using the first codebook $\mathcal{C}^1$

$$\hat{z}_1 = c_{i*}^1 = \arg\min_{c_j^1} \|z - c_j^1\|_2 \quad \text{s.t.} \quad c_j^1 \text{ is the } j\text{-th codeword of } \mathcal{C}^1, \tag{18}$$

and computes the residual of the first stage as $r_1 = z - \hat{z}_1$. Similarly in the second stage, RVQ takes $r_1$ as input and quantizes it to $\hat{r}_1$ using the second codebook $\mathcal{C}^2$

$$\hat{r}_1 = c_{i*}^2 = \arg\min_{c_j^2} \|r_1 - c_j^2\|_2 \quad \text{s.t.} \quad c_j^2 \text{ is the } j\text{-th codeword of } \mathcal{C}^2, \tag{19}$$

and computes the residual of the second stage as $r_2 = r_1 - \hat{r}_1$. Then, for the third (or last) stage, RVQ takes $r_2$ as input and quantizes it to $\hat{r}_2$ using the third codebook $\mathcal{C}^3$

$$\hat{r}_2 = c_{i*}^3 = \arg\min_{c_j^3} \|r_2 - c_j^3\|_2 \quad \text{s.t.} \quad c_j^3 \text{ is the } j\text{-th codeword of } \mathcal{C}^3. \tag{20}$$

Finally, RVQ computes the quantized input as

$$\hat{z} = c_{i*}^1 + c_{i*}^2 + c_{i*}^3, \tag{21}$$

where $\hat{z}$ is the final hard quantized version of the input $z$.

In this section, we compare the performance of different VQ optimization techniques in the VQ-VAE compression task when quantizing the latent space by RVQ. We train the VQ-VAE models over 100 epochs with a batch size of 32 and an initial learning rate of $lr = 5.5 \cdot 10^{-4}$ that is halved after 40 and 70 epochs. In all experiments, we apply RVQ using three different codebooks over four different codebook bits $B = \{12, 18, 24, 30\}$. For instance, for quantizing with 30 bits, we apply RVQ with three quantization stages, each with a 10-bit codebook (or codebook size of 1024).

Fig. 27 and Fig. 28 show the quantitative comparison of different VQ optimization techniques when reconstructing the test set images of CELEBA-HQ and AFHQ, respectively. Note that in the plots, the reported value for each metric is the average of that metric over all test set reconstructions and over three different individual runs. According to the figures, our proposed DiVeQ and SF-DiVeQ consistently obtain higher SSIM, PSNR, and lower LPIPS values than the other methods over different codebook sizes (except for the SSIM of NSVQ in Fig. 28). Furthermore, when increasing the codebook size, the performance gap between our proposed methods and STE is escalated. On the other hand, similar to the result for ordinary VQ, the RT performs poorly for low RVQ codebook sizes, and its performance improves with the increase in codebook size.

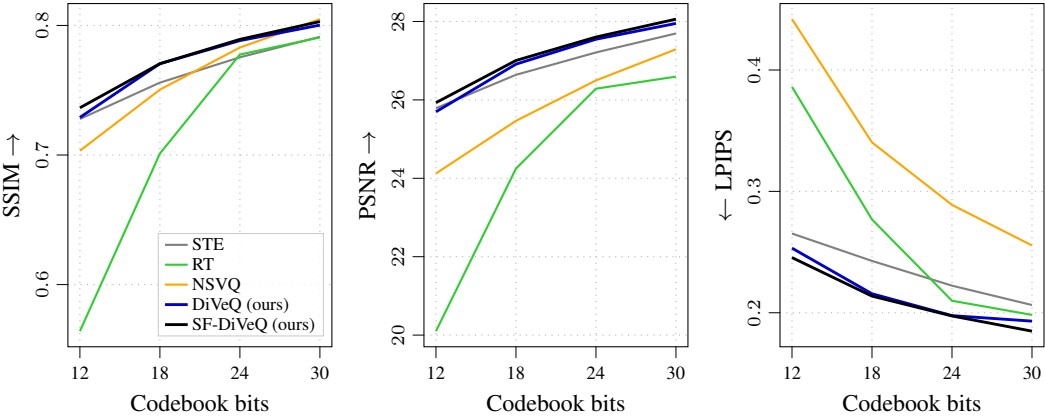

Figure 27: **Quantizing the latent space by Residual VQ: The proposed DiVeQ and SF-DiVeQ lead to consistent improvement in image reconstruction quality.** Quantitative comparison of reconstructed images from the CELEBA-HQ test set for different VQ optimization methods over different codebook sizes. Each curve is the average of the metric over all test set reconstructions and over three different individual runs.

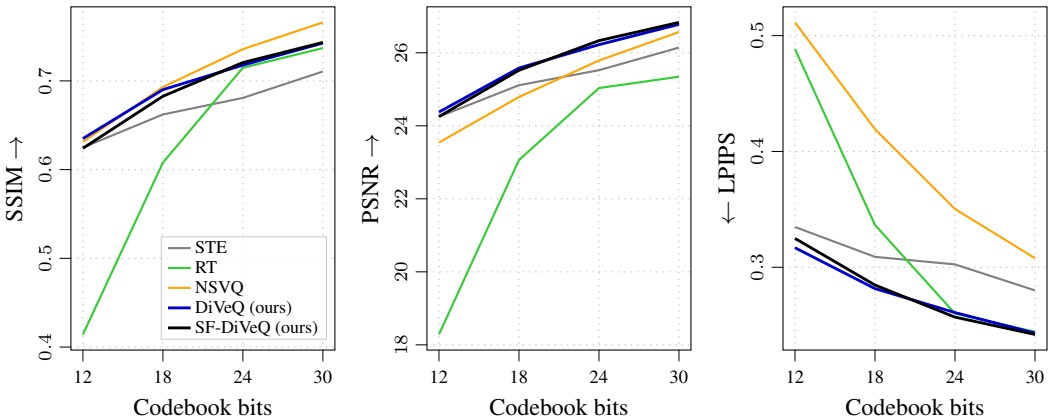

Figure 28: **Quantizing the latent space by Residual VQ: The proposed DiVeQ and SF-DiVeQ lead to consistent improvement in image reconstruction quality.** Quantitative comparison of reconstructed images from the AFHQ test set for different VQ optimization methods over different codebook sizes. Each curve is the average of the metric over all test set reconstructions and over three different individual runs.

### C.10    Training logs

In this section, we study the training logs when training the VQ-VAE for the image compression task on the AFHQ data set. Fig. 29 shows the codebook usage, perplexity (Eq. (15)), and reconstruction and perceptual losses over the course of training for different VQ optimization methods with an 11-bit codebook (or codebook size of $K = 2048$). To make the plots look smoother, the curves are smoothed via Blackman windowing with a window size of $15$ (except for the codebook usage plot). The codebook usage represents the percentage of codebook used in each training epoch. Apart from the ST-GS method, all other VQ optimization techniques reach the full codebook usage (but it happens a bit late for the EMA approach). Note that the codebook usage is reported when codebook replacement (App. B.1) is actively applied during training for all VQ methods. Even though codebook replacement is applied, it cannot compensate for full codebook usage for the ST-GS method.

Perplexity refers to the number of unique codewords selected for quantizing a typical training batch, and it ranges from 1 to codebook size $K$. Eq. (15) explains how perplexity is computed. According to Fig. 29, the highest perplexities belong to STE, EMA, and our proposed DiVeQ and SF-DiVeQ, while NSVQ reaches a bit lower perplexity compared to these methods. However, for ST-GS and RT approaches, the perplexity is very high at the start of training, and it suddenly drops for ST-GS, whereas it smoothly decays for RT during training. Reconstruction loss is the MSE loss between the input image $x$ and its reconstructed output $x_r$ (see Fig. 1 and Eq. (1)), and perceptual loss is the LPIPS between $x$ and $x_r$. According to the losses, our proposed DiVeQ and SF-DiVeQ reach lower loss values than STE, EMA, RT, and ST-GS while converging faster than STE, EMA, and ST-GS. In contrast to all methods, NSVQ reaches the lowest loss values with a big margin, especially for the perceptual loss. To make the difference between all VQ methods more clear, we plot the loss values on a logarithmic scale.

When training VQ in neural networks, computing the codebook usage once for the whole course of training is not enough to verify that codebook optimization is proceeding correctly. Because the latent representation $\mathcal{P}_z$ is dynamically changing, and as a result, one used codeword for the initial training iterations might become useless for the rest of the training by falling out of the latent distribution. Codebook replacement helps to keep the codebook usage at its maximum during training by actively reinitializing unused codewords. However, even a 100% codebook usage during the whole training does not guarantee a properly learned codebook. Because it is possible that all codewords are located inside the latent representation $\mathcal{P}_z$, but they are not necessarily well-fitted to $\mathcal{P}_z$. For instance, in Fig. 29, RT and NSVQ reach 100% codebook usage, but yield poor qualitative (Fig. 5) and quantitative (Fig. 6) results, and they are both prone to misalignment of codebook and latent representations (see Fig. 4).

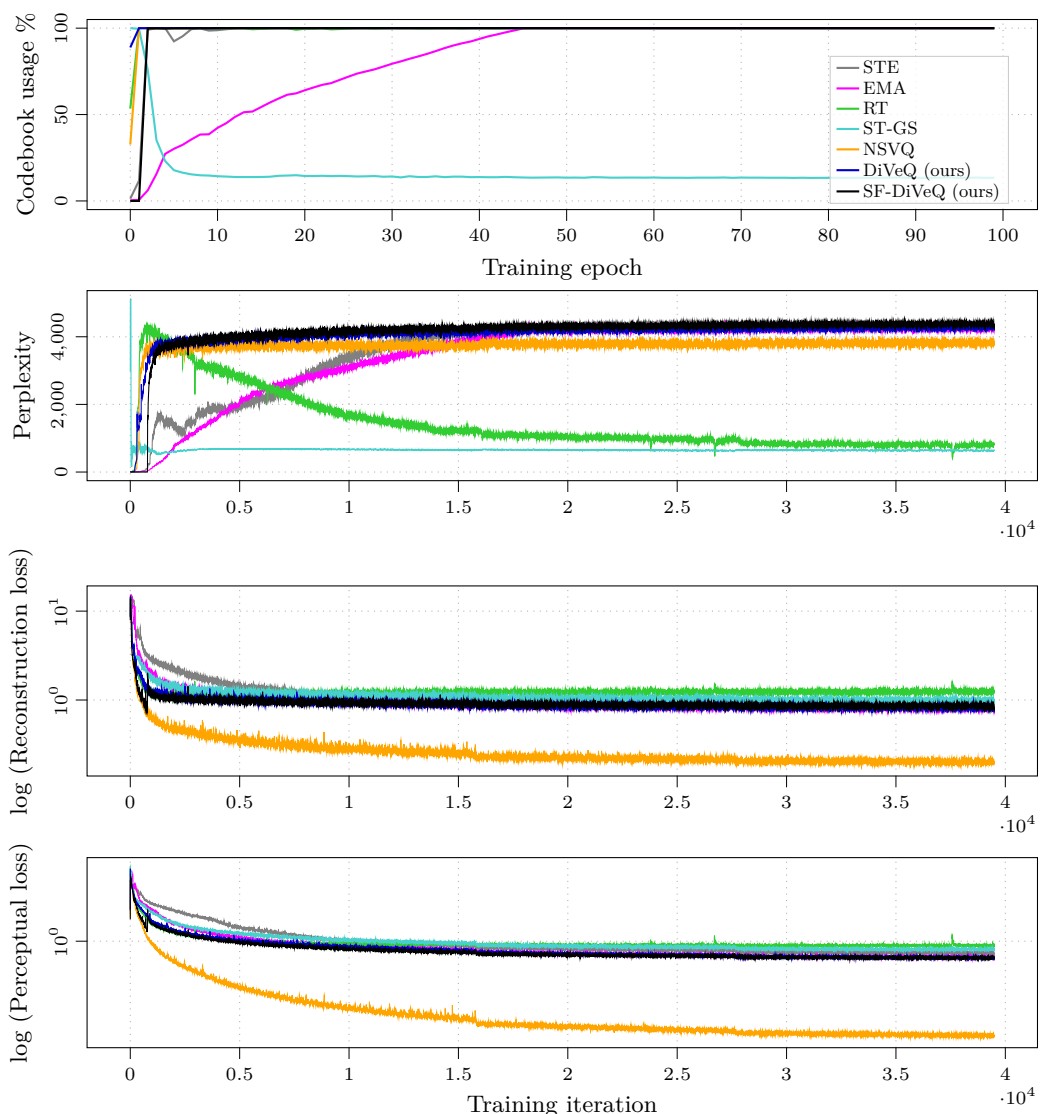

Figure 29: **Training logs do not necessarily reflect whether VQ codebook optimization is being carried out appropriately.** Comparison of the codebook usage, perplexity, reconstruction, and perceptual losses for different VQ optimization methods, when training the VQ-VAE compression model with an 11-bit codebook on the AFHQ train set.

Similar to codebook usage and perplexity factors, reconstruction and perceptual losses cannot be suitable metrics to reflect whether a VQ optimization method performs better than the other. For instance, in Fig. 29, NSVQ reaches a much higher perplexity than ST-GS and obtains the lowest loss values among the others, but it leads to the worst qualitative (Fig. 5) and quantitative (Fig. 6) results over different data sets compared to all other approaches. Therefore, these metrics are useful only to make sure the training process runs as expected.

## C.11 MORE SAMPLES GENERATED BY VQGAN

Apart from the quantitative evaluation of VQGAN generations over different VQ optimization methods using the FID metric (Tables 2 and 7), it is important to compare the generations qualitatively as well. Figs. 30 to 34 provide completely random generations from VQGAN for all five data sets with a 9-bit codebook (or codebook size of $K = 512$).

## D DISCLOSURE OF THE USE OF LARGE LANGUAGE MODELS

In this paper, LLMs were used only for minor grammatical edits, word polishing, or rephrasing. They did not contribute to research ideation, experiments, or core writing. All suggestions from LLMs were manually verified and edited by the authors prior to final inclusion.

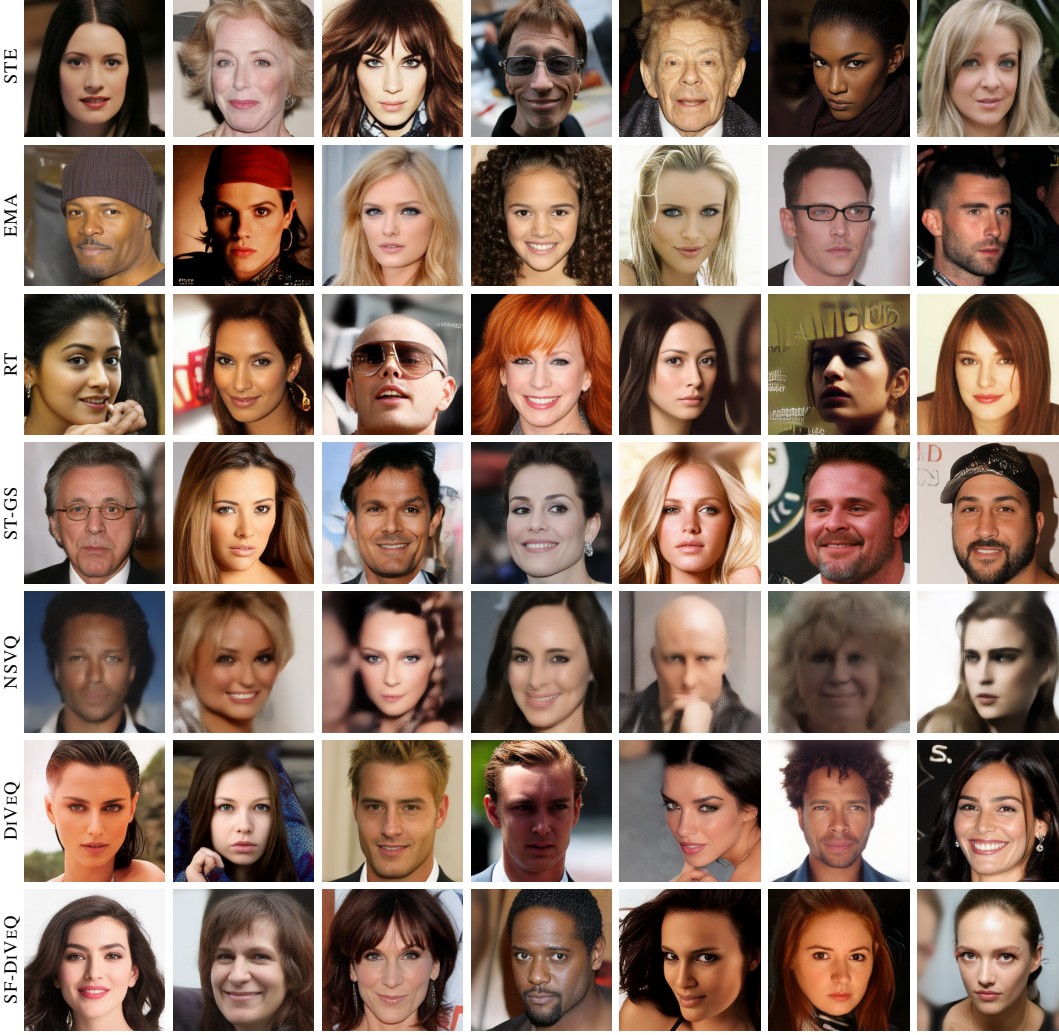

Figure 30: **Generation task.** Qualitative comparison of random generations for the CELEBA-HQ data set in the VQGAN generation task for different VQ optimization methods, with a 9-bit codebook (or codebook size of $K = 512$).

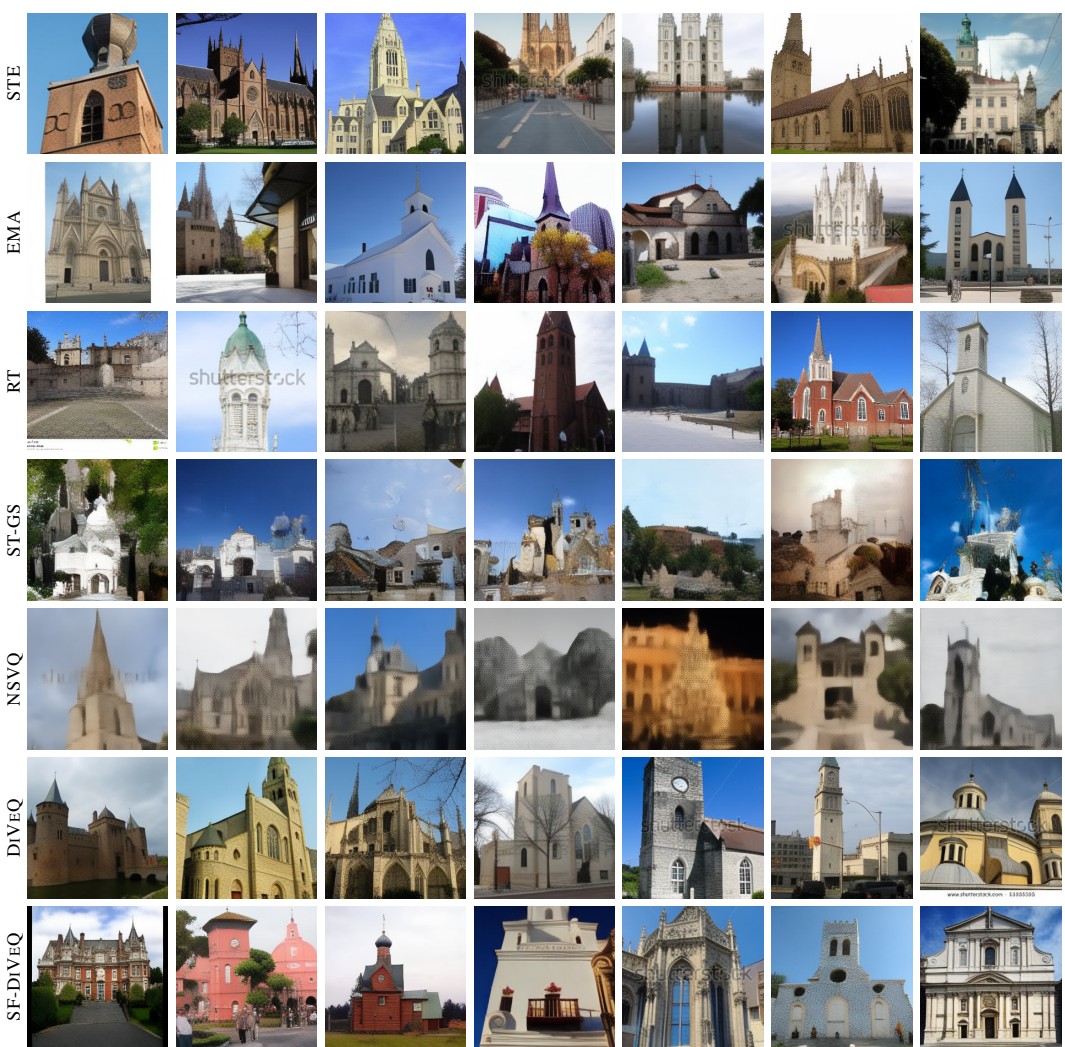

Figure 31: **Generation task.** Qualitative comparison of random generations for the LSUN Church data set in the VQGAN generation task for different VQ optimization methods, with a 9-bit code-book (or codebook size of $K = 512$).

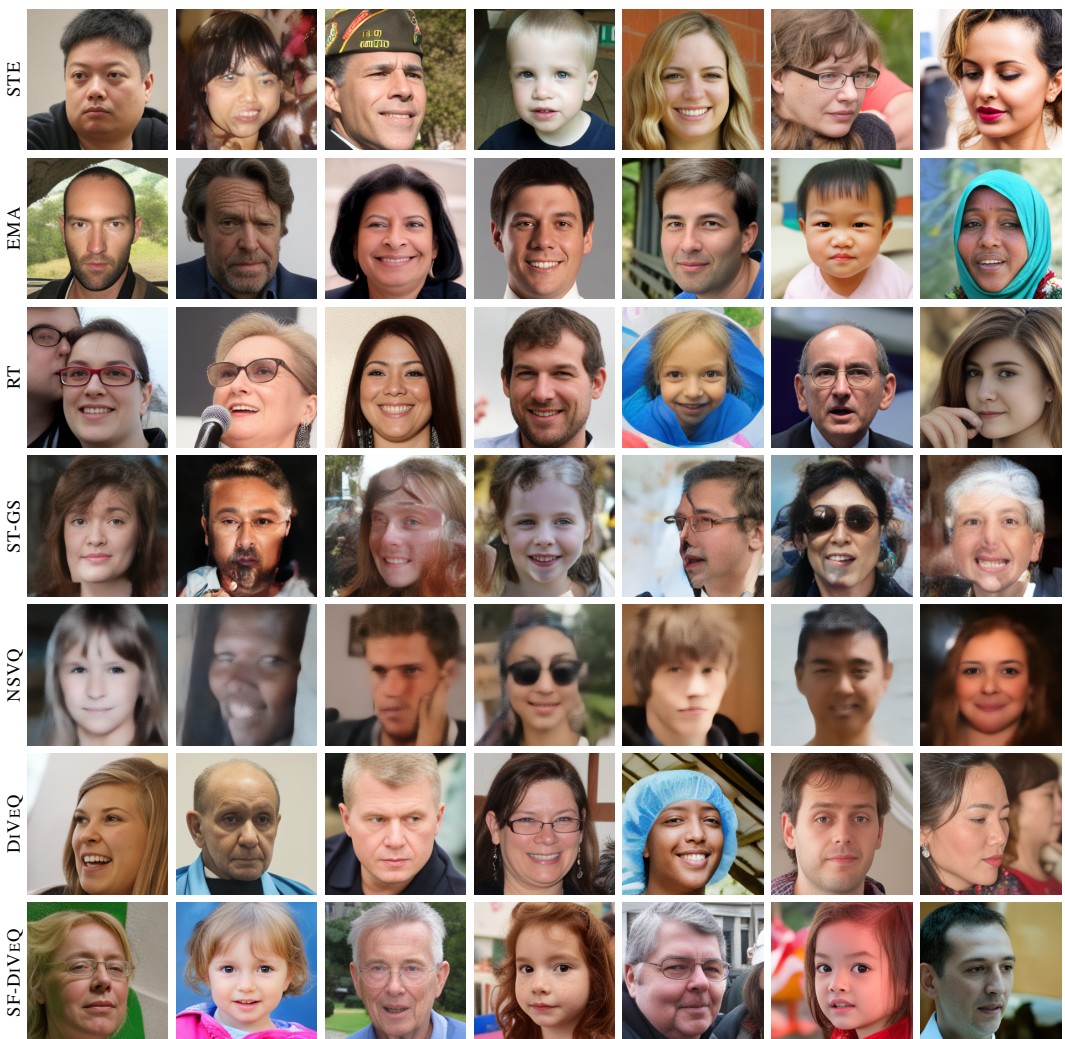

Figure 32: **Generation task.** Qualitative comparison of random generations for the FFHQ data set in the VQGAN generation task for different VQ optimization methods, with a 9-bit codebook (or codebook size of $K = 512$)

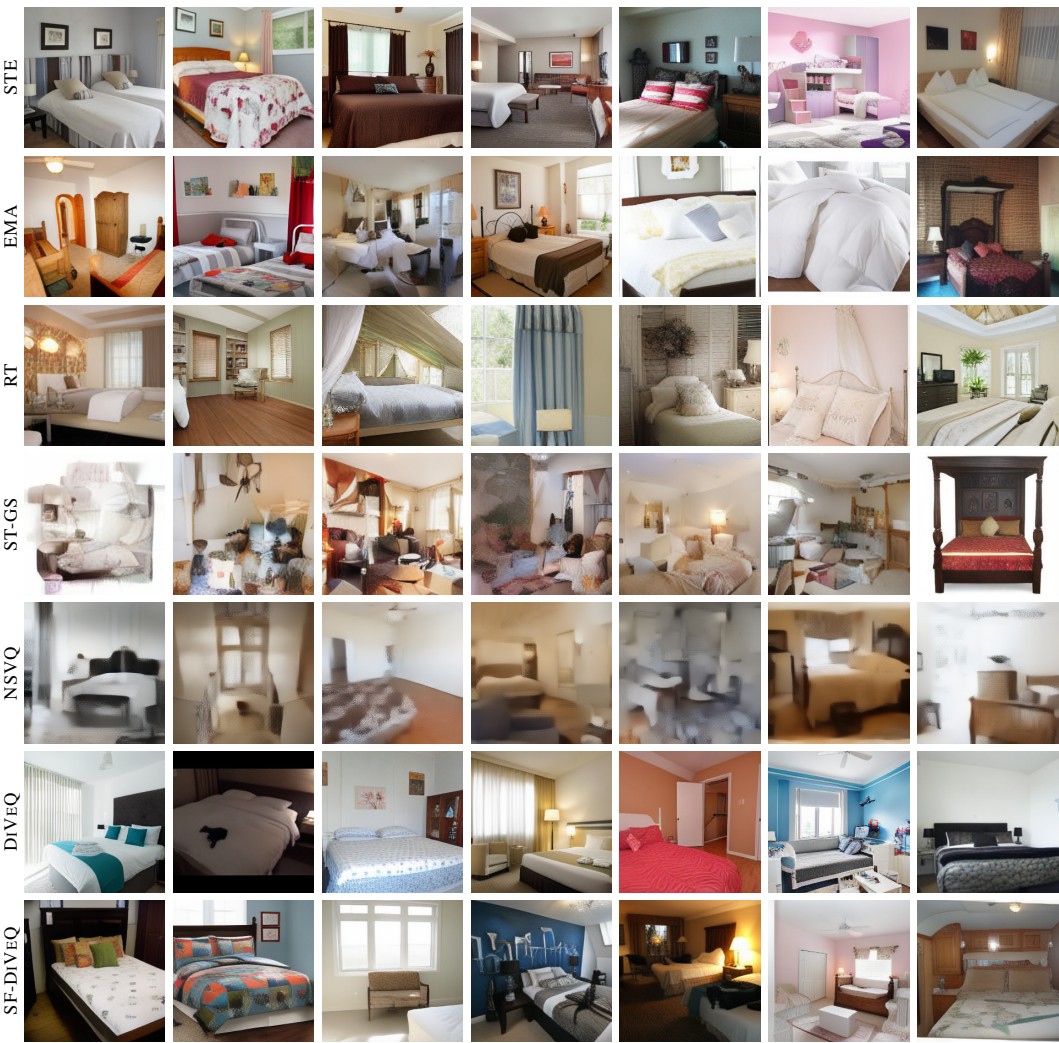

Figure 33: **Generation task.** Qualitative comparison of random generations for the LSUN Bedroom data set in the VQGAN generation task for different VQ optimization methods, with a 9-bit codebook (or codebook size of $K = 512$).

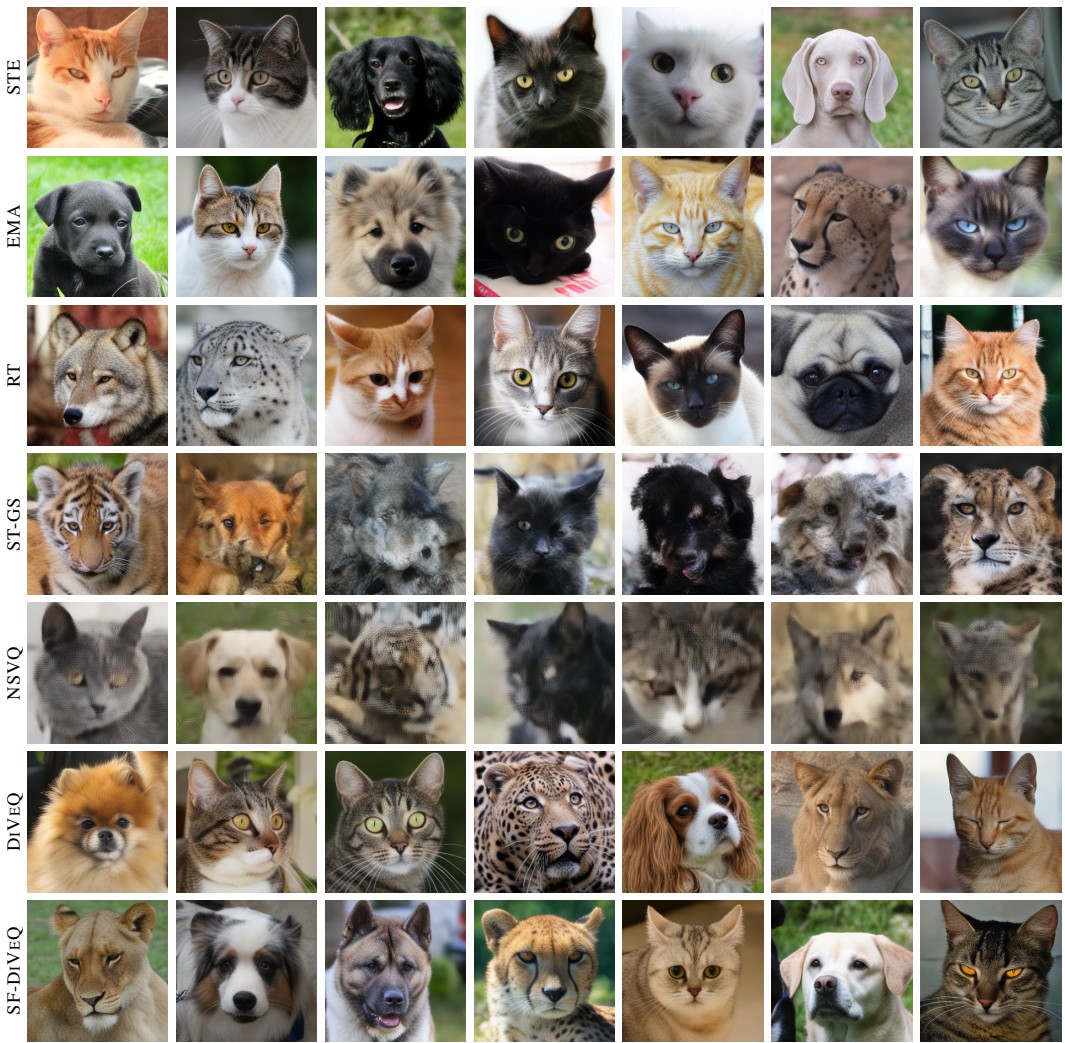

Figure 34: **Generation task.** Qualitative comparison of random generations for the AFHQ data set in the VQGAN generation task for different VQ optimization methods, with a 9-bit codebook (or codebook size of $K = 512$)

