# OpenReview forum: "DiVeQ: Differentiable Vector Quantization Using the Reparameterization Trick"
_ICLR.cc/2026/Conference — ICLR 2026 Poster_

### Official Review · Reviewer_Q4CB · 2025-10-20

**Soundness:** 3
**Presentation:** 4
**Contribution:** 4
**Rating:** 8
**Confidence:** 3

**Summary:**

This paper addresses the non-differentiability of vector quantization (VQ) by proposing Differentiable Vector Quantization (DiVeQ) and a variant, Space-Filling DiVeQ (SF-DiVeQ). The methods enable gradient flow while preserving hard assignments, with a directional reparameterization of the quantization error. Both methods train end-to-end without auxiliary losses or hyperparameter tuning. Key results on VQ-VAE compression and VQGAN generation tasks demonstrate that DiVeQ and SF-DiVeQ consistently outperform existing approaches in reconstruction and sample quality across multiple datasets.

**Strengths:**

1. DiVeQ introduces a novel directional reparameterization of the quantization error. The proposed techniques are elegantly derived from the reparameterization trick, and allow for true end-to-end training without auxiliary losses or complex schedules.
2. The authors perform a comprehensive evaluation across both image compression and image generation. The experiments are conducted on a wide range of standard, high-resolution datasets, and the proposed methods consistently outperform a diverse set of baselines.
3. The paper's claims are further confirmed by ablation studies, which analyze the impact of key factors like the noise variance, batch size, and learning rate.

**Weaknesses:**

1. In Page 4 and Figure 2, the NSVQ incurs a higher distortion than the true nearest-codeword assignment with a fixed probability of approximately 0.67. This claim can be extended beyond a 2D plane, where the probability of incurring higher quantization errors rapidly approaches one as the dimension $D$ increases. This can be mentioned to strengthen the argument.
2. In Section 4, the authors claim SF-DiVeQ avoids codebook misalignment (Fig. 4). However, the t-SNE plots show only a single run per method without providing statistical evidence (e.g., multiple runs with variance across seeds) to confirm that SF-DiVeQ consistently avoids misalignment. The authors are suggested to quantify the improvement across runs.
3. The arguments that DiVeQ requires no hyperparameter tuning seem to be overstated. DiVeQ involves the hyperparameter $\sigma^2$, and different values are used for compression ($10^{-3}$) and generation ($10^{-2}$).

**Questions:**

1. In Section 3.2, SF-DiVeQ is described as quantizing inputs to "random locations on the line connecting two subsequent codewords." Could the authors clarify the ordering of codewords along the space-filling curve?

---

> ### Author Response · Authors · 2025-11-20
>
> Thank you for the positive review and insightful comments. We address the remaining questions and suggestions below.
>
> >**Q1:** *In Section 3.2, SF-DiVeQ is described as quantizing inputs to "random locations on the line connecting two subsequent codewords." Could the authors clarify the ordering of codewords along the space-filling curve?*
>
> In a codebook with $K$ codewords of $\{c_1,c_2,...,c_K\}$, the order of the codewords is the actual existing order in the codebook matrix. For instance, for the codewords of $c_7$, $c_8$, $c_9$ (which refer to the 7th, 8th, 9th index of the codewords), the SF-DiVeQ curve is constructed by connecting $c_7$ to $c_8$, and by connecting $c_8$ to $c_9$. So, note that there is no connection between $c_7$ and $c_9$ (which refer to the 7th and 9th elements of the codebook). In other words, the index-wise subsequent codewords are connected to each other, and there is no connection between $c_i$ and $c_j$ if $|i-j|>1$. If the reviewer has any questions regarding how SF-DiVeQ maps the input to random locations on the curve (i.e., the dithering technique), please see the answer to Q3 of the reviewer 3NP6.
>
> >**W1:** *In Fig. 2, the NSVQ incurs a higher distortion than the true nearest-codeword assignment (...) claim can be extended beyond a 2D plane.*
>
> Thank you, it is true that this further strengthens our arguments. We have added a sentence regarding this in Sec. 2 (page 4).
>
> >**W2:** *The t-SNE plots show only a single run per method without providing statistical evidence. The authors are suggested to quantify the improvement across runs.*
>
> We fully agree. Please note that we use the t-SNE plots in Fig. 4 only as *per-method* qualitative illustrations of how the latent distribution $\mathcal P_z$ and codebook $\mathcal C_z$ relate to each other. Our conclusions about misalignment do not rely on t-SNE plots at all. We spot a misalignment in cases where the rate-distortion theory does not hold, or when quantitative metrics do not improve by the increase in codebook size in the image compression task. Therefore, t-SNE plots refer to the cases where rate-distortion does not hold in Fig. 26, and thus, t-SNE plots are provided only for illustration purposes. We have updated the "Misalignment of Codebook and Latent Representations" paragraph in Sec. 4 of the paper by highlighting that Fig. 4 should be interpreted as an intuitive visualization within each method, and we have added a short remark about the limitations of t-SNE in this context (see page 9).
>
> In all SF-DiVeQ experiments in the image compression task, we did not observe any misalignment cases (i.e., cases where increasing the codebook size does not lead to improvements in reconstruction quality). Therefore, that is why we claim that SF-DiVeQ is not prone to misalignment. So, this claim is based on all quantitative experiments done in the paper, and it is not based on t-SNE visualizations at all.
>
> >**W3:** *The arguments that DiVeQ requires no hyperparameter tuning seem to be overstated. DiVeQ involves the hyperparameter, and different values are used for compression and generation.*
>
> Our approach is only weakly sensitive to the value of $\sigma^2$. For this reason we originally described it as requiring 'no tuning', which we agree was too strong. To further show the weak sensitivity, we have added additional experiments and analysis in App. C.5 "Ablation on variance $\sigma^2$ of DiVeQ and SF-DiVeQ". The new experiments provided in Table 7 show that our proposed DiVeQ and SF-DiVeQ obtain similar FIDs across different variances. The main gist of the App. C.5 is that the value just needs to be small. Nevertheless, we have softened the statement regarding "no hyperparameters" and amended Sec. 1 and 5. We discuss and refer to these new experiments in the first paragraph of Sec. 4.

---

> ### Comment · Reviewer_Q4CB · 2025-11-23
>
> Thanks for the authors' response. My concerns have been addressed, and I would like to maintain my previous rating.

---

> > ### Author Response · Authors · 2025-11-24
> >
> > Thank you for going through our response, and thank you for recommending accepting our work.
> >
> > If you come to think of any further questions, we are happy to discuss them.

---

### Official Review · Reviewer_XAfN · 2025-10-28

**Soundness:** 2
**Presentation:** 3
**Contribution:** 1
**Rating:** 2
**Confidence:** 3

**Summary:**

The paper aims to improve vector quantization (VQ) in VQ-VAE compression and VQGAN generation. The paper first analyzes the baseline methods and compares them to identify gaps. Then, the paper proposes DiVeQ, which applies reparameterization tricks to the baseline NSVQ. The paper then proposes an improved version of DiVeQ by replacing the codebook replacement method with codebook interpolation, as suggested by another baseline, SFVQ. The paper conducts experiments on VQ-GAN image generation tasks at 256x256 resolution, and the experimental results show that the proposed method achieves better SSIM, PSNR, FID, and LPIPS scores.

**Strengths:**

* The paper provides an excellent background and related work section and presents a detailed comparison among them.
* The figures are helpful for understanding the idea.
* The experimental results show that DiVeQ achieves better results compared to the baselines.

**Weaknesses:**

* The contribution of the paper is not strong.

This reviewer believes that the proposed methods are minor extensions of the baselines, and the motivation and experimental results are not insightful enough. The paper proposes two methods, DiVeQ and SF-DiVeQ. DiVeQ is proposed by adding a direction vector to $v \sim N(0,\sigma^2I)$, and SF-DiVeQ is proposed by adding space-filling methods from SFVQ. In addition, although the paper’s title primarily emphasizes the reparameterization trick, the proposed methods are built on an existing baseline that already uses it. Also, since the motivation and experimental results do not present insightful analysis, this reviewer believes that the contribution of the paper is weak.
* The experiments section should be strengthened.

The paper primarily conducts experiments on VQ-GAN, generating 256x256-resolution images. From the experiments, the paper compares image-related metrics such as FID and PSNR. However, this reviewer believes that the experimental results section could be improved by using different models and tasks and by providing more detailed analysis beyond the numerical results. For example, many papers conducted experiments on various models and tasks such as recent stable diffusion models [C1], visual-language-action tasks [C2], and Gaussian splatting [C3]. As these papers are cited in the DiVeQ paper, the reviewer encourages broadening the experiments, as the proposed methods are drop-in replacements for standard VQ layers, as mentioned in the paper.

[C1] Zixin Zhu, Xuelu Feng, Dongdong Chen, Jianmin Bao, Le Wang, Yinpeng Chen, Lu Yuan, and Gang Hua. Designing a better asymmetric VQGAN for stable diffusion. arXiv preprint arXiv:2306.04632, 2023.

[C2] Seonghyeon Ye, Joel Jang, Byeongguk Jeon, Sejune Joo, Jianwei Yang, Baolin Peng, Ajay Mandlekar, Reuben Tan, Yu-Wei Chao, Yuchen Lin, Lars Liden, Kimin Lee, Jianfeng Gao, Luke Zettlemoyer, Dieter Fox, and Minjoon Seo. Latent action pretraining from videos. In International Conference on Learning Representations (ICLR), 2025.

[C3] Haishan Wang, Mohammad Hassan Vali, and Arno Solin. Compressing 3D Gaussian splatting by noise-substituted vector quantization. In Scandinavian Conference on Image Analysis, pp. 338–352. Springer, 2025.

**Questions:**

Please refer to the Weaknesses section. In addition, these are minor questions and points.
* Why are the proposed variants of DiVeQ and SF-DiVeQ in the supplementary materials, while they show better results?
* The baselines could be outdated.
* t-SNE plots are known to present misleading information, especially when compared to other t-SNE plots.

---

> ### Author Response · Authors · 2025-11-20
>
> Thank you for the review. We address your concerns point-by-point below.
>
> >**Q1:** *Why are the proposed variants of DiVeQ and SF-DiVeQ in the supplementary materials, while they show better results?*
>
> In case the reviewer is referring to the *detach* variants of DiVeQ/SF-DiVeQ in App. B.2: We put them in the appendix because these variants perform slightly worse than the DiVeQ/SF-DiVeQ based on the reparameterization trick in the image compression task. However, readers might find them interesting and useful for other applications they are working on.
>
> In case the reviewer is referring to the Residual VQ variants of DiVeQ/SF-DiVeQ in the App. C.9 (in the revised paper): We put them in the appendix because they are an extension on top of the main principles of DiVeQ/SF-DiVeQ, which are discussed in the main paper.
>
> We also moved these sections to the appendix due to space constraints.
>
> >**Q2:** *The baselines could be outdated.*
>
> We include what we believe is the standard and most relevant set of baselines: STE, EMA, RT (Rotation Trick)[2], ST-GS (Gumbel-softmax), and NSVQ. These are precisely the optimization schemes that underlie many recent VQ-based generative models; newer systems typically differ in encoder/decoder or transformer design, but still rely on these core VQ mechanisms. For instance, the well-known models of VQ-Diffusion [3] and MaskGIT [4] use STE, and Gumbel-softmax is used in [5], and NSVQ is used in [6], while RT [2] is a very recent VQ optimization method that we considered in our paper.
>
> To keep the comparison focused, we plug each method into the same VQ-VAE and VQGAN backbones and train them under identical conditions. We are not aware of a widely-used differentiable VQ baseline that we have omitted. If the reviewer has specific additional baseline methods in mind, we would be happy to hear these suggestions.
>
> [2] Fifty C., Junkins RG., Duan D., Iyengar A., Liu JW., Amid E., Thrun S., Re C. (2025). Restructuring Vector Quantization with the Rotation Trick. *In International Conference on Learning Representations.*
>
> [3] Gu S., Chen D., Bao J., Wen F., Zhang B., Chen D., Yuan L., Guo B. (2022). Vector quantized diffusion model for text-to-image synthesis. In *Proceedings of the IEEE/CVF conference on Computer Vision and Pattern Recognition*.
>
> [4] Chang H., Zhang H., Jiang L., Liu C., Freeman WT. (2022). MaskGIT: Masked generative image transformer. In *Proceedings of the IEEE/CVF conference on Computer Vision and Pattern Recognition*.
>
> [5] Zhang J., Zhan F., Theobalt C., Lu S. (2023). Regularized vector quantization for tokenized image synthesis. *In Proceedings of the IEEE/CVF Conference on Computer Vision and Pattern Recognition*.
>
> [6] Ye S., Jang J., Jeon B., Joo SJ., Yang J., Peng B., Mandlekar A., Tan R., Chao YW., Lin BY., Liden L. (2024). Latent Action Pretraining From Videos. In *CoRL 2024 Workshop on Whole-body Control and Bimanual Manipulation: Applications in Humanoids and Beyond*.
>
> >**Q3:** *t-SNE plots are known to present misleading information, especially when compared to other t-SNE plots.*
>
> We fully agree. For this reason, we use the t-SNE plots in Fig. 4 only as *per-method* qualitative illustrations of how the latent distribution $\mathcal P_z$ and codebook $\mathcal C_z$ relate to each other. Our conclusions about misalignment do not rely on t-SNE plots at all. We spot a misalignment in cases where the rate-distortion theory does not hold, or when quantitative metrics do not improve by the increase in codebook size in the image compression task. Therefore, t-SNE plots refer to the cases where rate-distortion does not hold in Fig. 26, and thus, t-SNE plots are provided for illustration purposes. We have updated the "Misalignment of Codebook and Latent Representations" paragraph in Sec. 4 of the paper by highlighting that Fig. 4 should be interpreted as an intuitive visualization within each method, and we have added a short remark about the limitations of t-SNE in this context (see page 9).

---

> > ### Author Response · Authors · 2025-11-20
> >
> > >**W1:** *The contribution of the paper is not strong.*
> >
> > We respectfully disagree and would like to clarify what we see as the main technical contributions.
> >
> > First, DiVeQ is not simply "NSVQ plus a direction vector." NSVQ samples an isotropic noise vector on a hypersphere, which we show analytically (Fig. 2) leads to an overshoot of the true nearest-codeword distortion with probability $\sim2/3$ in 2D and approaching $1$ in higher dimensions. In contrast, DiVeQ explicitly models *the quantization error itself* as a vector aligned with the nearest codeword and with magnitude equal to the input–codeword distance. This yields a differentiable surrogate that is geometrically consistent with the underlying nearest-neighbor operation, rather than optimizing a smoothed surrogate with systematically misaligned gradients. To our knowledge, this directional reparameterization of the quantization error has not been studied before.
> >
> > Second, SF-DiVeQ is more than "adding the space-filling method from SFVQ." The SFVQ proposed in [7] is applied only to pretrained models for interpretation purposes and is not intended to address the gradient collapse issue in VQ optimization. However, we get intuition from SFVQ [7] and proposed SF-DiVeQ that can pass the gradients through non-differentiable VQ by using our proposed directional reparameterization of the quantization process. We leverage the space-filling codebook ordering to reparameterize *along* the curve defined by codeword connections, thereby quantizing on line segments between neighboring codewords. This *(i)* reduces quantization error compared to hard nearest-codeword assignment and *(ii)* eliminates the need for ad-hoc codebook replacement by ensuring that gradients flow through all "edges" of the curve. Empirically, this leads to full codebook usage and removes the misalignment phenomena documented for several existing methods in the "Misalignment of Codebook and Latent Representations" paragraph in Sec. 4 (Fig. 4 and App. C.8).
> >
> > Third, beyond proposing DiVeQ and SF-DiVeQ, we provide a *systematic* comparative analysis of widely used VQ optimization schemes (STE, EMA, RT, ST-GS, NSVQ) under a unified framework with shared architectures, losses, and replacement strategies (Table 1, Sec. 4). Also, we present a comprehensive set of experiments with different ablations on batch size (App. C.3), learning rate (App. C.4), and codebook replacement (App. C.6) for all methods over a range of codebook sizes. This unifies several lines of work that have so far been evaluated in isolation and, we believe, provides useful insight for practitioners choosing a quantizer.
> >
> > We, indeed, build on existing tools, but the synthesis is novel, and the experiments show a clear distinction between NSVQ and our proposed DiVeQ.
> >
> > [7] Vali M. H., Bäckström T. (2023). Interpretable Latent Space Using Space-Filling Curves for Phonetic Analysis in Voice Conversion. In *Proceedings of Interspeech*.
> >
> > >**W2:** *The experiments section should be strengthened.*
> >
> > We agree that broad applicability is important, and we would like to clarify the scope of our current evaluation.
> >
> > Our focus in this paper is the *quantization layer itself*, not full downstream systems like stable diffusion or VLA agents. To isolate the effect of the quantizer, we evaluate DiVeQ and SF-DiVeQ in two canonical and widely used settings: VQ-VAE for image compression and VQGAN for image generation. Experiments are conducted on multiple standard high-resolution datasets (CelebA-HQ, FFHQ, AFHQ, LSUN Bedroom, LSUN Church), and we report both reconstruction quality (PSNR, SSIM, LPIPS) and generative quality (FID) across a range of codebook sizes.
> >
> > Beyond raw numbers, we also provide analysis: rate–distortion behavior as a function of codebook size, codebook–latent misalignment (Fig. 4 and App. C.8), codebook usage statistics, and comprehensive ablations on *(i)* batch size (App. C.3), *(ii)* learning rate (App. C.4), *(iii)* variance $\sigma^2$ (App. C.5), *(iv)* codebook replacement (App. C.6), and *(v)* SF-DiVeQ initialization (App. C.7). Also, in image compression task, all the results are averaged over three different individual runs for all experiments to reach statistically significant evaluations. These are precisely meant to give insight into *why* certain methods behave better or worse.
> >
> > Additionally, even if the method is applicable to Gaussian splatting, VLA agents, and diffusion models, these big systems have multitude of factors affecting the outcome, studying which would not be central for highlighting the VQ part alone. Thus, including such experiments would only be anecdotal or showcasing in nature.

---

> > > ### Author Response · Authors · 2025-11-26
> > >
> > > We hope that our rebuttal has addressed the reviewers' questions regarding our work. If there are any remaining concerns, we would be happy to discuss them during the ongoing discussion phase.

---

> > > > ### Author Response · Authors · 2025-12-03
> > > >
> > > > >**W2:** *The experiments section should be strengthened.*
> > > >
> > > > To demonstrate broader applicability beyond image compression and generation, we have added a DAC-based [8] speech coding experiment on the VCTK data set (**new Table 3 on p. 10 and details in App. A.3, with additional ablations in Tables 8–9 in App. C.3**). We train DAC with each VQ method and report LSD, MFCC distance, PESQ, and STOI between the original and decompressed speech. Across different codebook sizes and batch sizes, DiVeQ and SF-DiVeQ consistently achieve the best objective speech quality and avoid the misalignment failures that render some baseline reconstructions unintelligible. This supports that our methods transfer well to a standard speech coding benchmark.
> > > >
> > > > [8] Rithesh Kumar et al. (2023). High-fidelity audio compression with improved RVQGAN. In *Advances in Neural Information Processing Systems 36 (NeurIPS)*, pp. 27980–27993. Curran Associates, Inc.

---

### Official Review · Reviewer_3NP6 · 2025-10-30

**Soundness:** 3
**Presentation:** 3
**Contribution:** 3
**Rating:** 6
**Confidence:** 5

**Summary:**

Quantization is a mathematically non-differentiable process, which poses a major challenge for the end-to-end training of neural networks. To overcome this, this paper proposed a differentiable vector quantization (DiVeQ) using the reparameterization trick. The proposed DiVeQ ensures the hard assignment in the forward pass while allowing gradient flow to enable end-to-end training. Its variant, space-filling DiVeQ (SF-DiVeQ), alleviates codebook underutilization and achieves full utilization. The proposed methods excel the existing quantization method in both investigated image compression and generation tasks and can be easily used as a drop-in replacement for the standard vector quantization method.

**Strengths:**

1. The proposed DiVeQ addresses the gradient collapse problem by modeling quantization as the addition of a simulated quantization error to the input. A key improvement over previous noise-substitution vector quantization (NSVQ) is that the direction of the error vector is aligned with the selected codeword to guarantee a good reconstruction.
2. This paper also provides insights into existing quantization methods and establishes a basic understanding of the limitations.
3. The proposed SF-DiVeQ addresses quantization error and codebook underutilization by extending quantization from discrete codewords in the codebook to continuous line segments connecting neighboring codewords. SF-DiVeQ achieves full utilization and improved representational efficiency.
4. A codebook replacement algorithm was also given to enable stable codebook utilization and it can be applied to all investigated existing quantization methods.
5. The proposed methods do not require a specifically designed auxiliary loss or hyperparameter tuning compared to existing methods.
6. The proposed methods demonstrate consistent improvement over existing methods across multiple datasets on image compression and generation tasks. Comprehensive ablations on codebook size, batch size, learning rate, and the hyperparameter of the proposed methods are conducted.
7. The proposed methods can act as a drop-in replacement with minimal changes and the authors have investigated the performance within a residual vector quantization design.

**Weaknesses:**

1. This paper analyzed the limitations of NSVQ and its poor performance due to the high randomness. For the proposed DiVeQ, the paper highlighted the fundamental theory and the definition of the simulated error and how it improves quantization accuracy by adjusting the hyperparameter. But a more detailed theoretical analysis of convergence properties and gradient flow would be appreciated to provide more insights into the proposed method.
2. The proposed method approximates the quantization process by generating a directional noise vector and one component is sampled from a zero-mean Gaussian distribution. Adjusting the variance can affect the quantization accuracy (see Figure 3). Although the paper states that the variance parameter is not highly sensitive and a small value should be used, two different variance setups were used in the investigated compression and generation tasks. One can assume that some tuning or empirical selection is needed, especially in different domains or tasks. This is the same compared to existing methods, where hyperparameter tuning is needed.
3. The proposed SF-DiVeQ requires a specific and empirical initialization strategy. This introduces extra complexity and might increase implementation overhead if the codebook size is large. The results of SF-DiVeQ without this specific codebook initialization strategy are not discussed.
4. Similar to item #3, the recommended training setup for training SF-DiVeQ is to skip quantization for several initial epochs to stabilize the latent representation and then initialize the codebook vectors using the average of recent latent vectors. This process requires careful tracking of recent latent vectors and defining a specific number of training iterations to capture them, making it less of a drop-in replacement compared to existing methods that allow simple random initialization.
5. The proposed methods are only evaluated on image compression and generation tasks. Further investigation into speech/audio coding would be appreciated to confirm a broader applicability of the proposed methods.
6. The codebook replacement algorithm is used for all investigated methods. The proposed methods demonstrated good performance compared to existing exponential moving averages (EMA) and straight-through estimator (STE) methods by a smaller margin or performed comparably to EMA. This suggests that a well-tuned EMA can also deliver high-fidelity reconstruction. So, maybe, the codebook replacement algorithm plays a more important role in improving the overall performance while the contribution of the proposed DiVeQ is secondary.
7. Throughout the whole paper, the term 'VQ bitrate' is used to represent the codebook size. But bitrate refers to the number of bits used to represent the given bitstream per unit of time. For the image compression task, bitrate is measured in bits per pixel. In the paper, it is a misuse of the term 'bitrate' and it should be 'n-bit codebook' instead of 'VQ bitrate of n'.

**Questions:**

1. Could the authors provide a theoretical analysis regarding the convergence properties or gradient flow of the proposed methods?

2. The proposed SF-DiVeQ requires a specific initialization strategy. It would provide more insights if the authors could show the necessity of the pre-training step and the impact of the pre-training step on stabilizing the latent representation and improving the performance. Could the authors quantify the performance degradation when SF-DiVeQ uses the randomly initialized codebook? Is there a big performance drop or is it only marginal? Could the author also provide the extra complexity required by the initialization strategy?

3. For DiVeQ, it is straightforward to the nearest codeword during training. But in SF-DiVeQ, a dithered codebook sampled from two subsequent codewords is used to quantize the input. How do the authors select these two subsequent codewords during training?

4. Two different variance values were used for the investigated tasks. Could the authors explain how and why they chose different values for each task? Does this mean that a variance tuning step is required when applying to other domains, such as speech coding?

5. Could the author give more discussion regarding the strength of the proposed method as a well-tuned EMA/STE with the codebook replacement algorithm can also achieve comparable performance? What do we benefit from the proposed method? Or does the codebook replacement algorithm contribute more to the performance?

6. As mentioned in the weakness part, please change the term used to represent the codebook size. Codebook size is not the same as the bitrate.

7. For the ablation on variance values of the proposed method, the colors in Figure 20 are not easy to distinguish, especially for variances with small values. Could the author adjust it using different line types?

8. A few typo errors are found.
   For example, on page 9, 'but they are not homogeneously scattered In ST-GS,' probably should be 'but they are not homogeneously scattered. In ST-GS'.
   On page 16, 'we mask half the the true token indices forcing the'. Should be 'we mask half of the true token indices forcing the'.

---

> ### Author Response · Authors · 2025-11-20
>
> Thank you for the supportive and extensive review. We address your questions below.
>
> >**Q1/W1:** *theoretical analysis regarding the convergence properties or gradient flow of the proposed methods*
>
> The gradient geometry of NSVQ and DiVeQ can be analyzed through the following logic. Both NSVQ and DiVeQ are of the form $z_q=z+||c_i - z|| u$, where $u$ is a constant vector with respect to differentiation and $||u||=1$. In geometric terms, $z_q$ is thus a step $||c_i-z||$ away from $z$ in the direction of $u$. The distance between $z$ and $z_q$ is thus equal to the distance between $c_i$ and $z$,
>
> $$||z-z_q|| \equiv ||z-c_i||.$$
>
> A loss function that minimizes $||z-z_q||$ therefore also minimizes $||z-c_i||$, showing that the surrogate objective is geometrically aligned with the true nearest-codeword distortion.
>
> >**Q2/W3:** *(...) Could the authors quantify the performance degradation when SF-DiVeQ uses the randomly initialized codebook? Is there a big performance drop, or is it only marginal?*
>
> Firstly, custom initialization of SF-DiVeQ does not add implementation complexity, as we can capture the latent vectors from the latest 10--50 training batches using a simple conditional statement. Also, the number of training batches to capture is a suggestion, and we did not tune it; any value worked across all our experiments.
>
> Secondly, thanks for raising this question. We have added new experiments in App. C.7 (Fig. 24 and more importantly Fig. 25) in the paper by randomly initializing the SF-DiVeQ codebook. We show that for the SF-DiVeQ codebook, if we use learning rates that are multiple times higher than other model parameters, SF-DiVeQ with random initialization can reach similar (even slightly better in some cases) performance and perplexity (i.e., average codebook usage) compared to the custom initialization. We discuss and refer to these new findings in the "Codebook Replacement and Initialization" paragraph in Sec. 4.
>
> >**Q3:** *(...) in SF-DiVeQ, a dithered codebook sampled from two subsequent codewords is used to quantize the input. How do the authors select these two subsequent codewords during training?*
>
> In brief, at each specific training batch, a set of dithered codewords is generated, each of which is the random interpolation of subsequent original codewords $c_i$ and $c_{i+1}$. The input $z$ will be mapped to the closest dithered codeword, which is inherently an interpolation of two original subsequent codewords.
>
> Suppose we have a base codebook $C$ with $K$ codewords. For each training batch, SF-DiVeQ randomly samples $K-1$ interpolation values $\lambda$s from a uniform distribution of $U(0,1)$. Then, it generates a dithered codebook $C^d$ which has $K-1$ codewords each is created by $c_i^d = (1-\lambda_i)c_i + \lambda_i c_{i+1}$, where $c_i$ and $c_{i+1}$ are codewords from the base codebook $C$ and $c_i^d$ is the new generated dithered codeword at the current training batch such that $i\in\{1,2,...,K-1\}$. After generating dithered codewords for the current training batch, we compute the distance of the input $z$ to all $K-1$ generated dithered codewords $c_i^d$, and then we simply map $z$ to the closest dithered codeword. Then, for the next training batch, we generate new dithered codewords by using new $\lambda$s sampled from the uniform distribution. We refer you to Sec. 3.1 in paper [1] for more details.
>
> [1] Vali M. H., Bäckström T. (2025). Unsupervised Panoptic Interpretation of Latent Spaces in GANs Using Space-Filling Vector Quantization. *Transactions on Machine Learning Research (TMLR)*.
>
> >**Q4/W2:** *Parameter $\sigma^2$ is still a hyperparameter that can be tuned.*
>
> Our approach is only weakly sensitive to the value of $\sigma^2$. For this reason we originally described it as requiring 'no tuning', which we agree was too strong. To further show this, we have added additional experiments and analysis in App. C.5 "Ablation on variance $\sigma^2$ of DiVeQ and SF-DiVeQ". The new experiments provided in Table 7 show that our proposed DiVeQ and SF-DiVeQ yield similar FIDs across different variances. The main gist of the App. C.5 is that the value just needs to be small. Nevertheless, we have softened the statement regarding "no hyperparameters" and amended Sec. 1 and 5. We discuss and refer to these new experiments in a new paragraph in the beginning of Sec. 4.

---

> > ### Author Response · Authors · 2025-11-20
> >
> > >**Q5/W6:** *(...) maybe, the codebook replacement algorithm plays a more important role in improving the overall performance, while the contribution of the proposed DiVeQ is secondary.*
> >
> > Thanks for the concern you raised, which led to some interesting analysis in the appendix. We added new experiments in App. C.6 (Fig. 22) which show that by skipping codebook replacement, our proposed DiVeQ still obtains higher quality reconstructions than other VQ methods. In addition, for the special case of skipping codebook replacement, we analyzed the convergence of different VQ methods in Fig. 23. We discuss and refer to these new findings in the "Codebook Replacement and Initialization" paragraph in Sec. 4.
> >
> > >**Q6/W7:** *(...) please change the term used to represent the codebook size. Codebook size is not the same as the bitrate.*
> >
> > We have updated the terminology throughout the paper. We now use "Codebook size" when referring to the size of the codebook and  "Codebook bits" in the plots when referring to the powers of two.
> >
> > >**Q7:** *(...) the colors in Figure 20 are not easy to distinguish, especially for variances with small values. Could the author adjust it using different line types?*
> >
> > We changed the line styles for them, but it did not look good as the curves overlapped. We changed the colors and hope that helps different curves to be more distinguishable.
> >
> > >**Q8:** *A few typo errors are found.*
> >
> > Thank you. We have now fixed these. We also did some other typographical corrections when going through the paper.
> >
> > >**W4:** *Similar to item #3 (W3), the recommended training setup for training SF-DiVeQ requires careful tracking of recent latent vectors and defining a specific number of training iterations to capture them, making it less of a drop-in replacement compared to existing methods that allow simple random initialization.*
> >
> > (See also reply to Q2/W3 above) In our new experiments in App. C.7 (Fig. 24 and more importantly Fig. 25), we show that SF-DiVeQ with a random initialization can reach similar performance as SF-DiVeQ with custom initialization. Fig. 25 demonstrates the power of SF-DiVeQ in pulling inactive codewords inside the latent distribution without using any heuristic codebook replacement techniques and without the custom initialization. Therefore, the users can skip the custom initialization without any concerns if they find it complicated.
> >
> > >**W5:** *The proposed methods are only evaluated on image compression and generation tasks. Further investigation into speech/audio coding would be appreciated to confirm a broader applicability of the proposed methods.*
> >
> > We have provided a comprehensive set of experiments for both image compression and image generation tasks with different ablations on *(i)* batch size (App. C.3), *(ii)* learning rate (App. C.4), *(iii)* variance $\sigma^2$ (App. C.5), *(iv)* codebook replacement (App. C.6), and *(v)* SF-DiVeQ initialization (App. C.7) by using a wide range of codebook sizes. In addition, for all experiments in the image compression task, the results are averaged over three different individual runs to reach statistically significant evaluations. We believe this set of experiments is sufficient to substantiate the empirical advantages of our proposed methods in the considered settings.

---

> > > ### Author Response · Authors · 2025-11-26
> > >
> > > We hope that our rebuttal has addressed the reviewers' questions regarding our work. If there are any remaining concerns, we would be happy to discuss them during the ongoing discussion phase.

---

> > > ### Author Response · Authors · 2025-12-03
> > >
> > > >**W5:** *The proposed methods are only evaluated on image compression and generation tasks. Further investigation into speech/audio coding would be appreciated to confirm a broader applicability of the proposed methods.*
> > >
> > > To demonstrate broader applicability beyond image compression and generation, we have added a DAC-based [8] speech coding experiment on the VCTK data set (**new Table 3 on p. 10 and details in App. A.3, with additional ablations in Tables 8–9 in App. C.3**). We train DAC with each VQ method and report LSD, MFCC distance, PESQ, and STOI between the original and decompressed speech. Across different codebook sizes and batch sizes, DiVeQ and SF-DiVeQ consistently achieve the best objective speech quality and avoid the misalignment failures that render some baseline reconstructions unintelligible. This supports that our methods transfer well to a standard speech coding benchmark.
> > >
> > > [8] Rithesh Kumar et al. (2023). High-fidelity audio compression with improved RVQGAN. In *Advances in Neural Information Processing Systems 36 (NeurIPS)*, pp. 27980–27993. Curran Associates, Inc.

---

### Official Review · Reviewer_7ZbY · 2025-11-01

**Soundness:** 3
**Presentation:** 3
**Contribution:** 3
**Rating:** 6
**Confidence:** 3

**Summary:**

This work aims to overcome two problems that previous vector quantization works have had.
First, vector quantization uses a codebook that maps latents to the closest codeword, and the mapping operation is not mathematically differentiable.
That is, it suffers from a gradient collapse problem, which means the gradients will not pass through after the codebook in the backward pass.
Second, while training, several codewords remain inactive, and thus, they are not trained well.

To resolve the above problems, the paper proposes the following.
For the gradient collapse problem, the paper uses the reparameterization trick to align the direction of quantization error with the nearest codeword.
By representing a quantized vector with a latent vector and a directional noise and showing that this quantized vector is differentiable to the latent vector and the codeword, the paper defines vector quantization as a differentiable problem.

For inactive codewords, codebook replacement is widely used to replace inactive codewords with new codewords that are near active codewords.
This paper points out that codebook replacement is a heuristic method, and suggests that mapping latent to an interpolation of subsequent codewords, instead of mapping to a single codebook.

The above proposals yield high-quality images while achieving good indicators in both image reconstruction and image generation.

**Strengths:**

- The gradient collapse problem of vector quantization is a problem that many researchers have been trying to solve by detour for a long time, and the paper proposes to solve this problem directly.
- The paper summarizes well about prior works that try to solve the same problem and their problems.
- The paper explains the problem situation and the proposed method well, through mathematical formulas.
- The proposed method can help generate high-quality images in both image reconstruction and image generation tasks, while achieving better indicators such as FID and LPIPS than other prior works.

**Weaknesses:**

- The paper proposes SF-DiVeQ to eliminate the need for codebook replacement. However, the reviewer wonders if SF-DiVeQ can really supplant codebook replacement, in case both subsequent codewords are inactive.
- Unlike other prior works, the FID score of the proposed method becomes worse when the VQ bitrate becomes larger. It may imply that the proposed methods are not general.

**Questions:**

The paper proposes SF-DiVeQ to eliminate the need for codebook replacement. Then, while training VQ with DiVeQ, are inactive codewords replaced by codebook replacement techniques?

---

> ### Author Response · Authors · 2025-11-20
>
> Thank you for the thoughtful review. As you point out, our direct approach brings something new to the table. We address the concerns and questions below.
>
> >**Q1:** *(...) while training VQ with DiVeQ, are inactive codewords replaced by codebook replacement techniques?*
>
> Yes, for DiVeQ, we use the same replacement strategy as for the other VQ methods; this is described in Sec. 4 in the "Codebook Replacement and Initialization" paragraph. SF-DiVeQ is the only method that does not require replacement in our experiments.
>
> >**W1:** *(...) the reviewer wonders if SF-DiVeQ can really supplant codebook replacement, in case both subsequent codewords are inactive.*
>
> This is indeed a relevant question. Suppose we have three codewords $c_1, c_2, c_3$ such that only $c_1$ is active (i.e., located inside the latent distribution). As SF-DiVeQ maps the input $z$ on the line connecting $c_1$ and $c_2$ (i.e., $z_q=(1-\lambda)c_1 + \lambda c_2$), so we would have valid gradients also for the $c_2$. Then, after a period of training, to minimize the error between $z_q$ and the input $z$, the codeword $c_2$ will be automatically pulled inside the distribution. Now that $c_2$ becomes active, it will similarly pull the codeword $c_3$ inside the latent distribution. Therefore, the number of inactive codewords will decrease one by one until all codewords become active.
>
> This is also shown in our new experiments provided in the App. C.7 (Fig. 24 and more importantly Fig. 25), where we initialize the SF-DiVeQ codebook randomly. By random initialization, most of the codewords are initially out of the latent space, while they will be pulled inside the latent space during the course of training.
>
> >**W2:** *(...) the FID score of the proposed method becomes worse when the VQ bitrate becomes larger. It may imply that the proposed methods are not general.*
>
> For image compression, the DiVeQ/SF-DiVeQ curves in Fig. 6, 12, 13, 14, and 15 show consistent improvement by the increase in codebook size. For the VQGAN generation task (Table 2 and Table 6), when the bitrate grows, the differences become harder to pinpoint as many factors affect the FID score. This applies to all the methods, and there is no guarantee that FIDs improve with larger codebooks, especially when the transformer has to model a higher-entropy discrete index sequence. We include discussion on this in the "VQGAN Generation Results" paragraph in Sec. 4 that "FID value should not necessarily improve with the increase in codebook size K".
>
> In order to visualize the effect of a higher learning rate and a larger batch size, we added a second set of experiments to Table 2 and Table 6. With this new hyperparameter setting ($lr=2.5\cdot10^{-4}, \text{batch size}=32$), many methods simply fail (that is shown *in red*), and there is no clear consistency in *any* of the methods that a larger codebook would always increase the FID. For example, look at FIDs for STE, EMA, and ST-GS methods in Table 2 and Table 6, and the RT method for AFHQ and LSUN Church data sets in Table 6.

---

> > ### Author Response · Authors · 2025-11-26
> >
> > We hope that our rebuttal has addressed the reviewers' questions regarding our work. If there are any remaining concerns, we would be happy to discuss them during the ongoing discussion phase.

---

### Author Response · Authors · 2025-11-20

We thank all reviewers for the detailed and constructive feedback. We are encouraged that the paper is seen as directly tackling the long-standing gradient collapse problem in vector quantization [7ZbY], proposing a *"novel directional reparameterization of the quantization error"* that is *"elegantly derived from the reparameterization trick"* and allows *"true end-to-end training without auxiliary losses or complex schedules"* [Q4CB]. Reviewers also highlight that our methods *"excel the existing quantization method"* and can be used as "a drop-in replacement for the standard vector quantization method" [3NP6], and that the paper provides *"an excellent background and related work section"* with figures that are *"helpful for understanding the idea"* [XAfN].

The main concerns raised across reviews relate to (i) clarifying the novelty over NSVQ/SFVQ and the role of our reparameterization, (ii) the status of the variance $\sigma^2$ as a hyperparameter and the strength of our "no tuning" claim, (iii) the robustness of SF-DiVeQ to random initialization, (iv) the effect of codebook replacement in our DiVeQ and other methods, (v) the interpretation of the misalignment/t-SNE analysis, and (vi) the scope of the experimental evaluation. In the revised manuscript, we have:

* Strengthened the NSVQ discussion by explicitly extending the overshoot argument beyond 2D, noting that the probability of incurring higher distortion approaches one in higher dimensions.
* Clarified that $\sigma^2$ is the only additional scalar parameter, chosen once to be small ($\sigma^2 \leq 10^{-2}$) and reused across data sets, supported by new ablations in App. C.5; we have correspondingly softened our claims about "no hyperparameter tuning".
* Highlighted that the superior performance of DiVeQ is not tied to codebook replacement (new ablations in App. C.6).
* Expanded and better highlighted our treatment of SF-DiVeQ, including practical initialization and analysis of codebook usage (new ablations in App. C.7), and we stress that SF-DiVeQ does not require heuristic codebook replacement and empirically avoids misalignment across tested settings.
* Clarified that misalignment is quantitatively diagnosed via rate–distortion curves, while t-SNE plots in Fig. 4 are only used as per-method qualitative visualizations, not as cross-method quantitative evidence.
* Better explained the experimental scope: we focus on canonical VQ-VAE and VQGAN setups across several standard high-resolution image datasets, targeting widely used VQ optimization schemes (STE, EMA, RT, ST-GS, NSVQ), and we clarify that extending DiVeQ/SF-DiVeQ to other architectures (e.g., diffusion models, action models, 3D Gaussian splatting) can be done, but diverges from focusing on core VQ benchmarking.

We have also corrected minor typos, improved figure readability, and made small wording and notation adjustments throughout for clarity. Edits in the main paper are in blue. We respond to each reviewer point-by-point in the subsequent sections.

---

> ### Author Response · Authors · 2025-12-03
>
> Dear AC,
>
> Due to the OpenReview incident, the discussion was interrupted midway and only one reviewer was able to interact. To make things easier to follow, have already summarised the changes and the main learnings from the reviews in the message above.
>
> Additionally, Reviewers 3NP6 and XAfN also mentioned that it would be interesting to see how our proposed methods perform in applications beyond image reconstruction and generation. Reviewer 3NP6 (in W5) specifically suggested a **speech coding experiment**, which can be considered a standard benchmark task. We have now added the experiment results for this in **new Table 3 on p. 10 and the details in App. A.3, with additional ablations in Tables 8–9 in App. C.3**.
>
> Across different codebook sizes and batch sizes, our methods achieve the best objective speech quality (LSD, MFCC distance, PESQ, STOI) and avoid the misalignment failures that render some baseline reconstructions unintelligible (we also added speech samples to the supplementary material).
>
> With this final addition, **we believe all substantive reviewer requests have been met and their concerns addressed**.

---

### Meta-Review · Area_Chair_65pP · 2025-12-12

**Summary:**

Three out of four reviewers are positive with respect to the paper and contributions. The authors provided responses to most of the expressed concerns.

The ACs carefully read the paper, the reviews, and the authors' responses.
The ACs agee with the majority of the reviewers that the paper makes significant contributions and can be accepted. The authors are invited to benefit from the feedback received and further improve their camera ready paper.

**Reviewer Concerns:**

Three our of four reviewers are positive with respect to the paper and contributions. The authors provided responses to most of the expressed concerns.

The ACs carefully read the paper, the reviews, and the authors' responses.
The ACs agee with the majority of the reviewers that the paper makes significant contributions and can be accepted. The authors are invited to benefit from the feedback received and further improve their camera ready paper.

**Reviewer Scores:**

Three our of four reviewers are positive with respect to the paper and contributions. The authors provided responses to most of the expressed concerns.

The ACs carefully read the paper, the reviews, and the authors' responses.
The ACs agee with the majority of the reviewers that the paper makes significant contributions and can be accepted. The authors are invited to benefit from the feedback received and further improve their camera ready paper.

---

### Decision · Program_Chairs · 2026-01-26

Accept (Poster)